



# An advanced spatial co-registration of cloud properties for the atmospheric Sentinel missions: Application to TROPOMI

Athina Argyrouli[1,2], Diego Loyola[2], Fabian Romahn[2], Ronny Lutz[2], Víctor Molina García[2], Pascal Hedelt[2], Klaus-Peter Heue[2], and Richard Siddans[3]

[1]Technical University of Munich (TUM), Chair of Remote Sensing Technology, School of Engineering and Design, Munich, Germany
[2]German Aerospace Center (DLR), Remote Sensing Technology Institute (IMF), Wessling, Germany
[3]Rutherford Appleton Laboratory (RAL), Chilton OX11 0QX, United Kingdom

**Correspondence:** Ronny Lutz (Ronny.Lutz@dlr.de)

**Abstract.** The retrieval of cloud parameters from the atmospheric Sentinel missions require Earth reflectance measurements from a set of spectral bands. Frequently, the ground pixel footprints of the involved spectral bands are not fully aligned and therefore, special treatment is required within the operational algorithms. This so-called inter-band spatial mis-registration of passive spectrometers is present when the Earth reflectance measurements in different spectral bands are captured by different

spectrometers. The cloud retrieval algorithm requires reflectance measurements in the UV (ultraviolet)/VIS (visible) band, where the first cloud parameter (i.e., radiometric cloud fraction) is retrieved from the OCRA (Optical Cloud Recognition Algorithm) algorithm. In addition, Earth reflectances in the NIR (near-infrared) band are needed for the retrieval of two additional cloud parameters (i.e., cloud height and cloud albedo or cloud-top height and optical thickness) from the ROCINN (Retrieval of Cloud Information using Neural Networks) algorithm. In the former TROPOMI (TROPOspheric Monitoring Instrument)/S5P

(Sentinel-5 Precursor) retrieval, a co-registration scheme of the derived cloud parameters from the source band to the target band based on pre-calculated mapping weights from UV/VIS to NIR, and vice versa, is applied. In this paper we present a new scheme for the co-registration of the TROPOMI cloud parameters using collocated VIIRS (Visible Infrared Imaging Radiometer Suite)/SNPP (Suomi National Polar-orbiting Partnership) information. A great benefit of the new co-registration scheme based on the VIIRS data is that it improves the overall quality of the TROPOMI cloud products and, in addition, it

allows the re-construction of the cloud parameters on the first UV/VIS detector pixel, which was impossible with the former scheme based on the static mapping tables. The latter practically means that a significant number of valid data points are added to the TROPOMI cloud, total ozone, $SO_2$ and HCHO product since November $26^{th}$ 2023 (orbit 31705), when the UPAS version 2.6 with the new co-registration scheme was activated operationally. From a comparison analysis between the two techniques, we found that the largest differences mainly appear for inhomogeneous scenes. From a validation exercise of

TROPOMI against VIIRS in the across-track flight direction, we found that the old co-registration scheme tends to smooth out cloud structures along the scanline, whereas such structures can be maintained with the new scheme. The need to implement a similar inter-band spatial co-registration scheme is foreseen for the Sentinel-4/MTG-S (Meteosat Third Generation - Sounder) and Sentinel-5/MetOp-SG (Meteorological Operational Satellite - Second Generation) missions. In the case of Sentinel-4 in-





**Table 1.** Spectral information for the TROPOMI spectrometers: The reflectance measurements are organized according to the 8 spectral bands BD1-BD8 covering wavelength ranges from the UV, VIS, NIR and SWIR spectral windows.

| Spectrometer | UV-1 | UV-2 | VIS-1 | VIS-2 | NIR-1 | NIR-2 | SWIR-1 | SWIR-2 |
|---|---|---|---|---|---|---|---|---|
| Wavelength range [nm] | 267-300 | 300-332 | 305-400 | 400-499 | 661-725 | 725-786 | 2300-2343 | 2343-2389 |
| Band ID | BD1 | BD2 | BD3 | BD4 | BD5 | BD6 | BD7 | BD8 |

strument, the external cloud information will originate from collocated FCI (Flexible Combined Imager) data, on board the

MTG-I (Meteosat Third Generation - Imager) satellite.

# 1 Introduction

The operational algorithms for the retrieval of cloud parameters from the atmospheric Sentinel missions make use of Earth-shine reflectance measurements in the spectral window of UV, VIS and NIR. Often those reflectances are captured from

different spectrometers. For instance, the TROPOMI payload on board Sentinel-5 Precursor covers four distinct spectrometers (see Table 1) and each spectrometer of them is split electronically in two bands (i.e., UV-1 and 2, VIS-1 and 2, NIR-1 and 2, and shortwave infrared (SWIR) 1 and 2), see Veefkind et al. (2012). Using different spectrometers leads to different ground pixel footprints for the several bands that are not perfectly aligned, which is called inter-band spatial mis-registration. The ground pixel footprints mis-alignment is inter-connected to the spatial resolution of UV-VIS-NIR TROPOMI measurements, which is

equal to 3.5 x 5.5 km$^2$ (since August 2019) at nadir. The original spatial resolution in the along-track direction was 7 km, but the finer resolution of 5.5 km was introduced in August 2019 when the co-addition period for the Earth radiance measurements decreased by about 200 ms. The fine spatial resolution of 3.5 km is met in the center of the swath and in a large area around it.

The TROPOMI operational cloud algorithms OCRA/ROCINN (Loyola et al., 2018) have a big heritage from prior missions which have already been applied operationally to a large number of instruments starting with GOME (Global Ozone Monitoring

Experiment) on ERS-2 (European Remote Sensing Satellite) (Loyola et al., 2010). OCRA/ROCINN, described in Section 2, has been adapted for several follow-up missions including SCIAMACHY (SCanning Imaging Absorption spectroMeter for Atmospheric CartograpHY) on ENVISAT (ENVIromental SATellite) (Loyola, 2004), the GOME-2 instruments on board MetOp-A/B/C (Meteorological Operational satellite) (Lutz et al., 2016), and the EPIC (Earth Polychromatic Imaging Camera) instrument on the DSCOVR (Deep Space Climate Observatory) satellite, located at the Lagrangian point L1 (Molina García,

2022). Furthermore, OCRA/ROCINN will be applied operationally to the Sentinel-4 instrument.





## 2 The operational cloud algorithm

The operational processing of cloud products under DLR responsibility is performed using the UPAS (Universal Processor for Atmospheric Spectrometers) system. The two-step algorithm used for the UPAS cloud processing makes possible the simultaneous retrieval of three cloud properties as described in (Loyola et al., 2018). The first step is to derive the radiometric

cloud fraction $f_c$ in the UV/VIS spectral region. The OCRA algorithm retrieves the cloud fraction from the total measured reflectance by considering that the measured reflectance contains two contributions; one from the cloud-free background and a second one from the clouds. OCRA requires the clear-sky reflectance maps obtained from the same instrument (Lutz et al., 2016). The second step is to retrieve two additional cloud parameters within the O$_2$ A-band window (Schuessler et al., 2014) using the ROCINN algorithm. By using the independent pixel approximation (IPA) concept (Cahalan et al., 1994; Chambers

et al., 1997), the sun-normalized radiances can be expressed as the summation of two components for the cloud-free and cloudy part of the pixel using the retrieved OCRA cloud fraction (see Equations 1 and 2). Several atmospheric conditions, with and without clouds, are simulated using LIDORT (Linearized Discrete Ordinate Radiative Transfer radiances) (Spurr, 2006). For the cloudy skies, the simulations are performed for two different cloud models. The Cloud-as-Reflecting-Boundaries (CRB) model is a simplistic approach which assumes that the cloud performs as a Lambertian reflector. The retrieved cloud parameters

from the CRB model are the cloud albedo $A_c$ and the cloud height $Z_c$. Provided that this model contains a cloud which behaves like a simple reflecting boundary and does not have any geometrical extend, the retrieved height should not be considered as the height of the cloud top but the height at the radiometric middle of the cloud. A more sophisticated approach, called Cloud-As-Layer (CAL) parameterizes the cloud as a layer of liquid water particles with their scattering properties derived from the Mie theory (Van de Hulst, 1957; Bohren and Huffman, 1983). In this model, the cloud has a predefined geometrical thickness. The

retrieved quantities are the cloud-top height $Z_{ct}$ and the cloud optical thickness $\tau_c$. The following mathematical expressions refer to the simulated CRB and CAL sun-normalized radiances with $R_s$ being the radiance from the ground and $R_c$ the radiance from the cloud:

$$R_{sim}^{CRB}(\lambda) = (1 - f_c) R_s(\lambda, \theta, A_s, Z_s) + f_c R_c(\lambda, \theta, A_c, Z_c) \tag{1}$$

$$R_{sim}^{CAL}(\lambda) = (1 - f_c) R_s(\lambda, \theta, A_s, Z_s) + f_c R_c(\lambda, \theta, \tau_c, Z_{ct}, Z_{cb}, A_s, Z_s) \tag{2}$$

where $Z_s$ is the surface height, $A_s$ the surface albedo, $\theta$ the path geometry and $Z_{cb}$ the cloud bottom height, which is fixed at 1 km below the cloud-top height.

Table 2 summarizes the retrieved parameters from the operational cloud algorithm with the usual abbreviation notation and the corresponding mathematical symbol.

In the UPAS environment, LIDORT simulations are parameterized using Neural Networks (NNs) in order to speed up the

forward model simulations and be able to process in near-real-time (NRT) the Big Data of TROPOMI. For each input parameter (like $Z_{ct}$, $\tau_c$ etc.), a range is predefined and a large number of samples is generated using the smart sampling technique (Loyola



**Table 2.** List of cloud parameters, abbreviations and mathematical symbols referring to the operational cloud algorithm.

| Parameter (Abbreviation) | Retrieval Algorithm | Symbol |
|---|---|---|
| Cloud Fraction (CF) | OCRA | $f_c$ |
| Cloud Height (CH) | ROCINN_CRB | $Z_c$ |
| Cloud-Top Height (CTH) | ROCINN_CAL | $Z_{ct}$ |
| Cloud Albedo (CA) | ROCINN_CRB | $A_c$ |
| Cloud Optical Thickness (COT) | ROCINN_CAL | $\tau_c$ |

et al., 2016). Then, the training set for the NN is generated by computing simulated radiances for all the sampling sets. This part is the most computationally expensive as it requires line-by-line LIDORT calculations but it is only done once and offline. The accuracy of the NN is assessed by comparing the forward model simulations with samples not used in the NN training.

## 3 Special treatment of the mis-registration within the OCRA/ROCINN algorithm tandem: application on existing mission

Among others, the OCRA/ROCINN algorithm is the operational algorithm for the TROPOMI L2 cloud product within the Sentinel-5 Precursor mission. This section describes the old co-registration approach and introduces the new approach which is implemented on top of the old co-registration scheme for the operational S5P cloud algorithm. The mis-alignment of TROPOMI ground pixel footprints from the UV/VIS and NIR spectrometers is illustrated in Fig. 1. The ground pixel footprints of the detector rows 15-25 are shown for 5 continuous scanlines. Each TROPOMI scanline contains 450 pixels in BD3 and 448 pixels in BD6. In general, the BD6 ground pixel footprints appear shifted towards the East w.r.t. the BD3 ground pixel footprints. The spatial mis-alignment in the across-track direction is not a fixed number but instead it depends on the detector row and the respective pixel size. The ground pixel size is 3.5 km in the center of the swath and in a large area around it, but it becomes larger towards the edges of the swath due to the Earth's curvature and the instrument nadir angle. The so-called binning factors are selected to optimize the signal-to-noise ratio per pixel but with the aim to obtain a ground pixel size in the across-track direction with the minimum dispersion. For the TROPOMI radiance measurements in BD3, BD4 and BD6, which are inputs to OCRA/ROCINN, a binning factor of 2 is used in the center and in a large region around it resulting to the ground pixel size of 3.5 km. At the edges of the swath the binning factor is reduced from 2 to 1 and this results in a ground pixel size of 15 km (KNMI, 2022). The smallest mis-placement in the across-track direction is found at the center of the swath and it is about half a detector pixel, which is translated to about 1.75 km at nadir. Higher mis-alignment between BD3 and BD6 ground pixel footprints, which can reach up to about 4 km, is present at the east edge of the swath. The static mapping tables described in Section 3.1 indicate a mis-match in the in-flight direction, which on average is very small compared to the ground pixel



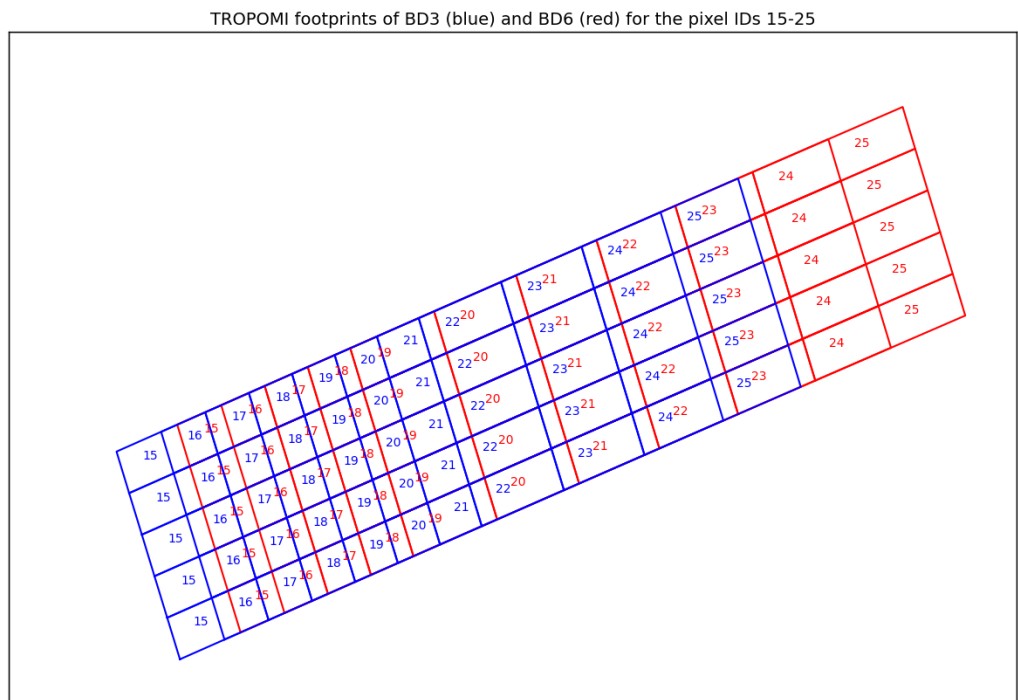

**Figure 1.** TROPOMI ground pixel footprints for the VIS-1 and NIR-2 spectrometers. The Band IDs for TROPOMI are descibed in Table 1:
BD3 (with blue) refers to VIS-1 spectrometer and BD6 (with red) refers to NIR-2 spectrometer.

size (i.e., roughly four orders of magnitude smaller) and thus it can be neglected. Therefore, the co-registration only needs

100     to be performed in the across-track direction. The complementary instrument, which is used for the treatment of the spatial

mis-registration of TROPOMI, is VIIRS on board the Suomi National Polar-orbiting Partnership. S5P satellite is located at

a low Earth orbit (LEO) and crosses the Equator in an ascending node at 13.30 h mean local solar time. This facilitates the

so-called loose formation operation with the SNPP spacecraft, which is also part of the A-train constellation and with only

3 to 5 minutes time difference from S5P. The spatial resolution of VIIRS at nadir is 750 m. The VIIRS cloud products are

re-gridded to the TROPOMI footprints as part of the S5P-NPP Cloud processor (Siddans, 2016). The pioneer methodology to

improve the existing co-registration scheme from the UV/VIS to NIR and the NIR to UV/VIS using collocated imager data is

presented in Sect. 3.3.1 and 3.3.2, respectively.



## 3.1 Previous treatment of the spatial mis-registration in the operational UPAS system

Due to the spatial mis-registration between the TROPOMI BD3/BD4 and BD6 bands, the operational cloud product contains a
flag called Cloud Co-registration Inhomogeneity Flag (CCIF). This flag is raised after the Cloud Co-registration Inhomogeneity
Parameter (CCIP), which is defined as the weighted averaged gradient of cloud fractions

$$CCIP_j = \frac{\sum_i w_{ij} |f_{ci} - f_{cj}|}{\sum_i w_{ij}}, \tag{3}$$

where the weights $w_{ij}$ correspond to the co-registration mapping values between UV bands (source, index i) and the NIR
band (target, index j). The CCIF is raised if the CCIP is larger than 0.4. The aforementioned threshold has been selected based
on tests from VIIRS cloud product re-sampled to the TROPOMI spatial grid.

Since UPAS version 2.0, the co-registration method is based on pre-calculated mapping weights (Sneep, 2015) between
BD3 and BD6 as illustrated in Fig. 2. This method for combining information from different bands is based on the fractions of
overlapping areas between the source and target pixels. The weights sum up to a total 1.0 and the most common situation is
that two source pixels contribute to the target pixel with very few exceptions, which are discussed later in the following Sect.
3.3.1 and 3.3.2. When the co-registration is done from BD3 to BD6, the BD6 target pixels have a difference of 1, 2 or 3 pixels
towards the East direction. When the co-registration is done from BD6 to BD3, the BD3 target pixels have a difference of 1, 2
or 3 pixels towards the West direction.

## 3.2 Evaluation of OCRA/ROCINN cloud properties for the TROPOMI instrument

Recent validation studies of the TROPOMI cloud properties against other satellite sensors (i.e., VIIRS, OMI (Ozone Monitoring
Instrument), MODIS (Moderate Resolution Imaging Spectroradiometer)) and the ground-based CloudNet network discussed
the similarities and differences between VIIRS and TROPOMI cloud parameters (Compernolle et al., 2021). The VIIRS geo-
metrical cloud fraction is usually higher than the OCRA radiometric cloud fraction because of the different definition but there
is an analogy between the two cloud fractions. One exception for the positive differences between VIIRS and OCRA cloud
fraction is the sun-glint region, where the dark ocean is perceived as a bright surface and very often misinterpreted as clouds.
The magnitude of the sun-glint effect and the affected area depends on the smoothness of the ocean, which is determined by the
wind properties over the ocean surface (Cox and Munk, 1954). The operational S5P cloud products include a flag indicating the
occurrence of sunglint. Similarly, the cloud height derived from TROPOMI is usually below the cloud-top height from VIIRS
because the infra-red bands of VIIRS are more sensitive to clouds than the UVN bands from TROPOMI.

The need to handle the spatial mis-registration between TROPOMI UV/VIS and NIR bands with a more dynamic and ad-
vanced approach was highlighted after the evaluation of OCRA/ROCINN cloud properties for the TROPOMI instrument. From
recent inter-comparison studies between TROPOMI cloud products, with the focus on cloud properties needed for trace gas
retrievals, showed that the co-registration has an impact along the cloud edges (Latsch et al., 2022). The proper co-registration
of the cloud properties is not only required for the improvement of the operational TROPOMI cloud product itself, but for the





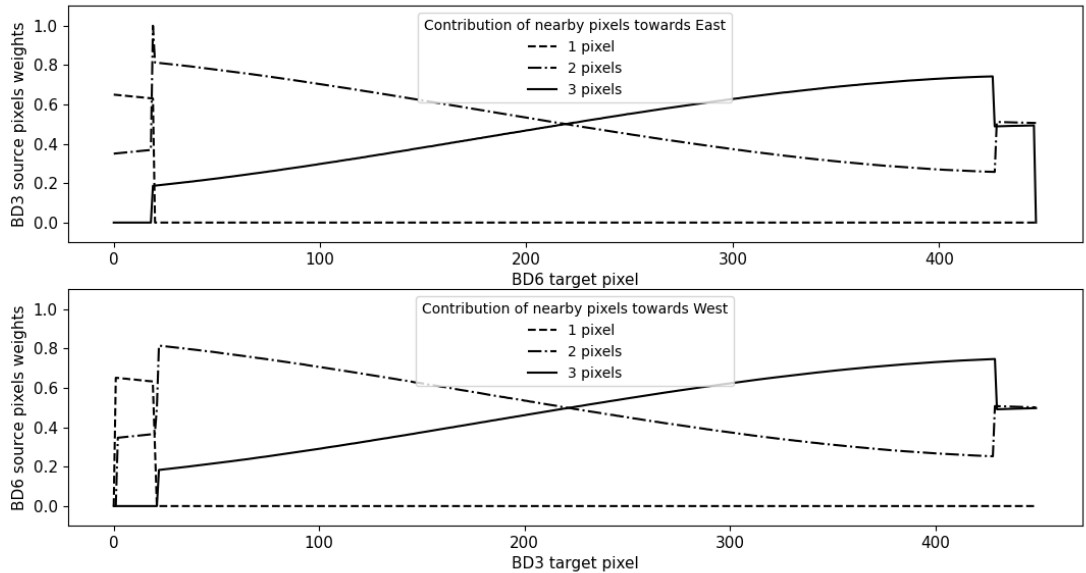

**Figure 2.** Weights from the static mapping tables (Sneep, 2015) used in the old co-registration scheme for Sentinel-5P. The upper panel refers to the mapping from UV source band to NIR target band. The lower panel refers to the mapping from NIR source band to UV target band.

direct impact that clouds have on the accurate retrieval of trace gases including total ozone (Spurr et al., 2021), tropospheric
ozone (Heue et al., 2018), HCHO (De Smedt et al., 2018) and $SO_2$ (Theys et al., 2017).

### 3.3  Advancement in the co-registration approach with the synergistic use of VIIRS cloud data

VIIRS has a much finer spatial resolution than TROPOMI, which is 750 m at nadir. With this high spatial resolution, VIIRS
captures small-scale cloud structures. VIIRS collects measurements in several spectral windows: VIS/NIR band, mid-IR and
LW IR, which makes its cloud product more sensitive to optically thin ice clouds.

The VIIRS cloud products are re-gridded to the TROPOMI footprints as part of the S5P-NPP Cloud processor (Siddans,
2016). An auxiliary product, which contains cloud information relevant to each TROPOMI ground pixel, can be derived from
observations captured by the VIIRS instrument. This operational L2 auxiliary product is called S5P-NPP Cloud product and
developed by the Rutherford Appleton Laboratory (Siddans, 2022). In this work, we present how the S5P-NPP cloud data is
used for the co-registration of the cloud product from BD3 (UV/VIS) to BD6 (NIR) and vice versa. The S5P-NPP cloud product
accepts as inputs a set of cloud-related VIIRS EDRs (Environmental Data Records): (a) the CloudMask which is necessary
for the co-registration of OCRA cloud fraction, (b) the CloudHeight EDR mandatory for the co-registration of ROCINN
cloud height and (c) the CloudDCOMP (Daytime Cloud Optical and Microphysical Properties) EDR for the co-registration of
ROCINN cloud albedo/optical thickness.





The VIIRS Enterprise Cloud Mask (ECM) describes the area of the earth's horizontal surface that is masked by the vertical
projection of detectable clouds (Heidinger and Straka, 2020). The ECM combines spectral and spatial tests to produce a 4-
level classification of cloudiness of the ECM cloud mask $\delta_{jk}^{CM=X}$, where X is any of the following categories: confidently
clear, probably clear, probably cloudy, confidently cloudy. Apart from solar reflectances in the VIS, the ECM makes use of
spectral channels in the IR that are more sensitive to clouds. The retrieval method is based on a naive Bayesian approach as part
of a library of machine learning (ML) methods, already well applied within Pathfinder Atmospheres Extended (PATMOS-x)
(Heidinger et al., 2012).

For the co-registration of TROPOMI cloud fraction, an equivalent VIIRS cloud fraction $M_c$ can be calculated as the number
of confidently cloudy pixels divided by the sum of all four cloudiness classes:

$$M_c = \frac{\delta_{jk}^{ConfidentlyCloudy}}{\delta_{jk}^{ConfidentlyCloudy} + \delta_{jk}^{ProbablyCloudy} + \delta_{jk}^{ConfidentlyClear} + \delta_{jk}^{ProbablyClear}}. \tag{4}$$

VIIRS cloud-top height is defined for each cloud-covered Earth location as the set of heights above mean sea level of the
tops of the cloud layers overlying the location (Heidinger et al., 2020). The Cloud Height Algorithm (ACHA) has already been
applied for the retrieval of the cloud height property from several sensors like MODIS and GOES-16/17 (US Geostationary
Operational Environmental Satellite R-series) ABI (Advanced Baseline Imager). ACHA makes use of only infrared channels
in order to provide consistent products for both day and night time as well as the terminator conditions. It uses an analytical
radiative transfer model embedded into an optimal estimation retrieval approach (Rodgers, 1976). The primary retrieved cloud
property is the cloud-top temperature and, at a later step, the cloud-top pressure and cloud-top height are derived from the
atmospheric temperature profile based on the Numerical Weather Prediction (NWP) data. For the co-registration of ROCINN
CRB cloud height and ROCINN CAL cloud-top height, the cloud-top height variable in the EDR CloudHeight can be directly
used.

VIIRS cloud optical thickness is defined as the optical thickness of the atmosphere due to cloud droplets, per unit cross
section, integrated over every distinguishable cloud layer and all distinguishable cloud layers in aggregate, in a vertical column
above a horizontal cell on the Earth's surface (Walther and Straka, 2020). The COT together with the Effective Particle Size
and Liquid/Ice Water Path are the cloud properties retrieved from the Daytime Cloud and Optical and Microphysical Properties
(DCOMP) algorithm (Walther and Heidinger, 2012). The DCOMP algorithm works not only for VIIRS but for more sensors
with observations in VIS and NIR. So far, it has been applied to the geo-stationary satellites SEVIRI (Spinning Enhanced
Visible and InfraRed Imager), GOES R-series, MTSAT (Multifunctional Transport Satellites) and HIMAWARI and the polar-
orbiting satellites NOAA-AVHRR (Advanced Very High Resolution Radiometer) series and MODIS. The retrieval approach
is based on solving the radiative transfer equation for a single-layered, plane-parallel homogeneously distributed cloud. The
retrieval originates from earlier methods that also retrieve cloud optical depth and cloud effective radius from visible and
near-infrared wavelengths (King, 1987; Nakajima and King, 1990a, b).





For the co-registration of ROCINN CAL cloud optical thickness, the COT variable in the EDR CloudDCOMP can be directly used. For the co-registration of the cloud albedo, the following approximation conversion formula is used to bring the VIIRS cloud optical thickness ($\tau_c$) to an equivalent cloud albedo ($A_c$) (Loyola, 2013; Kokhanovsky and Mayer, 2003):

$$A_c = 1 - \frac{1}{1.072 + 0.75\tau_c\left(1 - f_g\right)} \tag{5}$$

with $f_g$ being the constant for water clouds equal to 0.85 and the other constant numbers derived from semi-empirical
formulas (Kokhanovsky and Mayer, 2003).

### 3.3.1 New scheme for the co-registration of OCRA cloud fraction from UV/VIS to NIR

OCRA uses the reflectances from the UV/VIS spectral region and the co-registration is therefore done from the UV/VIS source band to the NIR target band. We denote with index $j$ the row in the NIR grid and with index $i$ the row in UV/VIS grid.

The most common situation is that two UV/VIS source pixels contribute to the NIR target pixel as demonstrated in Fig. 3.
When those UV/VIS pixels from the imager have different cloud fraction values, the weight for the j$^\text{th}$ target pixel is calculated according to the following mathematical formulation:

$$\gamma[j] = \frac{M_c^{NIR}[j] - M_c^{UV}[i+1]}{M_c^{UV}[i] - M_c^{UV}[i+1]}, \tag{6}$$

with the cloud fraction $M_c$ derived from Equation 4. Then, the cloud fraction at the j$^\text{th}$ target pixel is computed as:

$$f_c^{NIR}[j] = \gamma[j]\, f_c^{UV}[i] + \left(1 - \gamma[j]\right) f_c^{UV}[i+1] \tag{7}$$

In case the neighboring UV/VIS pixels from the imager have equal cloud fraction values, the weight calculation is simplified as:

$$\gamma[j] = \frac{M_c^{NIR}[j]}{M_c^{UV}[i]}. \tag{8}$$

Special treatments need to be considered for cases with only partial overlap between source and target band. For example, in TROPOMI there is only partial overlap between the source and target band at the east part of the swath, as shown in Fig. 4.
Therefore, the last target pixel of every scanline has the contribution of a single source pixel as depicted in Fig. 5. The weight calculation for this pixel is done similarly to Equation 8, and the co-registered cloud fraction is then

$$f_c^{NIR}[j] = \gamma[j]\, f_c^{UV}[i]. \tag{9}$$





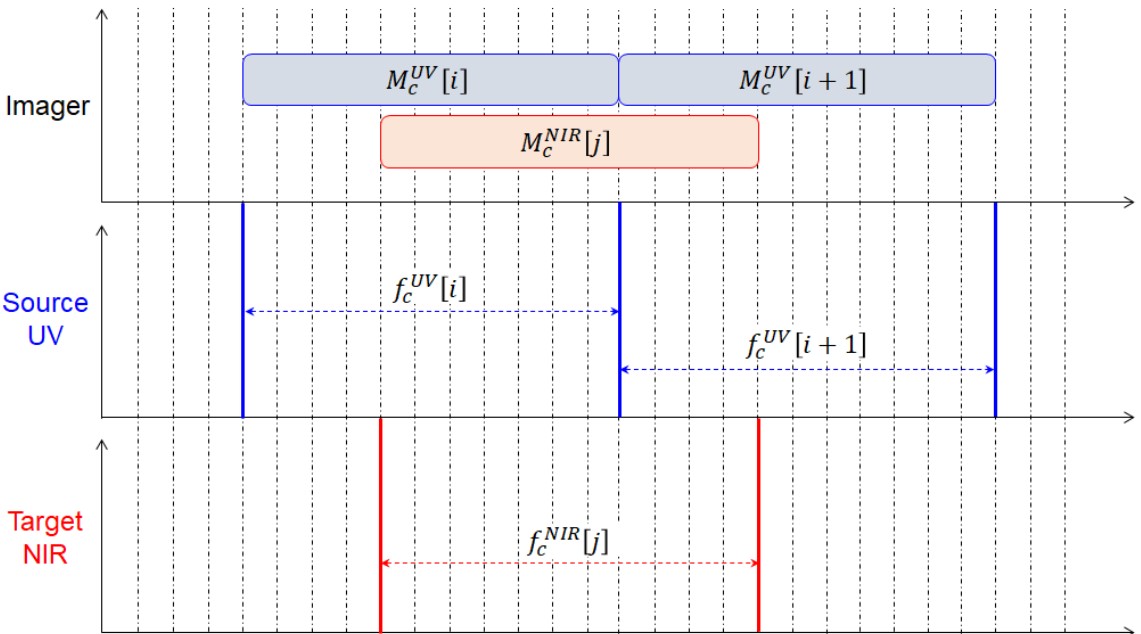

**Figure 3.** Co-registration of OCRA Cloud Fraction from UV/VIS to NIR: 2 source pixels contribute to the target pixel. The UV/VIS ground pixels are denoted with the blue boxes and the NIR ground pixels with the red boxes. The dashed vertical lines compose a grid for illustration purposes.

Other exceptions might refer to pixels affected by a binning change, where the binning factor changes from 2 to 1 (see Sect. 3). For TROPOMI, such a binning change occurs at the target pixel number 19 of every scanline. In this case, as presented in

Fig. 6, three UV/VIS pixels contribute to the target NIR pixel. Then, the calculation of two weighting factors is required; one between the $(i-1)^{th}$ and $i^{th}$ pixel and a second one between the $i^{th}$ and $(i+1)^{th}$ pixel.

$$\gamma_1[j] = \frac{M_c^{NIR}[j] - M_c^{UV}[i]}{M_c^{UV}[i-1] - M_c^{UV}[i]} \tag{10}$$

$$\gamma_2[j] = \frac{M_c^{NIR}[j] - M_c^{UV}[i+1]}{M_c^{UV}[i] - M_c^{UV}[i+1]} \tag{11}$$

with the final co-registered cloud fraction at the target NIR pixel expressed as following:

$$f_c^{NIR}[j] = \frac{1}{2} \left[ \gamma_1[j] f_c^{UV}[i-1] + (1 - \gamma_1[j] + \gamma_2[j]) f_c^{UV}[i] + (1 - \gamma_2[j]) f_c^{UV}[i+1] \right]. \tag{12}$$

### 3.3.2   New scheme for the co-registration of ROCINN cloud parameters from NIR to UV/VIS

ROCINN retrieves the additional cloud parameters in the Oxygen A-band of NIR (source band with index j). However, the trace gases are derived in a different band and need the cloud information in UV/VIS (target band with index i). When the



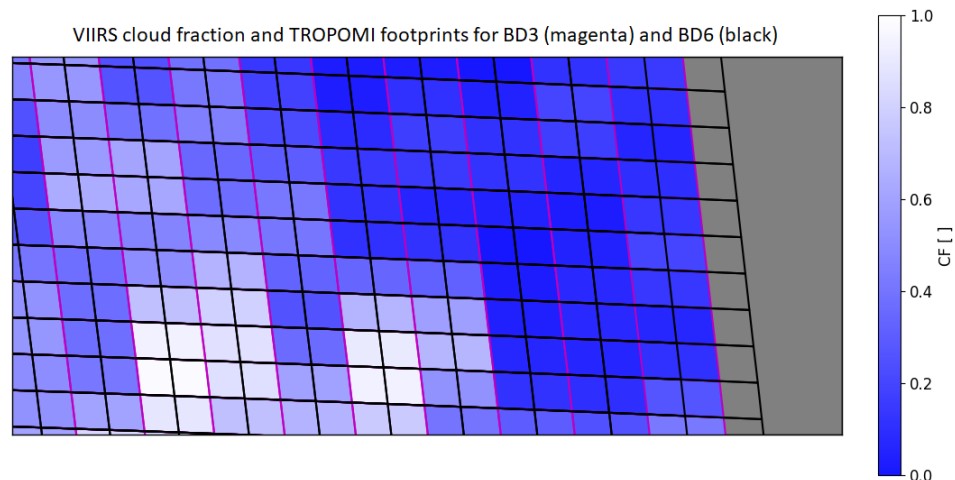

**Figure 4.** TROPOMI footprints for Bands 3 and 6 at the east edge of the orbit

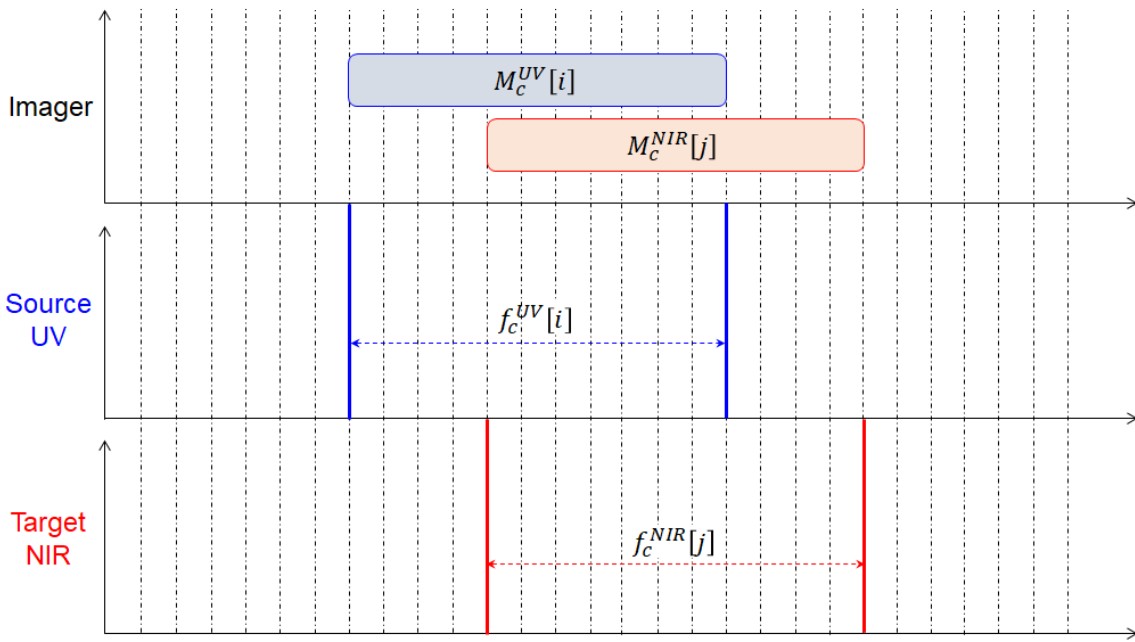

**Figure 5.** Co-registration of OCRA Cloud Fraction from UV/VIS to NIR: 1 source pixel contributes to the target pixel.

co-registration takes place from NIR to UV/VIS, the most frequent scenario is that two source pixels contribute to the target pixel as shown in Fig. 8. Following Equation 6, the weight for the $i^{th}$ UV/VIS target pixel is then calculated as:



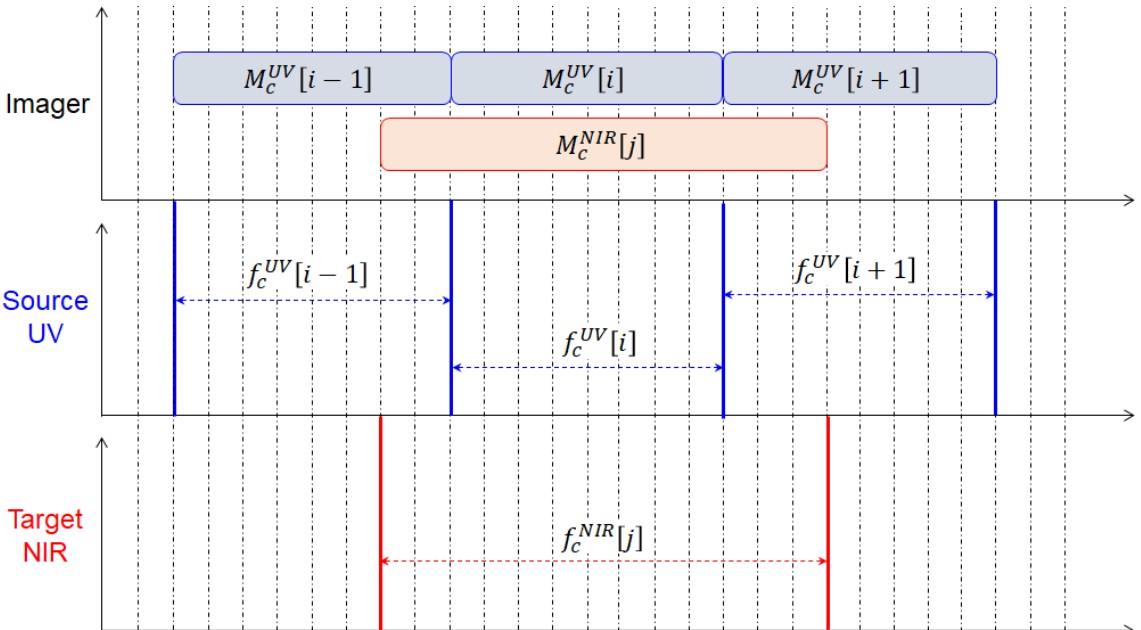

**Figure 6.** Co-registration of OCRA Cloud Fraction from UV/VIS to NIR: 3 source pixels contribute to the target pixel

$$\gamma[i] = \frac{H_c^{UV}[i] - H_c^{NIR}[j+1]}{H_c^{NIR}[j] - H_c^{NIR}[j+1]} \tag{13}$$

Then, the cloud-top height at the target pixel is expressed as:

$$Z_c^{UV}[i] = \gamma[i]\, Z_c^{NIR}[j] + (1 - \gamma[i])\, Z_c^{NIR}[j+1]. \tag{14}$$

The upper mathematical expressions are valid along the scanline. However, for every sensor there might be exceptions and
thus adaptations. For TROPOMI such exceptions occur at *(a)* UV/VIS target pixel 21, where the binning is changing, *(b)*
UV/VIS target pixel 1, where there is partial overlap with the NIR source pixel 0, and *(c)* UV/VIS target pixel 0, where there is
no overlap with any source pixel. The first two cases, illustrated in Fig. 9, need to follow the mathematical formulations from
Equations 8 and 9:

$$\gamma[i] = \frac{H_c^{UV}[i]}{H_c^{NIR}[j]}. \tag{15}$$

$$Z_c^{UV}[i] = \gamma[i]\, Z_c^{NIR}[j]. \tag{16}$$

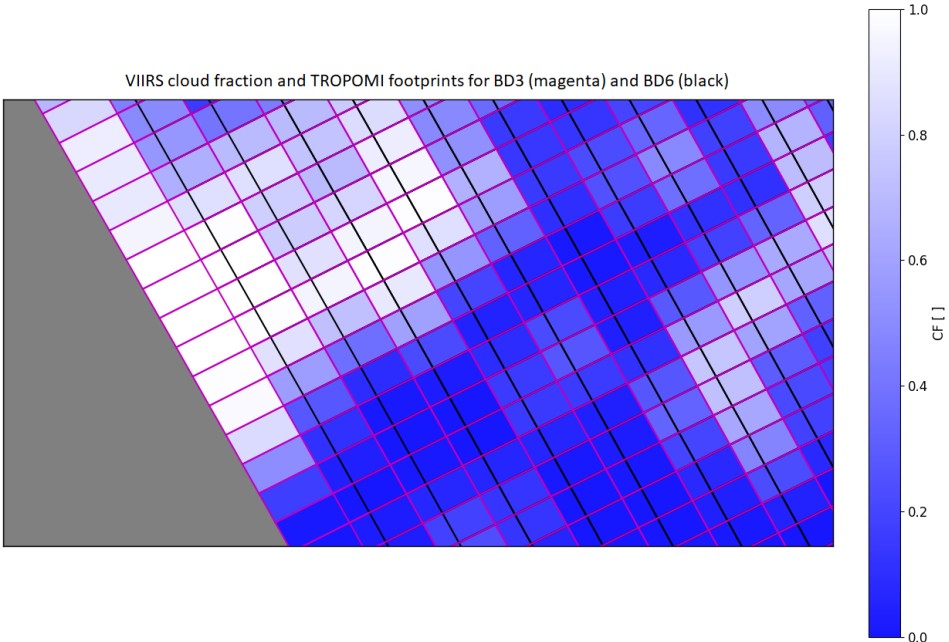

**Figure 7.** TROPOMI footprints for Bands 3 and 6 at the west edge of the orbit

The special case of the UV/VIS target pixel 0 has been treated independently since there is lack of overlap between the source and target bands. A graphical illustration of this scenario is shown in Fig. 10. The cloud information from VIIRS can be used for the reconstruction of the cloud parameters at the S5P target pixel 0. The basic principle is that VIIRS and TROPOMI cloud data are interconnected and therefore, each point from the VIIRS dataset can be mapped to the respective TROPOMI

point. The adjacent 15 pairs $\left( H_c^{UV}[i], Z_c^{UV}[i] \right), i \in [2, 17]$ are used to create the mapping function:

$$Z_c^{UV} = f_{Z_c}\left(H_c^{UV}\right) \approx \alpha H_c^{UV} + \beta \tag{17}$$

The mapping function for the cloud-top height $f_{Z_c}$ can be well approximated with a linear regression model. Therefore, first the mapping function is found for each scanline and secondly the value at the target pixel 0 is then estimated as $Z_c^{UV}[i=0] = f_{Z_c}\left(H_c^{UV}[i=0]\right)$. One example scene of the target pixel 0 is shown in Fig. 11.





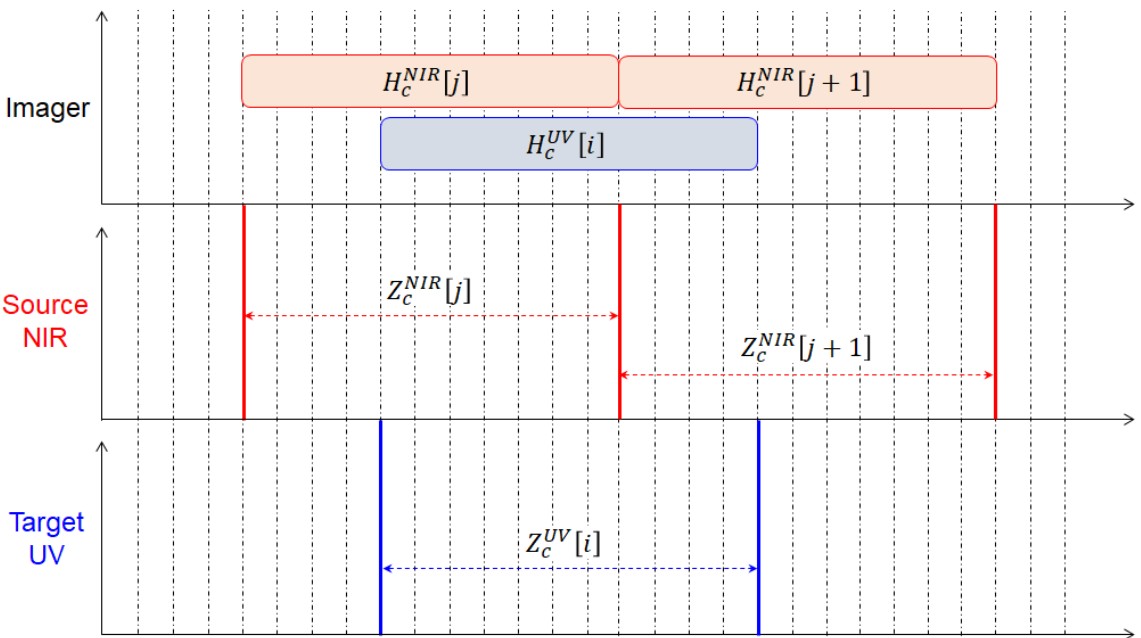

**Figure 8.** Co-registration of ROCINN Cloud Height from NIR to UV: 2 source pixels contribute to the target pixel

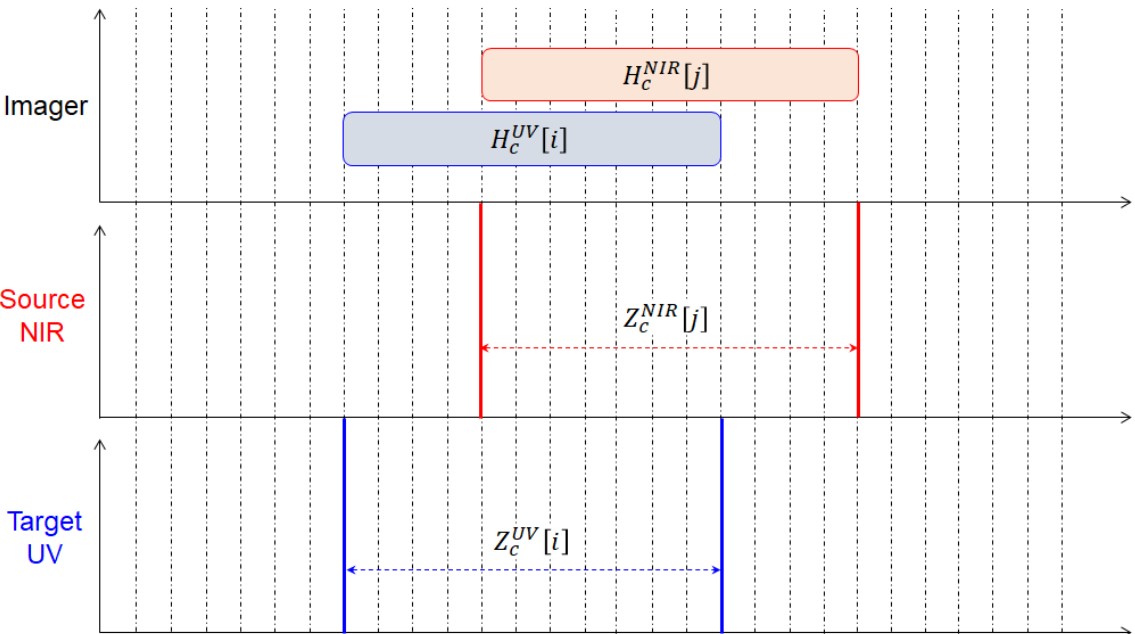

**Figure 9.** Co-registration of ROCINN Cloud Height from NIR to UV: 1 source pixel contributes to the target pixel





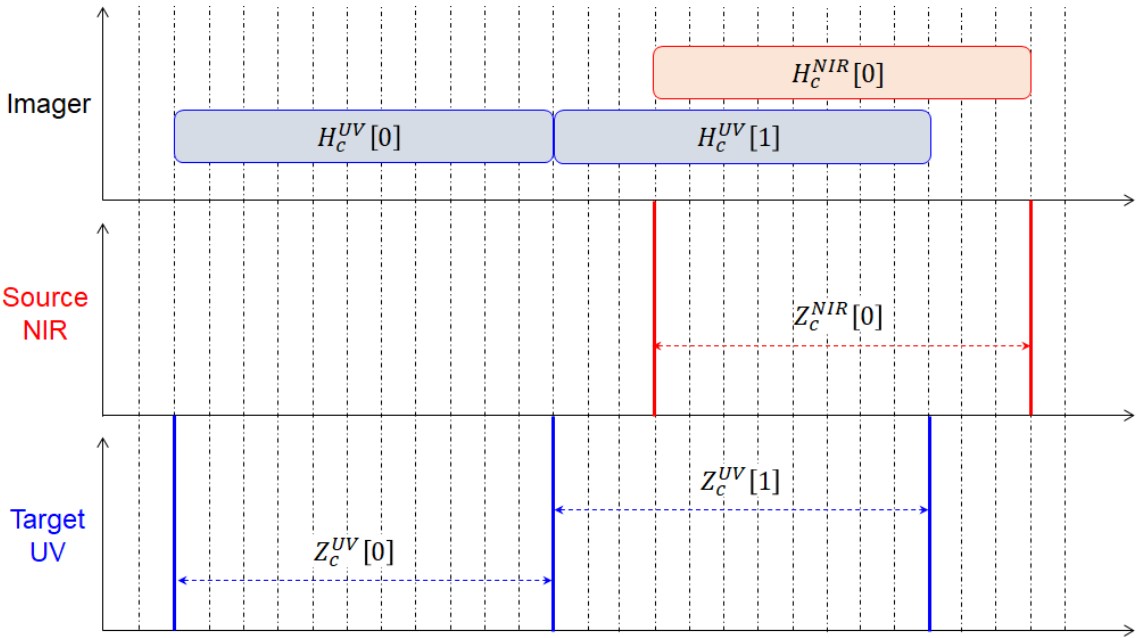

**Figure 10.** Co-registration of ROCINN Cloud Height from NIR to UV: A special case of the $0^{th}$ target pixel for TROPOMI

The co-registration of the other ROCINN parameters at the target pixel 0 is possible after finding the mapping function $f_{A_c}$ and $f_{\tau_c}$ for the cloud albedo and cloud optical thickness, respectively. Those functions are approximated with the use of one linear and one logarithmic model (Loyola et al., 2023).

## 4  Application to TROPOMI/S5P with collocated VIIRS/Suomi-NPP data

The new approach has been evaluated using several means of comparison and validation. The VIIRS product has been re-gridded to the TROPOMI footprints for the following six test days: 2018-09-09 (orbits 04691-04704), 2019-09-11 (orbits 09898-09911), 2020-09-11 (orbits 15091-15104), 2020-09-26 (orbits 15303-15316), 2021-04-11 (orbits 18098-18111) and 2021-09-11 (orbits 20269-20282).

### 4.1  Evaluation of the new approach

The new approach has been applied on top of the old scheme to ensure that the co-registration is performed with the mapping tables when there are no VIIRS data available. The new scheme is in principle not applicable on the following situations: *(a)* when TROPOMI or VIIRS pixels contain fill value, *(b)* when the neighboring VIIRS pixels contain zeros due to some numerical errors at the weight calculations, *(c)* when the weight calculation results to values outside the expected range. Another special case, where the new approach for the CF co-registration is not applicable, is when all three VIIRS BD3 pixels





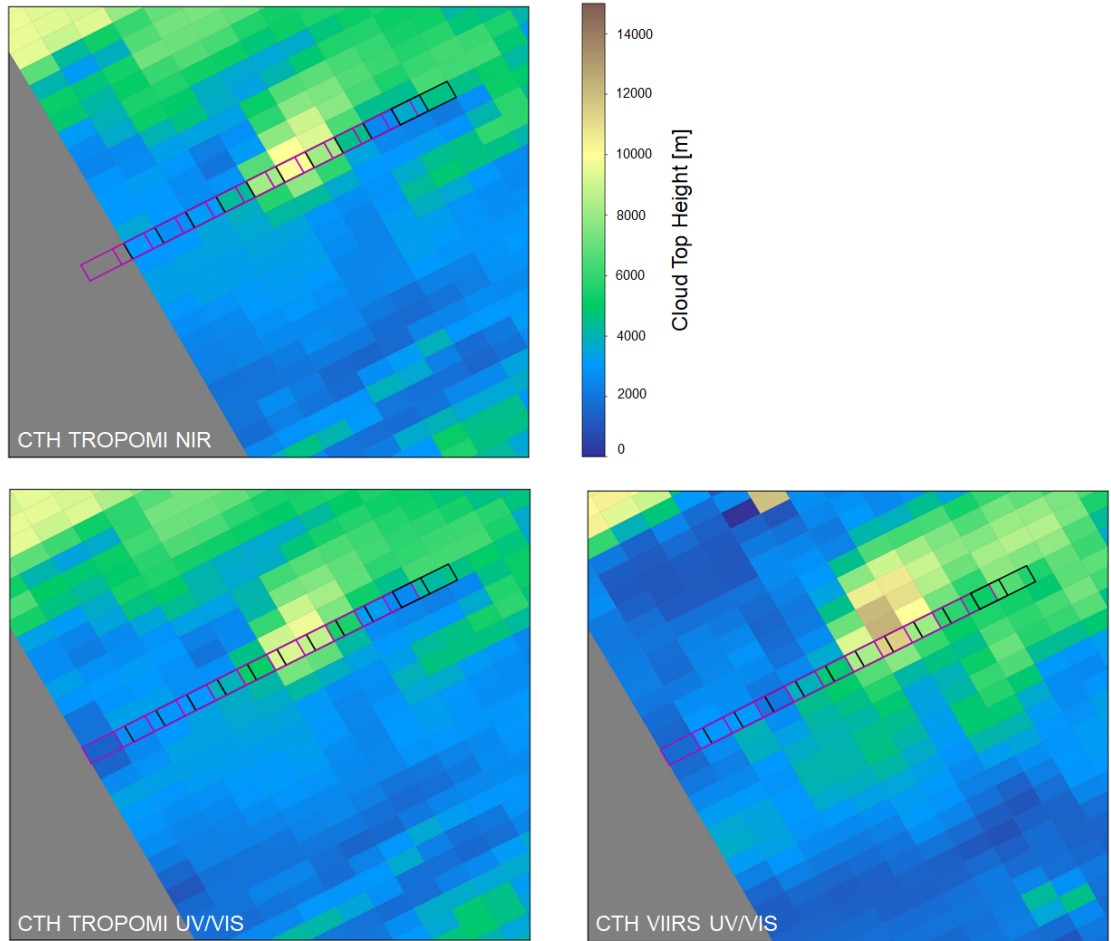

**Figure 11.** Co-registration of ROCINN Cloud Height from BD6 to BD3 using VIIRS first pixel: example scene. Pixel footprints in the source NIR band are indicated with the black frames and the UV/VIS footprints are highlighted with the magenta frames.

are equal $M_c^{UV}[j-1] = M_c^{UV}[j] = M_c^{UV}[j+1]$, while S5P BD3 pixels are different $f_c^{UV}[j-1] \neq f_c^{UV}[j] \neq f_c^{UV}[j+1]$.

The combination of both schemes ensures that the cloud product contains as many data as possible.

### 4.1.1   Overview of comparisons between the two schemes

An illustration of the global daily map in Fig. 12 shows how often the new scheme is used for the co-registration of the OCRA cloud fraction. The new co-registration is applied in average for slightly more than 70% of the total pixels, whereas the rest are co-registered with the fallback. The average frequency differs slightly from one day to the other but it can be considered rather

stable when there is VIIRS data availability. The co-registration from NIR to UV/VIS for the ROCINN parameters is performed with the new scheme with an average frequency of about 70% for the cloud-top height (and cloud height for CRB) and with



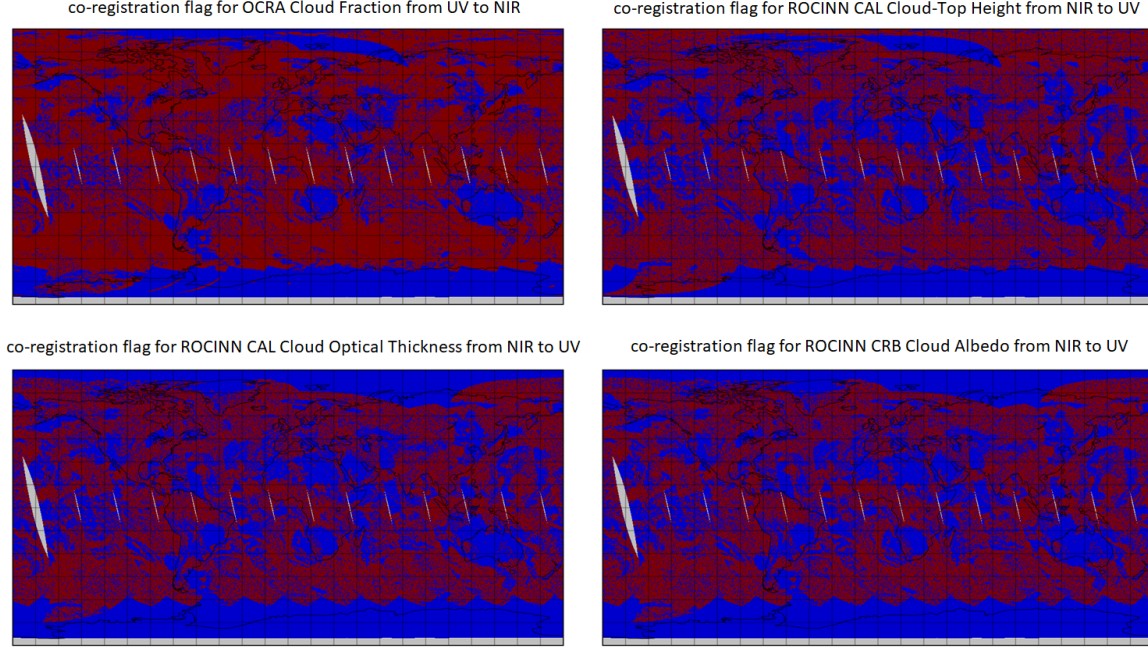

**Figure 12.** RB color global maps presenting the co-registration flag for the OCRA cloud fraction (upper left), ROCINN CAL cloud-top height (upper right), ROCINN CAL cloud optical thickness (lower left) and ROCINN CRB cloud albedo (lower right) : Red color shows the co-registration with the use of VIIRS data and Blue color shows the co-registration with the fallback. More than 70% of the OCRA cloud fraction pixels are co-registered with the new scheme. Approximately 70% of the ROCINN CAL cloud-top height pixels are co-registered with the new scheme. Only 55-60% of the ROCINN CAL cloud optical thickness / ROCINN CRB cloud albedo pixels are co-registered with the new scheme.

an average frequency of 55-60% for the cloud optical thickness (and cloud albedo for CRB). The co-registration flag for the cloud-top height covers similar regions with the cloud fraction but slightly less densely populated. From the co-registration flag of the optical parameter (i.e., cloud optical thickness for CAL and cloud albedo for CRB as shown in lower left and right panel

of Fig. 12 respectively), the new scheme seems to be successfully applied up to a certain latitude, forcing the pixels around the poles to be co-registered with the fallback.

In Figures 13, 14, 15, 16 and 17, the correlation between the old and new approach is illustrated for one of the days. High correlation coefficients are found for all cloud parameters. For the cloud fraction, below the identity line scatter is in principle not that extended in comparison to area above the identity line. Pixels of fully cloudy conditions (i.e., with cloud fraction 1) in

the old scheme have been differentiated in several cases; the cloud fraction obtains lower values with the new scheme. Several partially cloudy pixels, with an associated cloud fraction different than 0, have been characterized as cloud-free with the new co-registration scheme. The cloud height CRB and cloud-top height CAL are scattered symmetrically around the identity line. Some asymmetry is observed at the cloud optical thickness in Fig. 17 with the scatter below the identity line being much





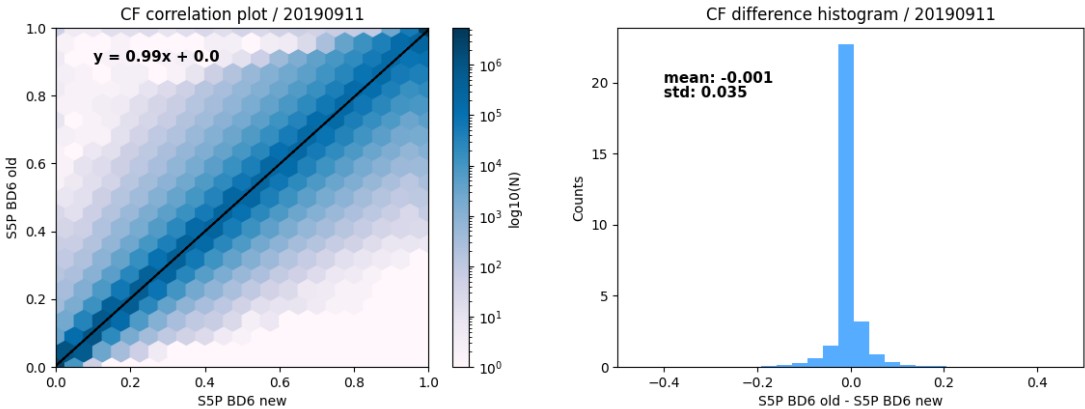

**Figure 13.** The co-registered cloud fraction for the new versus the old scheme: data analysis refers to 2019-09-11

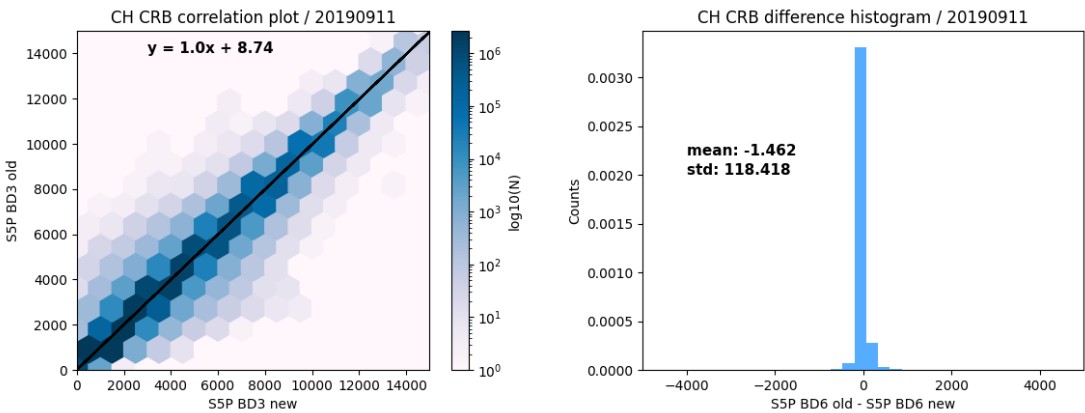

**Figure 14.** The co-registered ROCINN CRB cloud height for the new versus the old scheme: data analysis refers to 2019-09-11

higher than the scatter above the line. The correlation plots for the rest of the days can be found as supplementary material in

the Appendix A.

The absolute differences between the two co-registration schemes in a global scale are shown in Figures 18, 19, 20, 21 for the OCRA cloud fraction, ROCINN cloud-top height, cloud albedo and cloud optical thickness, respectively. It is expected that the differences are exactly zero when VIIRS data are not available because the co-registration is done with the fallback. Examples of VIIRS data unavailability (i.e., missing granules or entire orbits) are shown in green ellipsoids on the maps.

Based on these comparisons, the following conclusions can be drawn:

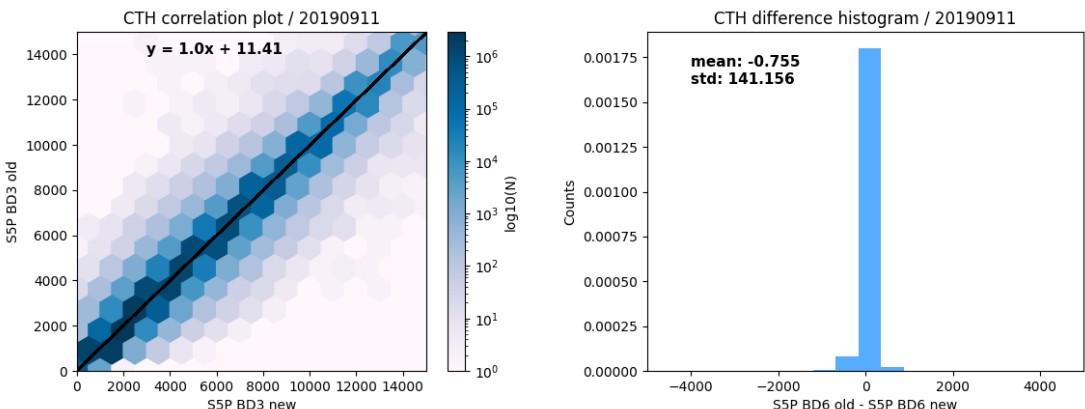

**Figure 15.** The co-registered ROCINN CAL cloud-top height for the new versus the old scheme: data analysis refers to 2019-09-11

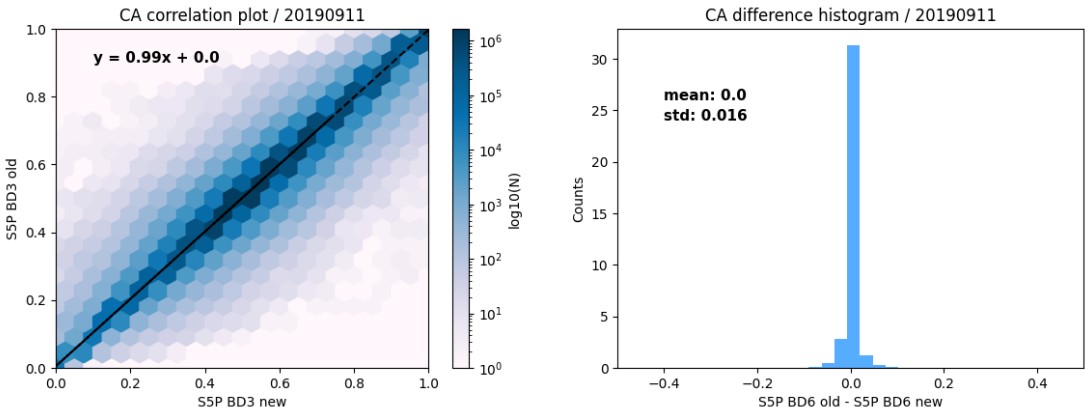

**Figure 16.** The co-registered ROCINN CRB cloud albedo for the new versus the old scheme: data analysis refers to 2019-09-11

- The differences are not systematically present in certain regions but rather spread everywhere.

- There is not a latitudinal dependence.

- Viewing geometry dependencies are not present.

The absolute differences of the co-registered NIR cloud fraction $\Delta f_c = f_c^{NIR_{old}} - f_c^{NIR_{new}}$ are grouped into the categories:

$\mathbf{A}_{f_c}$, $\mathbf{B}_{f_c}$, $\mathbf{C}_{f_c}$, $\mathbf{D}_{f_c}$, $\mathbf{E}_{f_c}$, $\mathbf{F}_{f_c}$, $\mathbf{G}_{f_c}$, $\mathbf{J}_{f_c}$, $\mathbf{K}_{f_c}$ of Table 3. The dependencies on the source UV/VIS cloud fraction are illustrated in the barplot of Fig. 22. The zero absolute differences of group $\mathbf{E}_{f_c}$ are in general present under fully cloudy conditions with a cloud fraction higher than 0.8. The negative differences of groups $\mathbf{A}_{f_c}$, $\mathbf{B}_{f_c}$, $\mathbf{C}_{f_c}$ and $\mathbf{D}_{f_c}$ are primarily present in clear sky





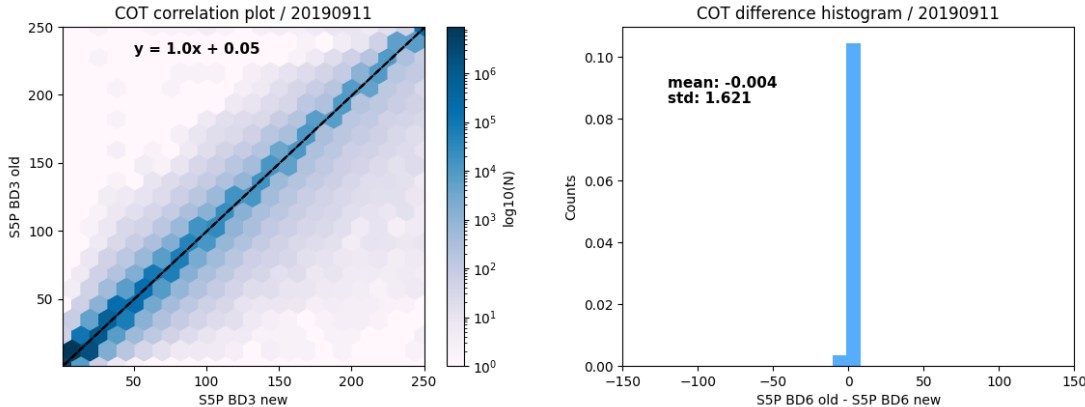

**Figure 17.** The co-registered ROCINN CAL cloud optical thickness for the new versus the old scheme: data analysis refers to 2019-09-11

**Table 3.** The absolute differences of the co-registered cloud parameters are split into 9 groups: column 2 refers to differences in NIR cloud fraction ($\Delta f_c$) [ ], column 3 refers to differences in UV/VIS cloud-top height ($\Delta Z_c$) [km], column 4 refers to differences in UV/VIS cloud optical thickness ($\Delta \tau_c$) [ ] and column 5 refers to differences in UV/VIS cloud albedo ($\Delta A_c$) [ ].

|  | $\Delta f_c = f_c^{NIR_{old}} - f_c^{NIR_{new}}$ | $\Delta Z_c = Z_c^{UV_{old}} - Z_c^{UV_{new}}$ | $\Delta \tau_c = \tau_c^{UV_{old}} - \tau_c^{UV_{new}}$ | $\Delta A_c = A_c^{UV_{old}} - A_c^{UV_{new}}$ |
|---|---|---|---|---|
| Group A | $\mathbf{A}_{f_c}$ [-1.0, -0.75) | $\mathbf{A}_{Z_c}$ [-10, -5) | $\mathbf{A}_{\tau_c}$ [-250, -50) | $\mathbf{A}_{A_c}$ [-1, -0.75) |
| Group B | $\mathbf{B}_{f_c}$ [-0.75, -0.5) | $\mathbf{B}_{Z_c}$ [-5, -2) | $\mathbf{B}_{\tau_c}$ [-50, -10) | $\mathbf{B}_{A_c}$ [-0.75, -0.50) |
| Group C | $\mathbf{C}_{f_c}$ [-0.5, -0.25) | $\mathbf{C}_{Z_c}$ [-2, -1) | $\mathbf{C}_{\tau_c}$ [-10, -5) | $\mathbf{C}_{A_c}$ [-0.5, -0.25) |
| Group D | $\mathbf{D}_{f_c}$ [-0.25, 0) | $\mathbf{D}_{Z_c}$ [-1, 0) | $\mathbf{D}_{\tau_c}$ [-5, 0) | $\mathbf{D}_{A_c}$ [-0.25, 0) |
| Group E | $\mathbf{E}_{f_c}$: zero differences | $\mathbf{E}_{Z_c}$: zero differences | $\mathbf{E}_{\tau_c}$: zero differences | $\mathbf{E}_{A_c}$: zero differences |
| Group F | $\mathbf{F}_{f_c}$ (0, 0.25] | $\mathbf{F}_{Z_c}$ (0, 1] | $\mathbf{F}_{\tau_c}$ (0, 5] | $\mathbf{F}_{A_c}$ (0, 0.25] |
| Group G | $\mathbf{G}_{f_c}$ (0.25, 0.5] | $\mathbf{G}_{Z_c}$ (1, 2] | $\mathbf{G}_{\tau_c}$ (5, 10] | $\mathbf{G}_{A_c}$ (0.25, 0.50] |
| Group J | $\mathbf{J}_{f_c}$ (0.5, 0.75] | $\mathbf{J}_{Z_c}$ (2, 5] | $\mathbf{J}_{\tau_c}$ (10, 50] | $\mathbf{J}_{A_c}$ (0.50, 0.75] |
| Group K | $\mathbf{K}_{f_c}$ (0.75, 1.0] | $\mathbf{K}_{Z_c}$ (5, 10] | $\mathbf{K}_{\tau_c}$ (50, 250] | $\mathbf{K}_{A_c}$ (0.75, 1] |

conditions and the respective positive differences of groups $\mathbf{F}_{f_c}$, $\mathbf{G}_{f_c}$, $\mathbf{J}_{f_c}$ and $\mathbf{K}_{f_c}$ are for the cloudy conditions with relative high cloud fractions. The latter conclusion in principle means that when the UV/VIS OCRA cloud fraction is high, the co-registration to NIR with the new scheme results to a slightly lower cloud fraction compared to the old scheme. Exactly the



**Figure 18.** Daily global maps with comparisons between the two co-registration schemes: the co-registered OCRA cloud fraction in the NIR is shown for the six available days. When VIIRS granules or orbits are missing (depicted in the green frames), the parameters are co-registered with the fallback old scheme based on the mapping tables.

opposite happens when the UV/VIS OCRA cloud fraction takes low values: the co-registration with the new scheme results to higher NIR cloud fraction than with the old scheme.

For the several cloudy conditions (i.e., from almost clear sky to fully cloudy scenes), the absolute differences of the co-registered UV/VIS cloud-top height $\Delta Z_c = Z_c^{UV_{old}} - Z_c^{UV_{new}}$ are presented in the respective groups: $\mathbf{A}_{Z_c}$, $\mathbf{B}_{Z_c}$, $\mathbf{C}_{Z_c}$, $\mathbf{D}_{Z_c}$,
$\mathbf{E}_{Z_c}$, $\mathbf{F}_{Z_c}$, $\mathbf{G}_{Z_c}$, $\mathbf{J}_{Z_c}$, $\mathbf{K}_{Z_c}$, as introduced in Table 3. With only six days it is difficult to draw final conclusions but it appears that for almost fully cloudy conditions with cloud fraction larger than 0.8, the co-registered cloud-top height with the new and the old scheme agree in a high degree as the group $\mathbf{E}_{Z_c}$ is dominated by the fully cloudy scenes for the 4 days. Exceptions appear

**Figure 19.** Daily global maps with comparisons between the two co-registration schemes: the co-registered ROCINN CAL cloud-top height in the UV/VIS is shown for the six available days. The comparison for the ROCINN CRB cloud height is excluded as it is considered redundant information and does not lead to further conclusions.

for the day 2019-09-11 where only 20% of the cases in group $\mathbf{E}_{Z_c}$ is associated to high OCRA cloud fractions and for day 2020-09-11 where about 45% of the cases refer to scenes with such high cloud fractions. For the relative clear scenes with low cloud fractions below 0.2, the cloud-top height differences cover with a mean percentage of 20% equally any group other than the $\mathbf{E}_{Z_c}$. The same conclusion could be valid for the partially cloudy scenes based on the bar-plot of Fig. 23. Similar bar-plots have been produced for the cloud optical thickness in Fig. 24 with the cloud optical thickness differences $\Delta\tau_c = \tau_c^{UV_{old}} - \tau_c^{UV_{new}}$ grouped into $\mathbf{A}_{\tau_c}, \mathbf{B}_{\tau_c}, \mathbf{C}_{\tau_c}, \mathbf{D}_{\tau_c}, \mathbf{E}_{\tau_c}, \mathbf{F}_{\tau_c}, \mathbf{G}_{\tau_c}, \mathbf{J}_{\tau_c}, \mathbf{K}_{\tau_c}$ (see Table 3). It is quite clear that under nearly fully cloudy conditions, the new co-registered cloud optical thickness is smaller than the old one whereas for the relative clear sky scenes, the new co-



**Figure 20.** Daily global maps with comparisons between the two co-registration schemes: the co-registered ROCINN CAL cloud optical thickness in the UV/VIS is shown for the six available days.

registered cloud optical thickness in UV is larger than the old one. In particular, the scenes with very low cloud fractions below 0.2 are usually associated with a co-registered COT difference of the groups $\mathbf{C}_{\tau_c}$ and $\mathbf{D}_{\tau_c}$, which practically means that we expect a small increase of the cloud optical thickness for the low OCRA cloud fraction scenes (e.g., in the presence of optically thin clouds). The latter conclusion is valid for the co-registered cloud albedo difference $\Delta A_c = A_c^{UV_{old}} - A_c^{UV_{new}}$ as can be seen from Fig. 25, where the groups are $\mathbf{A}_{A_c}$, $\mathbf{B}_{A_c}$, $\mathbf{C}_{A_c}$, $\mathbf{D}_{A_c}$, $\mathbf{E}_{A_c}$, $\mathbf{F}_{A_c}$, $\mathbf{G}_{A_c}$, $\mathbf{J}_{A_c}$, $\mathbf{K}_{A_c}$ based on the categorization of Table 3. At the relative cloud free scenes with OCRA cloud fraction lower than 0.2, we expect increase at the co-registered cloud



**Figure 21.** Daily global maps with comparisons between the two co-registration schemes: the co-registered ROCINN CRB cloud albedo in the UV/VIS is shown for the six available days.

albedo with the new scheme. It's worth referring to group $\mathbf{A}_{A_c}$ of days 2020-09-11, 2020-09-26 and 2021-04-11 because there are no fully cloudy scenes with such high negative cloud albedo differences between the two co-registration schemes.

### 4.1.2 The first UV pixel as a special case

The first TROPOMI UV detector pixels were lacking of cloud height and cloud optical thickness properties up to now due to the missing TROPOMI NIR overlap detector pixels. The use of VIIRS data made possible to re-construct the UV/VIS cloud height information for these pixels through the mapping linear function of Equation 17. The benefit of making use of the VIIRS cloud information to fill in the first UV/VIS detector row is two-fold: (a) the apparent advantage of closing the gaps between



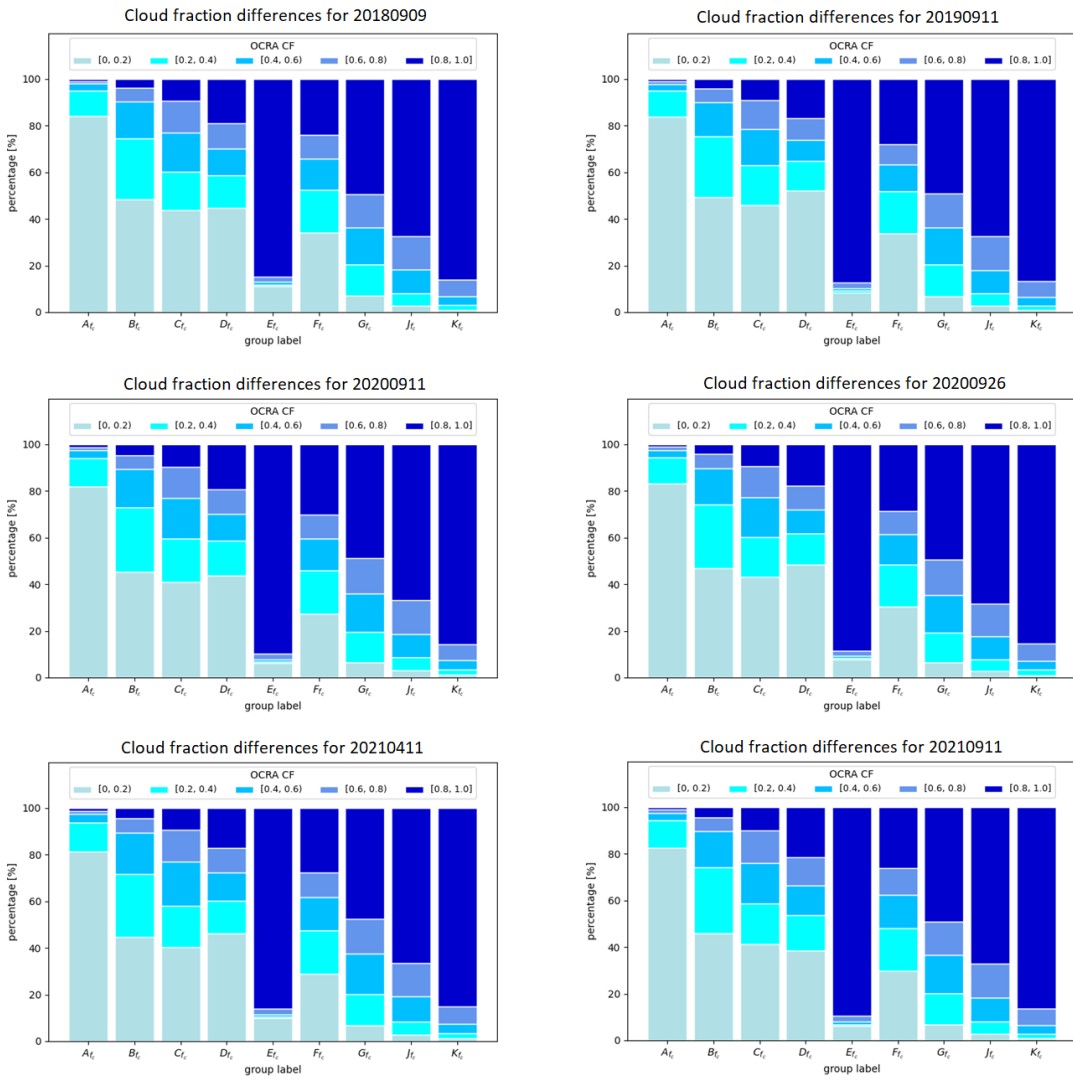

**Figure 22.** Bar-plot showing the difference in the co-registered NIR cloud fraction. The color scale applies to the UV/VIS cloud fraction. The groups are $\mathbf{A}_{f_c}$ [-1.0, -0.75), $\mathbf{B}_{f_c}$ [-0.75, -0.5), $\mathbf{C}_{f_c}$ [-0.5, -0.25), $\mathbf{D}_{f_c}$ [-0.25, 0), $\mathbf{E}_{f_c}$: zero differences, $\mathbf{F}_{f_c}$ (0, 0.25], $\mathbf{G}_{f_c}$ (0.25, 0.5], $\mathbf{J}_{f_c}$ (0.5, 0.75], $\mathbf{K}_{f_c}$ (0.75, 1.0].

two adjacent orbits and (b) the actual retrieval of tropospheric and stratospheric trace gases which require the knowledge of cloud parameters. The data gaps between two adjacent orbits are expected around the Equator as shown in Fig. 26. With the new co-registration scheme, those gaps are decreased by approximately 15 km after the addition of meaningful cloud data in the first row (see Fig. 27). The approach seems to work smoothly for all cloud types since the cloud heights of the first row are well harmonized with the neighboring rows. Similar conclusions can be drawn for all the cloud parameters.

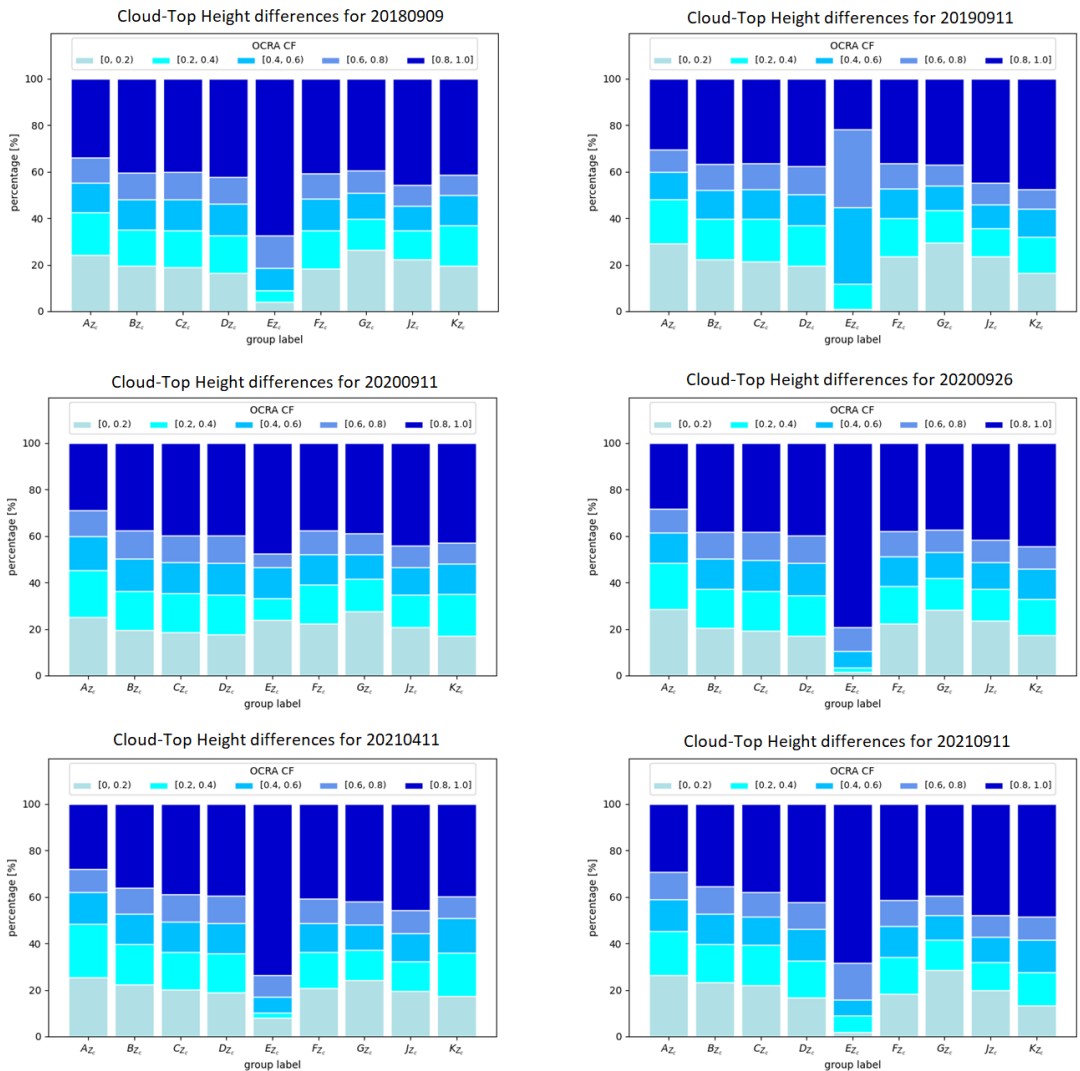

**Figure 23.** Bar-plot showing the difference in the co-registered UV cloud-top height. The groups are $\mathbf{A}_{Z_c}$ [-10, -5), $\mathbf{B}_{Z_c}$ [-5, -2), $\mathbf{C}_{Z_c}$ [-2, -1), $\mathbf{D}_{Z_c}$ [-1, 0), $\mathbf{E}_{Z_c}$: zero differences, $\mathbf{F}_{Z_c}$ (0, 1], $\mathbf{G}_{Z_c}$ (1, 2], $\mathbf{J}_{Z_c}$ (2, 5], $\mathbf{K}_{Z_c}$ (5, 10] in km.

The air mass factor (AMF) calculation was evaluated for the total vertical column densities (VCDs) of formaldehyde (HCHO), ozone (O3), and sulfur dioxide (SO$_2$) in the first UV/VIS row. The plot in Fig. 28 depicts the total vertical column of ozone, where in most cases we see an additional column of O3 data and the data agree very well with the neighboring column. This smooth transition is found for all cloud fractions, as shown in Fig. 29. In the selected area, there are cloudy and cloud free pixels, as well as partly cloudy pixels. Even in the partly cloudy pixels, the O3 columns agree well with the neighboring ones. For HCHO and SO$_2$ retrievals, the VCDs look smooth and reasonable for this additional row. For SO$_2$, the detection algorithm could even identify elevated VCDs at the first row and flag them with volcanic origin. In Fig. 30, we can





**Figure 24.** Bar-plot showing the difference in the co-registered UV cloud optical thickness. The groups are $\mathbf{A}_{\tau_c}$ [-250, -50), $\mathbf{B}_{\tau_c}$ [-50, -10), $\mathbf{C}_{\tau_c}$ [-10, -5), $\mathbf{D}_{\tau_c}$ [-5, 0), $\mathbf{E}_{\tau_c}$: zero differences, $\mathbf{F}_{\tau_c}$ (0, 5], $\mathbf{G}_{\tau_c}$ (5, 10], $\mathbf{J}_{\tau_c}$ (10, 50], $\mathbf{K}_{\tau_c}$ (50, 250].

see the $SO_2$ column densities after the Sierra Negra volcanic eruption. In the selected area, the pixels from the additional first row highlighted with a red frame have been automatically flagged as "volcanic".

### 4.2 Further evaluation in the across-track flight direction

An extensive investigation of the co-registration scheme impact on the cloud fraction in the across-flight direction has been performed. As expected, the major improvements have been identified at heterogeneous scenes in the vicinity of local minima



**Figure 25.** Bar-plot showing the difference in the co-registered UV cloud albedo. The groups are $\mathbf{A}_{A_c}$ [-1, -0.75), $\mathbf{B}_{A_c}$ [-0.75, -0.50), $\mathbf{C}_{A_c}$ [-0.5, -0.25), $\mathbf{D}_{A_c}$ [-0.25, 0), $\mathbf{E}_{A_c}$: zero differences, $\mathbf{F}_{A_c}$ (0, 0.25], $\mathbf{G}_{A_c}$ (0.25, 0.50], $\mathbf{J}_{A_c}$ (0.50, 0.75], $\mathbf{K}_{A_c}$ (0.75, 1].

and maxima. Usually, the co-registered value is closer to the one retrieved at the original band when using the new scheme. So far, we have seen that the co-registration process with the static mapping tables tends to smooths out structure which appears initially at the original band. When the co-registration is done based on the VIIRS data, the cloud structure is maintained simply because it is captured by the VIIRS-based weighting factors. The co-registration impact can be larger in inhomogeneous scenes with relatively small cloud fractions; a small fluctuation of the cloud fraction at a low cloud fraction results in a large fractional

difference. An example of a single scanline at longitude range [-34.7, -34.1] is presented in Fig. 31. The improvement with the new scheme is shown in points A-B-C around longitude -34.5 degree. The OCRA cloud fraction at the original BD3



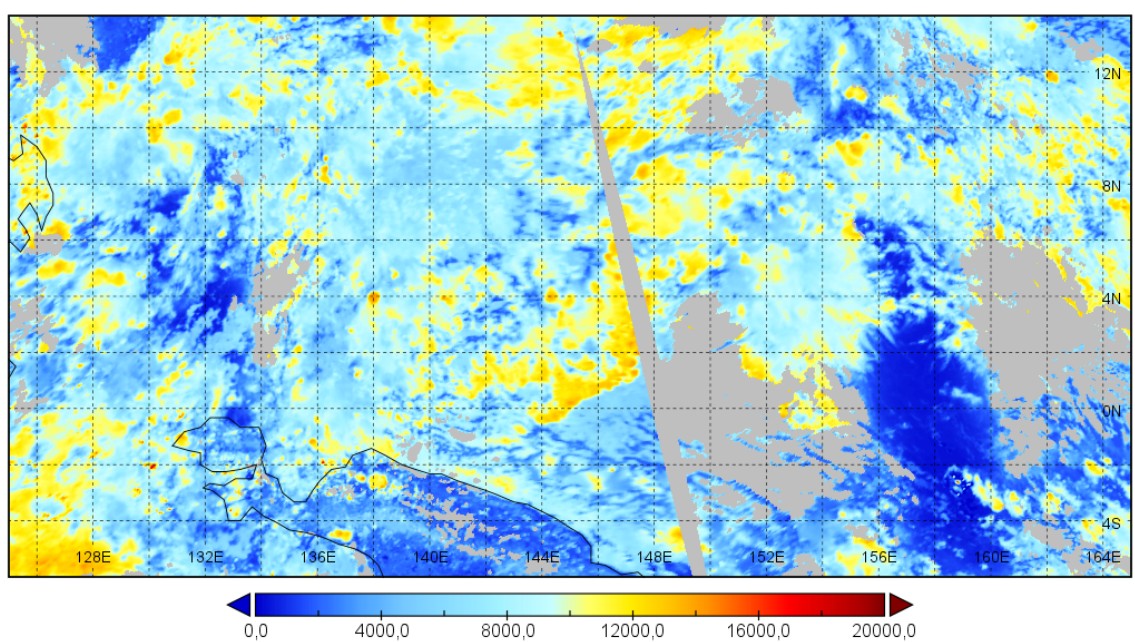

**Figure 26.** Two adjacent orbits displaying the co-registered cloud-top height at the UV/VIS grid using the static LUTs. There are small gaps between the adjacent orbits around the Equator.

has the same value at BD6 after the co-registration in point A. In other words, both co-registration schemes agree with the value obtained at the original band. However, the drop of the cloud fraction at point B demonstrates the importance on the co-registration scheme selection. The co-registered value obtained with the new scheme at point B is closer to the original one of BD3 than with the old scheme. In Table 4, an absolute difference of 0.04 is found between the two schemes, which at

first does not seem significant. However, due to the low original cloud fraction of 0.13, a fractional difference of 30% seems to be introduced just by using a different co-registration. Another important aspect at pixel B is that the cloud fraction gets below 0.2 with the new scheme. A cloud fraction of 0.2 is usually considered the cut-off threshold of clear sky versus partially cloudy to obtain clear sky series of tropospheric trace-gas concentrations (Liu et al., 2021). The cloud fraction with the old

co-registration was 0.21, which would mean that trace-gas retrieval for pixel B will be triggered with the new co-registration scheme. Similar improvement is found at pixels D-E-F between longitude range [-34.3,-34.2]. At pixel D, both co-registered cloud fractions are equal to the original one. But at pixel E, the new co-registered cloud fraction is 16% lower than the original value. At this pixel E, when co-registration is done with the old scheme the fractional difference is 32%. Even more interesting is the situation at pixel F where the original value is extremely low at 0.01. There, the selection of the co-registration scheme

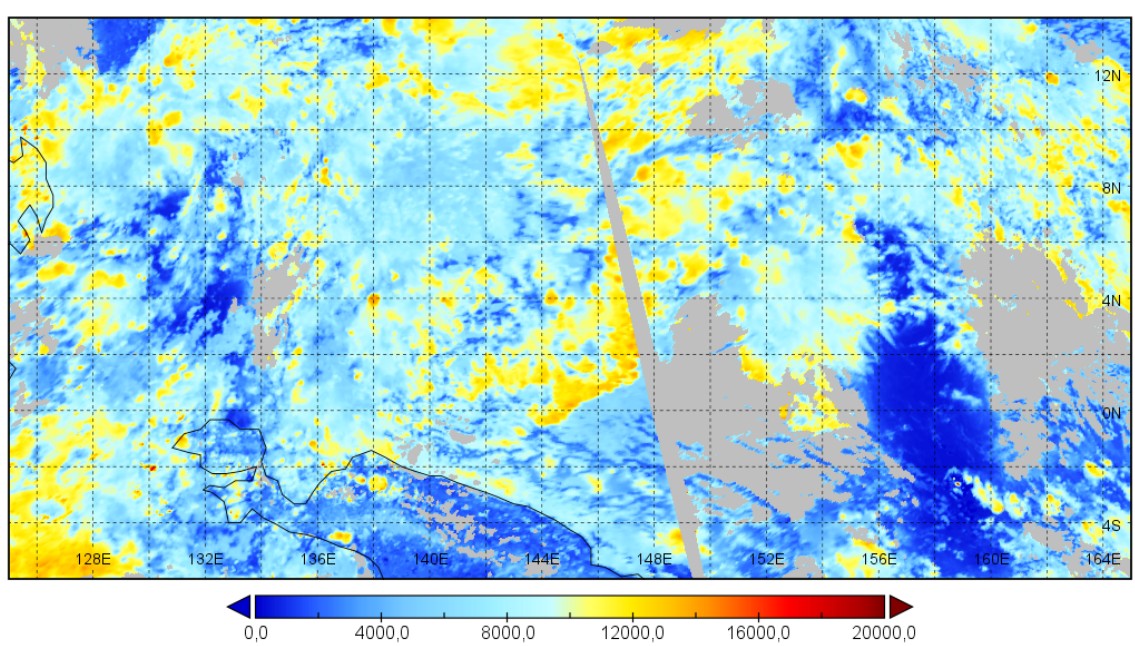

**Figure 27.** Two adjacent orbits displaying the co-registered cloud-top height at the UV/VIS grid using VIIRS data. The gaps between the adjacent orbits are decreased with the addition of the first row cloud data.

will determine the activation of ROCINN algorithm. The new co-registered value is still below 0.05, which is the threshold currently used to continue with the ROCINN retrievals of the remaining cloud parameters. Therefore, the use of VIIRS in the cloud co-registration process can act as a tool to remove existing cloud outliers.

In this section, we summarize the cases where the new scheme does not improve the cloud product. In Fig. 32, the blocks B and D highlighted with red color appear problematic, because the new co-registered cloud fraction seems to be shifted towards the East direction. The reason for this effect is that VIIRS cloud fraction at both bands 3 and 6 are equal to 1, and thus the weight calculation does not work properly. As a consequence, we use the fallback co-registration scheme at the fully cloudy VIIRS pixels. For the blocks A and C, where the VIIRS cloud fractions are different to 1, such shift to the East is no longer visible.

A second interesting scene with scattered clouds close to the Brazilian Coast line is presented in Fig. 33. At longitude around -36.10, the cloud fraction obtained in BD3 was equal to 0.96. This is a scene over ocean with the effect of sun-glint. An enhanced TROPOMI cloud fraction is generally expected under sun-glint geometry, and this is a possible reason that OCRA cloud fraction is found larger than VIIRS; VIIRS gives a 0.72 cloud fraction in both bands. The old co-registration scheme



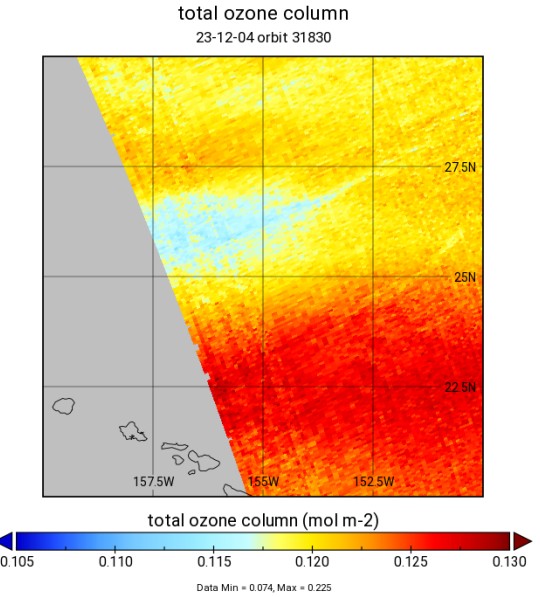

**Figure 28.** The total ozone column for the scene of Fig. 29. The total ozone column for the first row looks smooth w.r.t. the adjacent rows.

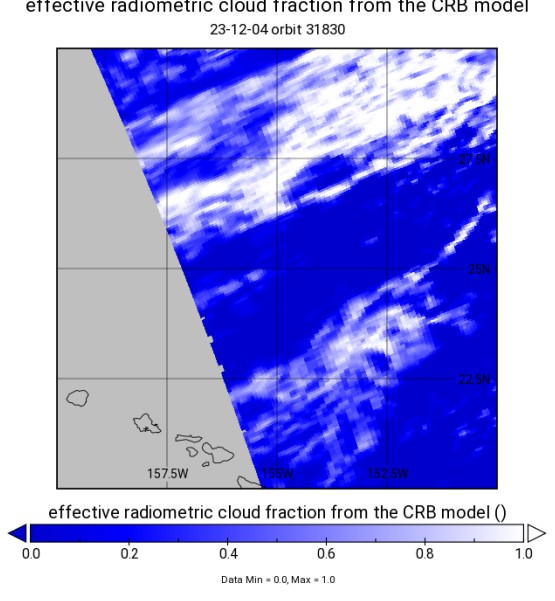

**Figure 29.** The radiometric cloud fraction from the ROCINN CRB model is used as an input parameter for the cloud correction in the ozone OFFL algorithm.



**Figure 30.** The SO$_2$ column density for a scene of the Sierra Negra volcanic eruption. Pixels highlighted with the red frames in the first detector row are detected with volcanic origin. The detection algorithm for volcanic SO$_2$ seems to work well for the additional first row as some pixels are flagged as volcanic.

moves the cloud fraction closer to VIIRS. Nevertheless, the new co-registration seems to reflect better the original BD3 cloud fraction of 0.96 to a 0.94 BD6 cloud fraction.

Investigation of the co-registration scheme impact on the cloud-top height in the across-flight direction has been done too. In general, similarly to the cloud fraction, the largest impact is shown at the inhomogeneous scenes at the local maxima and minima. At first, the new co-registered cloud-top height is closer to the one obtained with ROCINN CAL at the original BD6 band. Two examples are shown in Fig. 34 at points A and B, with the new co-registered cloud-top height being 300 m higher than the old co-registered value and in both cases closer to the original values. The cloud-top heights at the original BD6 were 375    7600 m and 9400 m at points A and B respectively. After the co-registration at point A, the CTH at BD3 was 6900 m with the

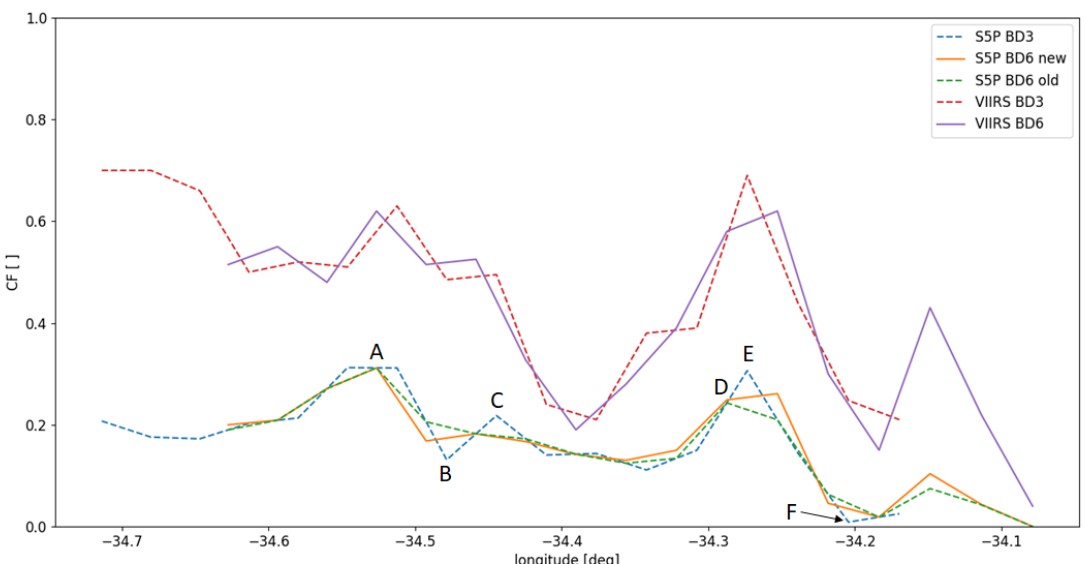

**Figure 31.** Inhomogeneous scene with low TROPOMI cloud fraction: the new co-registration scheme has a considerable positive effect at local minimum (e.g, points B and F) or maximum (e.g., point E). The data refer to 2020-09-11, orbit 15099, scanline 1820, pixels 268-285.

**Table 4.** Cloud fraction values at the original BD3 and the co-registered BD6 with both new and old techniques. The longitudes at the first column refer to points A-B-C-D-E-F of Fig. 31.

| Longitude (pixel) | ORIGINAL BD3 | CO-REGISTERED NEW BD6 | CO-REGISTERED OLD BD6 |
|---|---|---|---|
| -34.53 (A) | 0.31 | 0.31 | 0.31 |
| -34.48 (B) | 0.13 | 0.17 | 0.21 |
| -34.44 (C) | 0.22 | 0.18 | 0.18 |
| -34.29 (D) | 0.24 | 0.24 | 0.24 |
| -34.25 (E) | 0.31 | 0.26 | 0.21 |
| -34.21 (F) | 0.01 | 0.04 | 0.06 |

new scheme and 6600 m with the old scheme. At point B, the co-registered CTH at BD3 was 9300 m with the new scheme versus 9000 m with the old. A big advantage of using the VIIRS re-gridded data is that cloud structure from those data is introduced back to the TROPOMI CTHs.





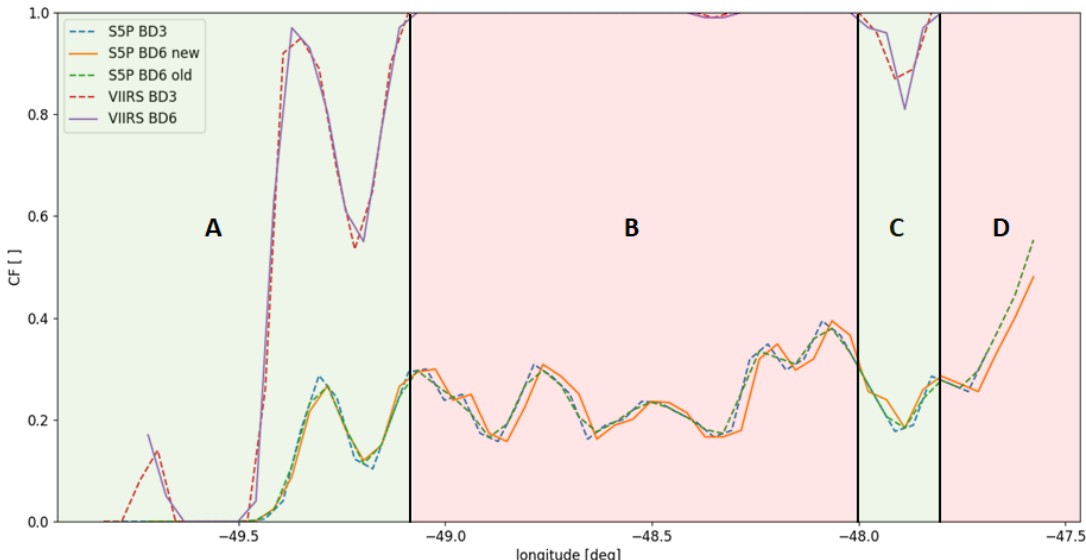

**Figure 32.** Inhomogeneous scene with variant cloud fraction. The green highlighted blocks A and C suggest that the new co-registration can be applicable but the red blocks B and D demonstrate examples that the fallback is preferred. The data refer to 2020-09-11, orbit 15099, scanline 2820, pixels 200-250.

### 4.3 Evaluation in the along-track flight direction: comparisons against CALIPSO overpasses

The evaluation of the new co-registration scheme in the along-track direction for the cloud-top height parameter was done using the independent instrument CALIOP (Cloud-Aerosol Lidar with Orthogonal Polarization) which is part of the CALIPSO (Cloud-Aerosol Lidar and Infrared Pathfinder Satellite Observations) payload. CALIPSO satellite was launched in April 2006 in formation with the CloudSat satellite as part of the A-Train constellation of satellites (Winker et al., 2003, 2004, 2007). For 12 years, it maintained a sun-synchronous orbit with an altitude of 685 km and inclination of 98.2° crossing the equator

each day at around 1:30 pm solar time. After September 2018, it was moved to a lower orbit together with CloudSat, part of the C-train approximately 688 km above the Earth's surface (Atkinson, 2018). CALIOP is a two-wavelength (i.e., operating at 532 nm and 1064 nm) polarization-sensitive lidar that provides high-resolution vertical profiles of aerosols and clouds. CALIOP can identify cloud and aerosol layers down to the level in which the lidar signal is totally attenuated. Frequently, the atmosphere contains multi-layer clouds limiting the lidar capabilities and making the cloud retrievals in such conditions more

challenging (Liu et al., 2020). Even though CALIOP can provide the cloud information with the fine spatial resolution of 1 km, in this study the spatial resolution of 5 km in the Level 2 cloud layer information (Version 4) was used.

CALIPSO overpasses have been collocated to the TROPOMI orbits. The TROPOMI/CALIPSO collocation method is described in the Appendix B. An example comparison at mid-latitudes of the north hemisphere over the Pacific ocean is presented in Fig. 35. This is quite representative for the comparison between the three instruments. Good agreement between all is seen





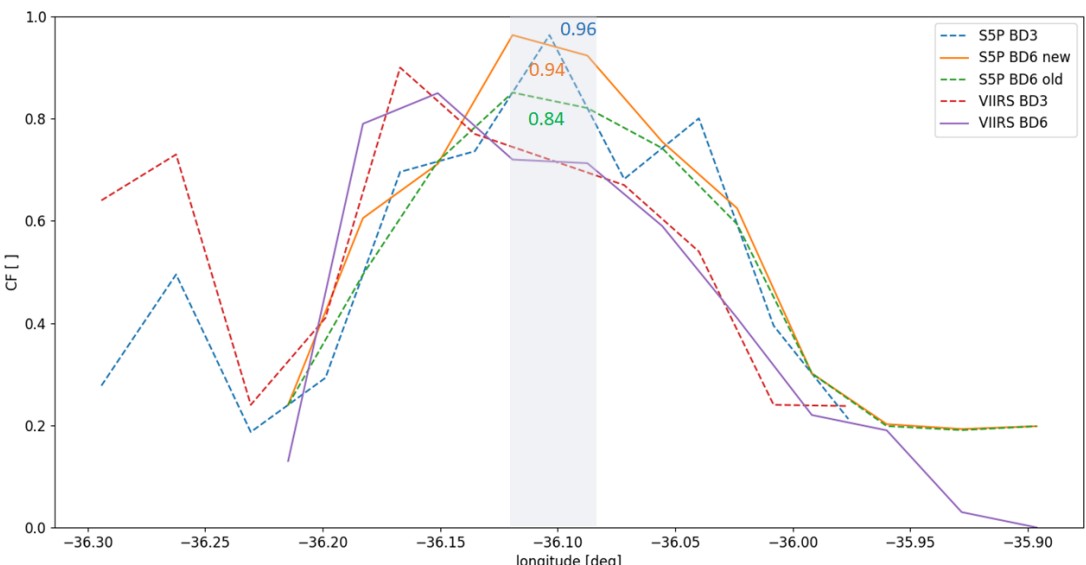

**Figure 33.** Inhomogeneous scene with high TROPOMI cloud fraction but with small horizontal extend. This could be considered a "single-cloud" scene where the new co-registration is preferable at local maximum because S5P BD6 new cloud fraction agrees better with S5P BD3 cloud fraction. The data refer to 2020-09-11, orbit 15099, scanline 1820, pixels 219-230.

for the low maritime clouds with a cloud-top height lower than 2 km. For medium and high clouds, the agreement of TROPOMI with CALIPSO depends on the phase of the detected clouds. The ice clouds are not well represented in the forward model of TROPOMI and larger biases could originate from the mis-treatment of clouds with a liquid water scattering model. Moreover, TROPOMI does not have any special treatment of multi-layer clouds and this is another reason for differences to be expected. Nevertheless, the focus of this study is on the improvements that could arise from the new co-registration scheme alone. Two

cases are presented in the following Figs 36 and 37. In general, the differences due to the co-registration (comparison between green and red lines) are small. The largest improvement is seen when the cloud structure, introduced by the use of VIIRS data, results in a better agreement with CALIPSO. In Fig. 36, the small peak (highlighted by the circle A) appearing in the CALIPSO data is seen in the TROPOMI data only when the new co-registration is used. The red line is flat around 16 deg latitude North, meaning that the old co-registration smooths this cloud structure and the impact on the CTH is an absolute difference of about

2 km compared to the new scheme. Similarly, in another example shown in Fig. 37, there are two peaks (A and B) around 15.5 deg latitude North in the CALIPSO data. None of them is present in the TROPOMI data with the old co-registration (red line). When the new scheme is used, the peak C appears approximately at the same latitude with CALIPSO peak B. At point C, the absolute difference between the two co-registration methods is approximately 800 m.





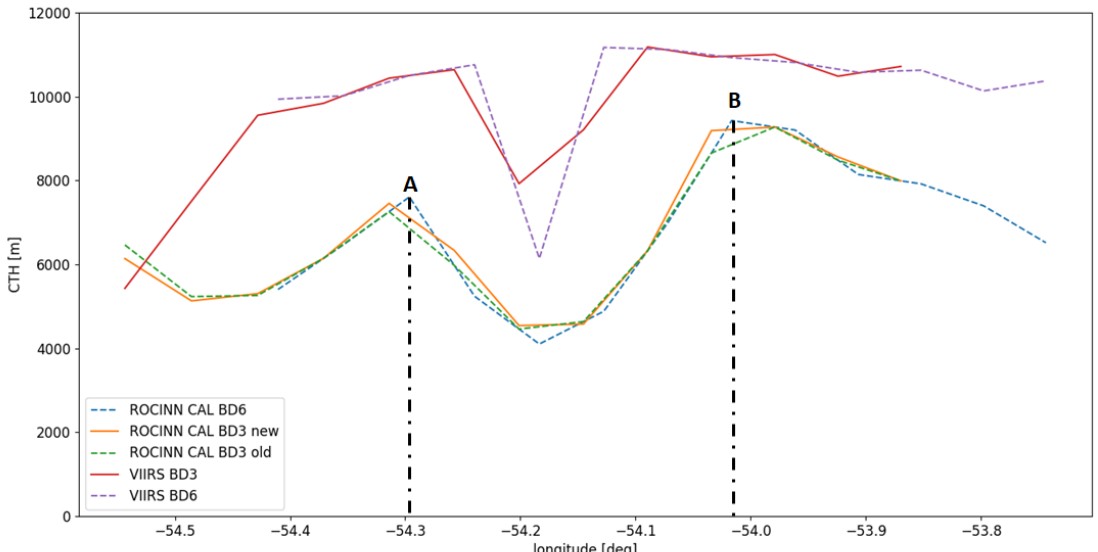

**Figure 34.** Inhomogeneous scene with two local maxima at points A and B where the new co-registration scheme has a positive impact on the ROCINN CAL CTH at BD3. The data refer to 2020-09-11, orbit 15099, scanline 2820, pixels 107-120.

## 5    Conclusions

The existence of collocated cloud information from VIIRS allowed the improvement of TROPOMI cloud properties through a better treatment of the spatial mis-alignment between UV/VIS and NIR footprints. The new scheme is applied on top of the old static mapping tables. The improvement on the TROPOMI data quality together with the optimizations of the co-registration scheme are summarized in the aforementioned bullets:

– From the daily scatter plots, we saw that under fully cloudy conditions (i.e., with cloud fraction 1 in the old scheme) the

co-registered cloud fraction obtains lower values with the new co-registration scheme. Several partially cloudy pixels have been characterized as cloud-free with the new co-registration scheme. The ROCINN CRB cloud height and CAL cloud-top height are scattered symmetrically around the identity line. Some asymmetry is observed at the cloud optical thickness with the scatter below the identity line being much higher than the scatter above the line.

– From the daily global maps showing the differences between the two schemes, we excluded systematic differences

present in certain geographical regions. In addition, we haven't found any latitudinal or viewing geometry dependency.

– From the bar-plot analysis, we found out that that the co-registration from UV/VIS to NIR with the new scheme results to a lower cloud fraction compared to the old scheme, when the original UV/VIS OCRA cloud fraction is high. Exactly the opposite happens when the UV/VIS OCRA cloud fraction takes low values: the co-registration with the new scheme results to higher NIR cloud fraction than with the old scheme. For almost fully cloudy conditions with a cloud fraction

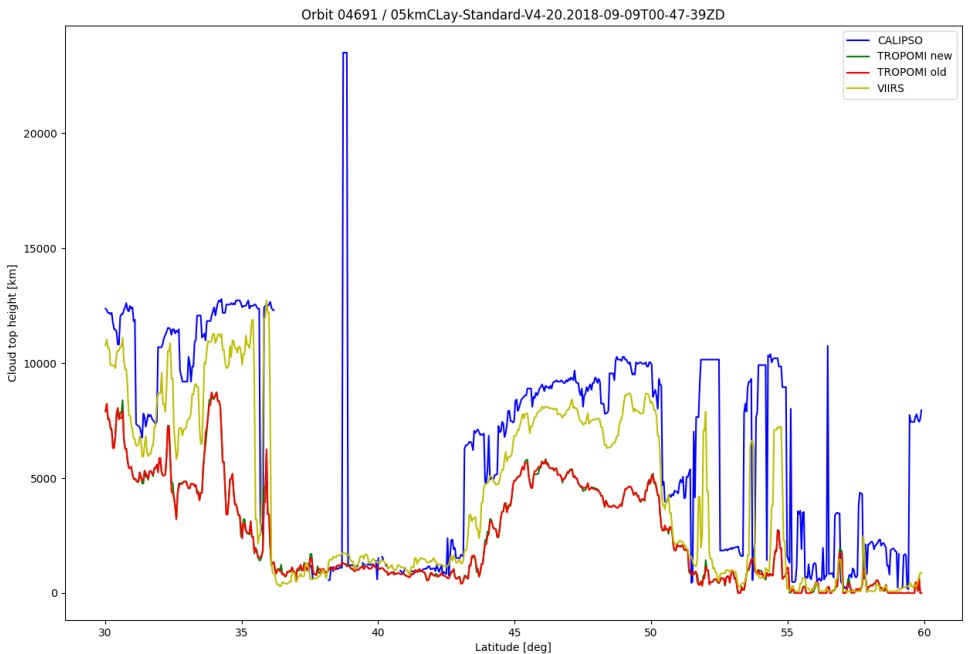

**Figure 35.** Comparison of the CTH for TROPOMI, VIIRS and CALIPSO for mid-latitudes in the northern hemisphere over the Pacific ocean. The TROPOMI orbit 04691 for day 2018-09-09 is collocated with the CALIPSO measurements from the 2018-09-09T00-47-39ZD overpass.

larger than 0.8, the co-registered from NIR to UV/VIS cloud-top height with the new and the old scheme agree to a high degree as the group of zero differences is dominated by the fully cloudy scenes. Under nearly fully cloudy conditions, the new co-registered cloud optical thickness is smaller than the old one whereas for the relative clear sky scenes, the new co-registered cloud optical thickness in UV/VIS is larger than the old one. In particular, for the scenes with very low cloud fractions below 0.2, we should expect a small increase of the co-registered cloud optical thickness. At the
relative cloud free scenes with OCRA cloud fraction lower than 0.2, we should expect an increase at the co-registered cloud albedo with the new scheme.

  – The cloud information from the complementary sensor (e.g., VIIRS for the TROPOMI co-registration) allows the re-structuring of the ROCINN retrieved parameters on the first UV/VIS detector pixel. The addition of the first row had primarily improved the gaps between two adjacent orbits when looking at a global map. Moreover, columnar information of UV/VIS trace gases is successfully retrieved for this first row, with a positive initial feedback for the accuracy of those



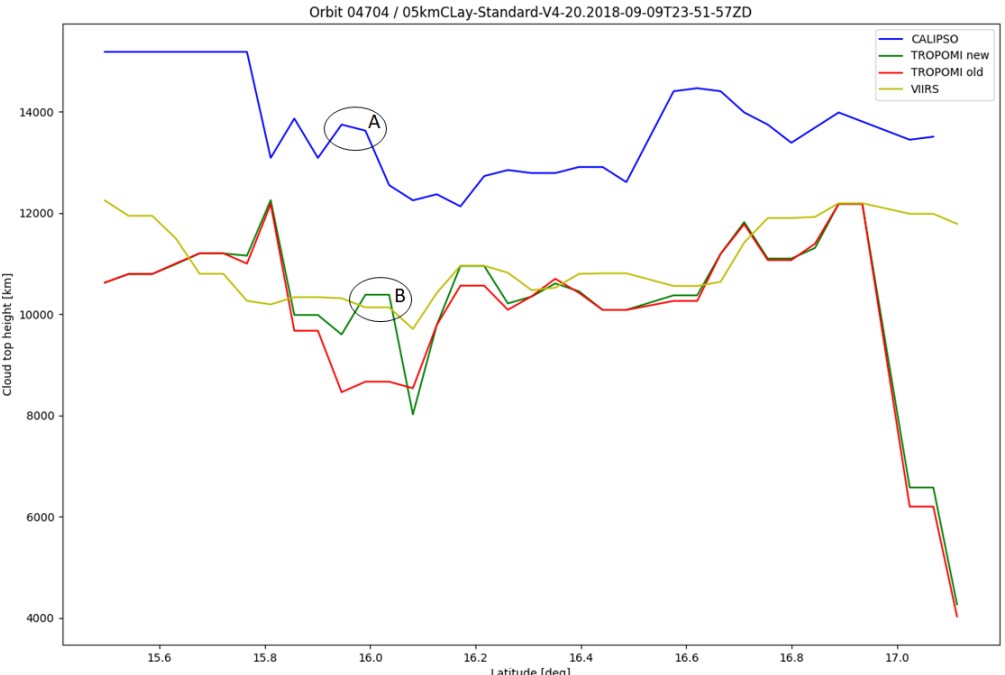

**Figure 36.** Comparison of the CTH for TROPOMI, VIIRS and CALIPSO for the tropics in the northern hemisphere over the Pacific ocean. The TROPOMI orbit 04704 for day 2018-09-09 is collocated with the CALIPSO measurements from the 2018-09-09T23-51-57ZD overpass.

retrievals for total ozone and tropospheric $SO_2$. An example scene in a day with a volcanic eruption showed that the detection algorithm for flagging $SO_2$ pixels with volcanic origin seems to work well for the additional row.

– From the validation exercise of TROPOMI against VIIRS in the across-track flight direction, the general conclusion is that the old co-registration scheme tends to smooth out local maxima and minima along the scanline. This is quite important finding because the original cloud parameter loses some structure which could be re-constructed through the use of the VIIRS data. This finding is valid for the cloud fraction and the cloud-top height.

– Another important point is that the co-registered value at the target band agrees much better with the original value at the source band when the new technique is used. This is valid for both cloud fraction and cloud-top height.

– The old co-registration scheme is applied when the old scheme for co-registering the OCRA CF when the VIIRS CF at both UV/VIS and NIR bands are equal to 1. This means that the applicability of the new scheme for the cloud fraction should be limited to non-fully cloudy scenes.

– From the validation exersice of TROPOMI against CALIPSO, we found several cases with better agreement with CALIPSO when using the new co-registration scheme. The agreement refers exclusively to the CTH structure in a qual-



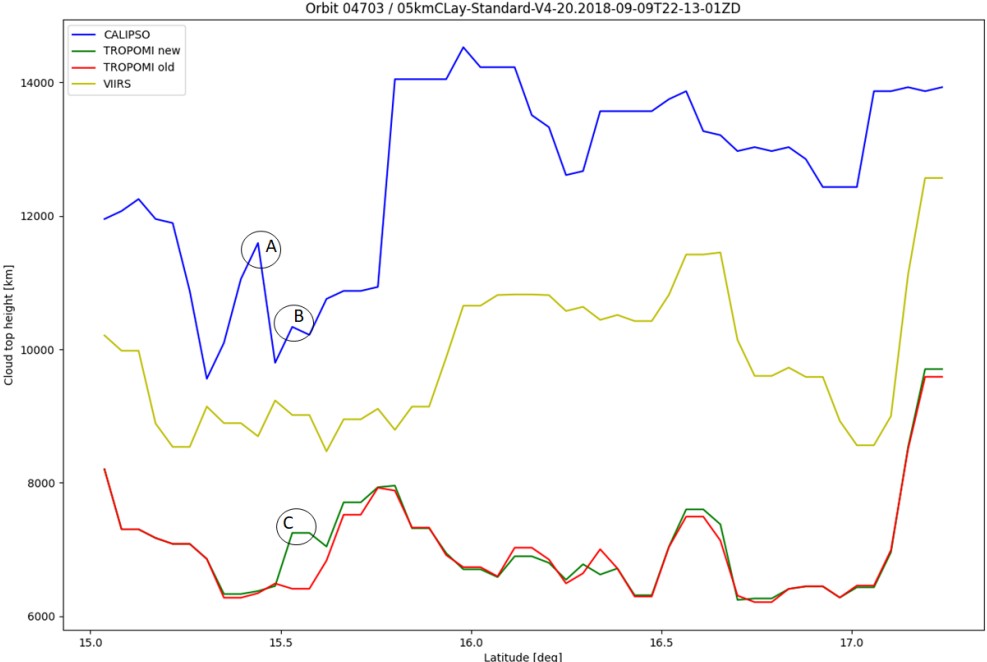

**Figure 37.** Comparison of the CTH for TROPOMI, VIIRS and CALIPSO for the tropics in the northern hemisphere over the Pacific ocean. The TROPOMI orbit 04703 for day 2018-09-09 is collocated with the CALIPSO measurements from the 2018-09-09T22-13-01ZD overpass.

itative manner. Quantitative comparison against CALIPSO CTHs would not be appropriate because there is a systematic
bias in TROPOMI CTH associated with the lack of ice cloud parameterization in the forward cloud model.

The new co-registration scheme has been incorporated into the operational processing system for S5P. The latest UPAS processor version 2.6 has been effective starting from 2023-11-26, orbit 31705. Later, when the Sentinel-4 satellite will be launched, a similar approach will be used for the treatment of the spatial mis-registration, using collocated FCI data.

*Data availability.* The S5P Level-2 CLOUD product refers to UPAS Version 2.4 and can be accessed from the Copernicus Data Space
Ecosystem search tool (https://dataspace.copernicus.eu/). The S-NPP VIIRS data mapped to the Tropomi grids for band 3, 6, and 7 are also available in the Copernicus Data Space Ecosystem search tool. The re-gridded S5P-NPP cloud data in BD3 and BD6 for the test days are not publicly available since this dataset was explicitly built to support the development of the new co-registration scheme. The L2 cloud layer "CAL_LID_L2_05kmCLay-Standard-V4-20" Version 4-20 data product has been used (NASA/LARC/SD/ASDC, 2018). Data generation and distribution of this V4.20 product ended on July 1, 2020 to support a change in the operating system of the CALIPSO production clusters.
The V4.21 data product covers July 1, 2020 to current.



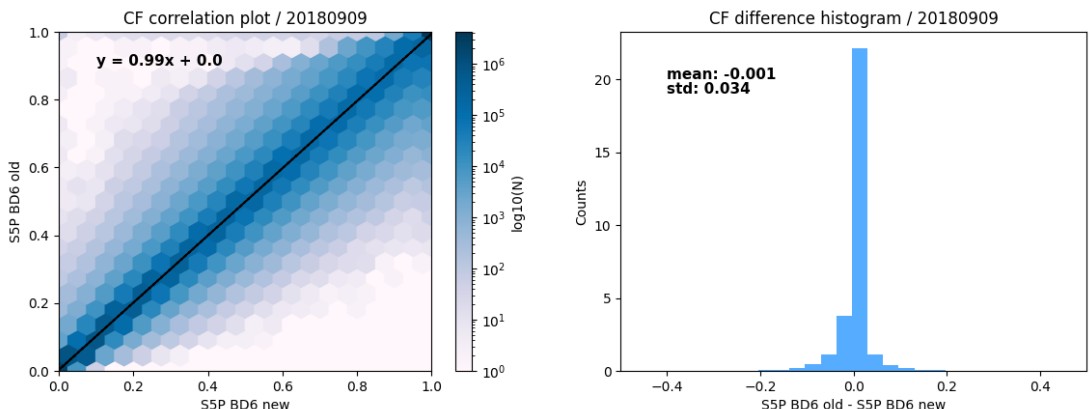

**Figure A1.** The co-registered cloud fraction for the new versus the old scheme: data analysis refers to 2018-09-09

## Appendix A: Additional plots from the comparison of the two co-registration schemes

This section contains the additional plots for the rest of the comparisons between new and old co-registration scheme. Those plots are complementary to the ones at the main text.

## Appendix B: Collocation method for TROPOMI and CALIPSO

In Fig. B1, the visualization of the CALIPSO measurements within the TROPOMI pixels in BD3 is presented. The TROPOMI/-CALIPSO collocation method requires the calculation of the distance between the CALIPSO data and the center of each pixel in a predefined search area of the given TROPOMI orbit. The search window takes the geographic coordinates from CALIPSO and creates a rectangular with $[\phi_C - 0.1, \phi_C + 0.1]$ and $[\lambda_C - 0.05, \lambda_C + 0.05]$ where $\phi_C$, $\lambda_C$ refer to the latitude and longitude of CALIPSO points. The distance between the CALIPSO coordinates and the center of the TROPOMI pixels within the search

window are calculated according to the formulas below:

$$\Delta\phi = \mathrm{rad}(\phi_2 - \phi_1), \Delta\lambda = \mathrm{rad}(\lambda_2 - \lambda_1), \tag{B1}$$

$$\alpha = \left(\sin\frac{\Delta\phi}{2}\right)^2 + \left(\sin\frac{\Delta\lambda}{2}\right)^2 \cos\left(\mathrm{rad}\left(\phi_1\right)\right)\cos\left(\mathrm{rad}\left(\phi_2\right)\right) \tag{B2}$$

$$c = 2\,\mathrm{atan2}\left(\sqrt{\alpha}, \sqrt{1-\alpha}\right) \tag{B3}$$

$$d = Rc, R = 6378137m \tag{B4}$$

The CALIPSO measurement is then located within the TROPOMI pixel with the minimum found distance $d$.





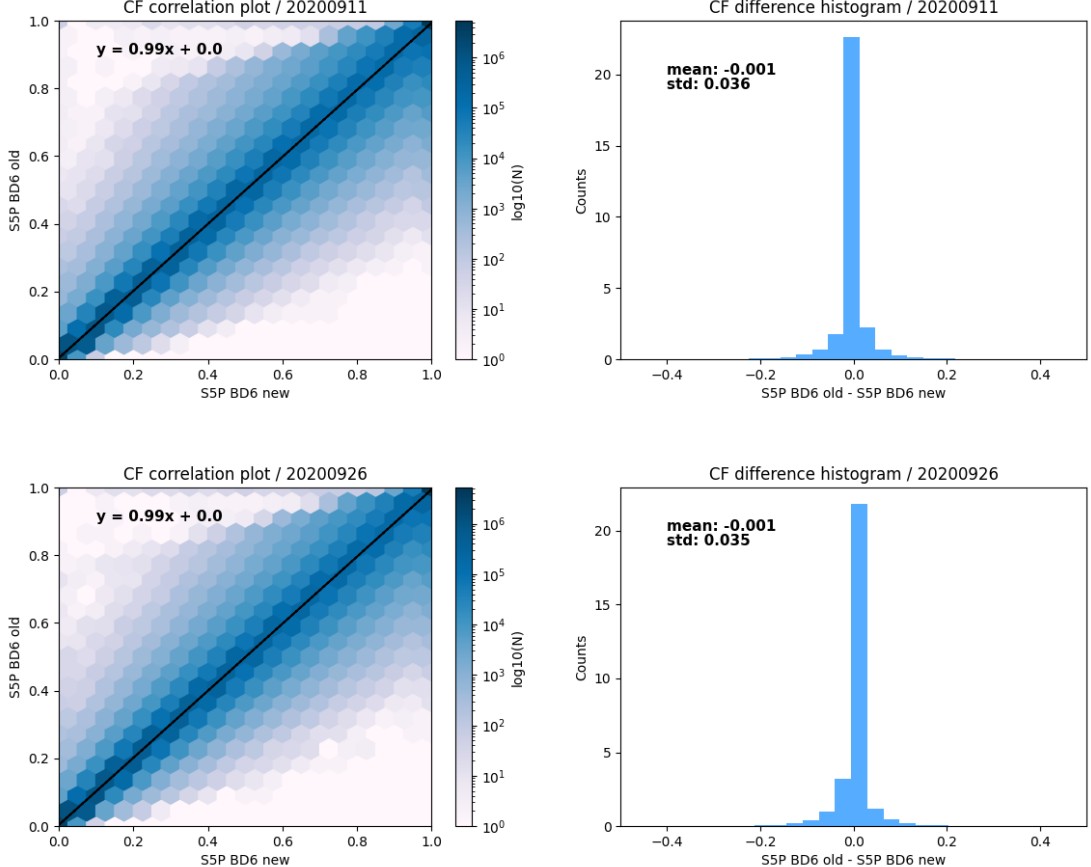

**Figure A2.** The co-registered cloud fraction for the new versus the old scheme: data analysis refer to 2020: upper panel for 11th Sept. and lower panel for 26th Sept.

*Author contributions.* AA, RL and DL developed the conceptual approach of the new co-registration scheme. AA implemented the co-registration prototype algorithm in Python. FR incorporated the prototype algorithm into the operational UPAS system. For TROPOMI, RL was responsible for the development of the prototype OCRA algorithm. AA and VMG were responsible for the development of the prototype ROCINN cloud retrieval algorithm. RS developed the S5P-NPP processor and provided the VIIRS re-gridded data. AA performed the analysis of the impact of the new algorithm on the cloud parameters based on the comparison of the available datasets from VIIRS and CALIPSO. K-PH and PH studied the impact of the new scheme on total ozone column and $SO_2$, respectively. AA prepared the manuscript with contributions from all co-authors.


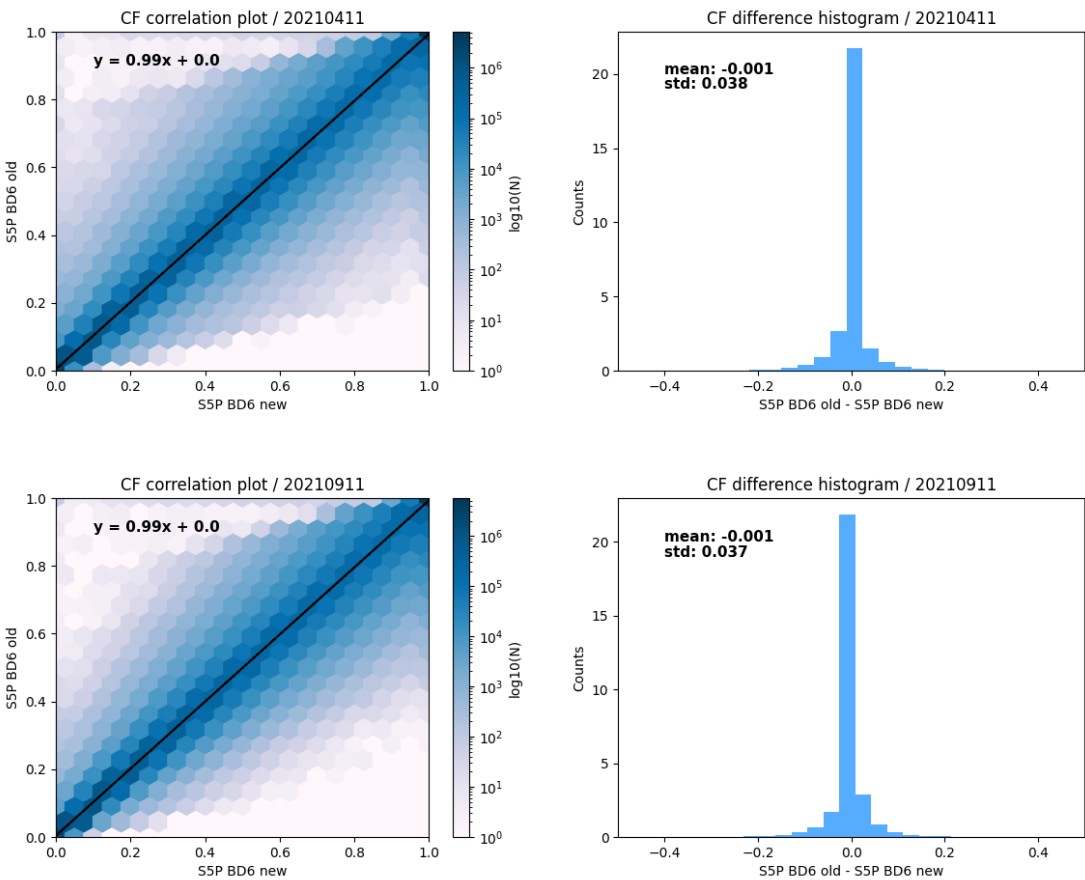

**Figure A3.** The co-registered cloud fraction for the new versus the old scheme: data analysis refer to 2021: upper panel for 11th Apr. and lower panel for 11th Sept.

*Competing interests.*   The authors have the following competing interests: At least one of the (co-)authors is a member of the editorial board of Atmospheric Measurement Techniques.

*Acknowledgements.*   We would like to thank Rob Spurr for his continuous support on LIDORT. In addition, we thank our colleagues from the Royal Belgian Institute for Space Aeronomy (BIRA-IASB) Michel Van Roozendael, Jeroen Van Gent and Nicolas Theys for their support on the O3 GODFit and SO$_2$ data. Moreover, we thank all the former and current DLR colleagues who contributed to the algorithm development in the UPAS system. We also acknowledge DLR internal S5P project (KTR 2472046) for financing the algorithm development, ESA ATM-MPC and ESA S5P PDGS projects for financing the operational implementation.





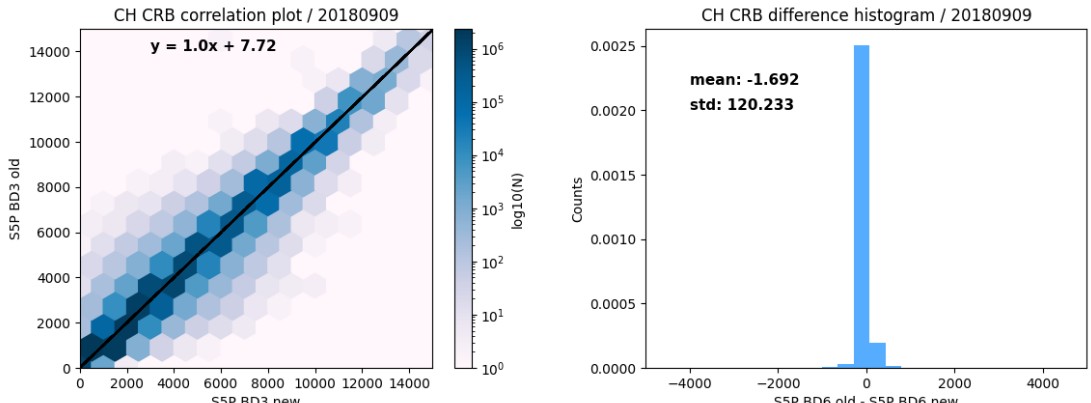

**Figure A4.** The co-registered ROCINN CRB cloud height for the new versus the old scheme: data analysis refers to 2018-09-09

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



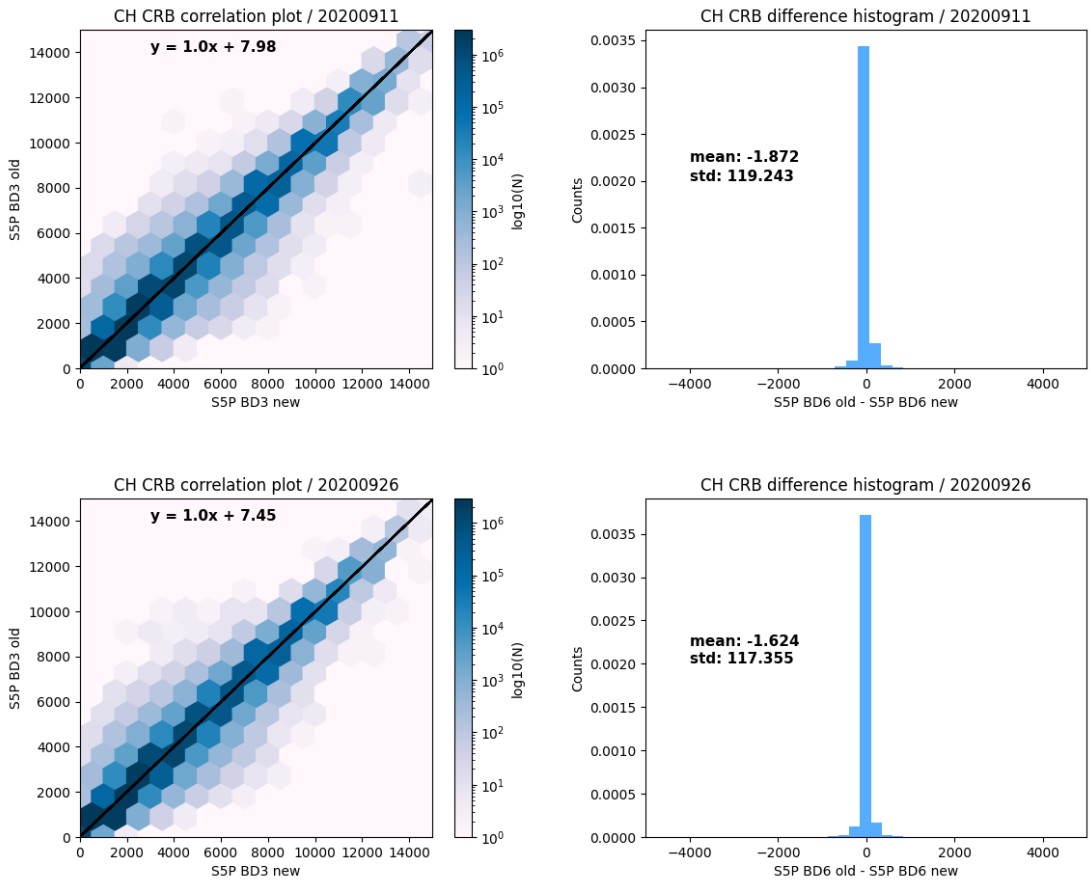

**Figure A5.** The co-registered ROCINN CRB cloud height for the new versus the old scheme: data analysis refer to 2020: upper panel for 11th Sept. and lower panel for 26th Sept.

Heidinger, A., Li, Y., and Wanzong, S.: Algorithm Theoretical Basis Document For Enterprise AWG Cloud Height Algorithm (ACHA), Version 3.4, https://www.star.nesdis.noaa.gov/jpss/documents/ATBD/ATBD_EPS_Cloud_ACHA_v3.4.pdf, 2020.

Heidinger, A. K., Evan, A. T., Foster, M. J., and Walther, A.: A Naive Bayesian Cloud-Detection Scheme Derived from CALIPSO and Applied within PATMOS-x, Journal of Applied Meteorology and Climatology, 51, 1129 – 1144, https://doi.org/10.1175/JAMC-D-11-02.1, 2012.

Heue, K.-P., Eichmann, K., and Valks, P.: TROPOMI/S5P ATBD of tropospheric ozone data products, Tech. Rep. S5P-L2-IUP-ATBD-400C, issue 1.6, Deutsches Zentrum fur Luft- und Raumfahrt e.V. in der Helmholtz Gemeinschaft and Institute for Environmental Physics (IUP), https://sentinel.esa.int/documents/247904/2476257/Sentinel-5P-ATBD-TROPOMI-Tropospheric-Ozone, 2018.

King, M. D.: Determination of the Scaled Optical Thickness of Clouds from Reflected Solar Radiation Measurements, Journal of Atmospheric Sciences, 44, 1734–1751, https://doi.org/10.1175/1520-0469(1987)044<1734:DOTSOT>2.0.CO;2, 1987.



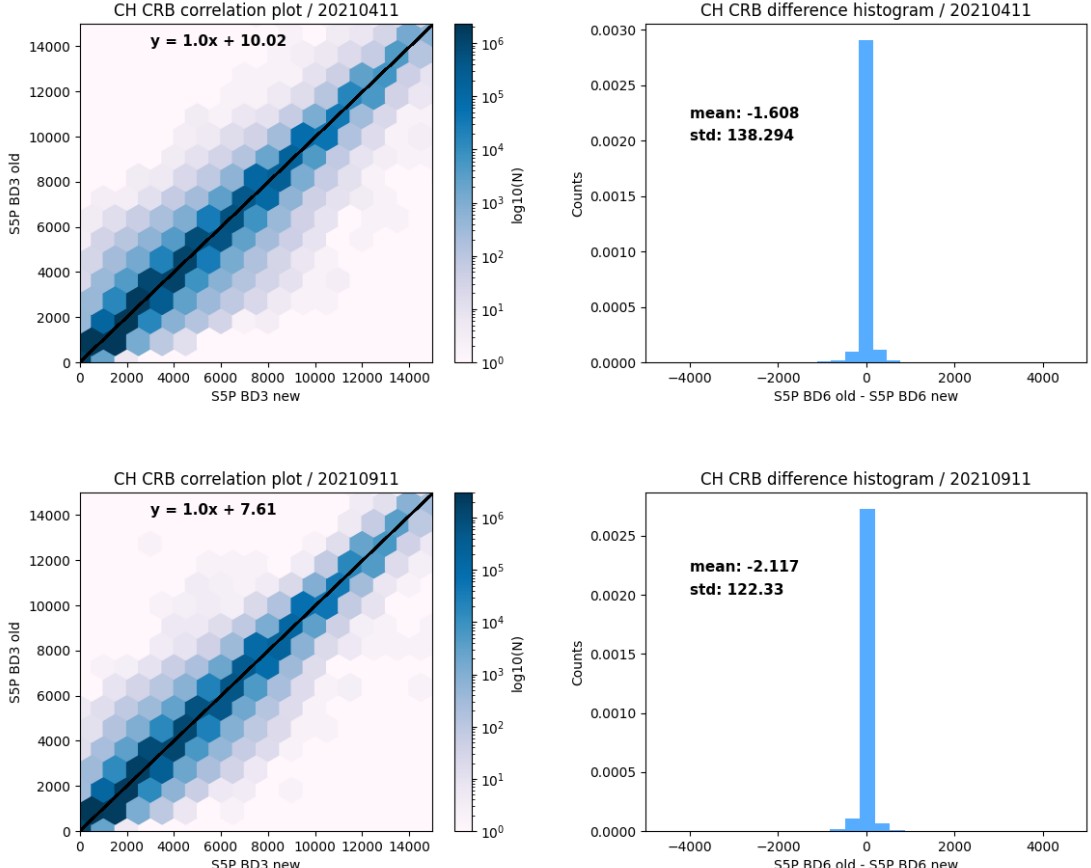

**Figure A6.** The co-registered ROCINN CRB cloud height for the new versus the old scheme:data analysis refer to 2021: upper panel for 11th Apr. and lower panel for 11th Sept.

KNMI: S5P/TROPOMI Algorithm theoretical basis document for the TROPOMI L01b data processor, S5P-KNMI-L01B-0009-SD, issue 10.0.0, 2022.

Kokhanovsky, A. A. and Mayer, B.: Light reflection and transmission by non-absorbing turbid slabs: simple approximations, Journal of Optics A: Pure and Applied Optics, 5, 43–46, https://doi.org/10.1088/1464-4258/5/1/306, 2003.

Latsch, M., Richter, A., Eskes, H., Sneep, M., Wang, P., Veefkind, P., Lutz, R., Loyola, D., Argyrouli, A., Valks, P., Wagner, T., Sihler, H., van Roozendael, M., Theys, N., Yu, H., Siddans, R., and Burrows, J. P.: Intercomparison of Sentinel-5P TROPOMI cloud products for tropospheric trace gas retrievals, Atmospheric Measurement Techniques, 15, 6257–6283, https://doi.org/10.5194/amt-15-6257-2022, 2022.




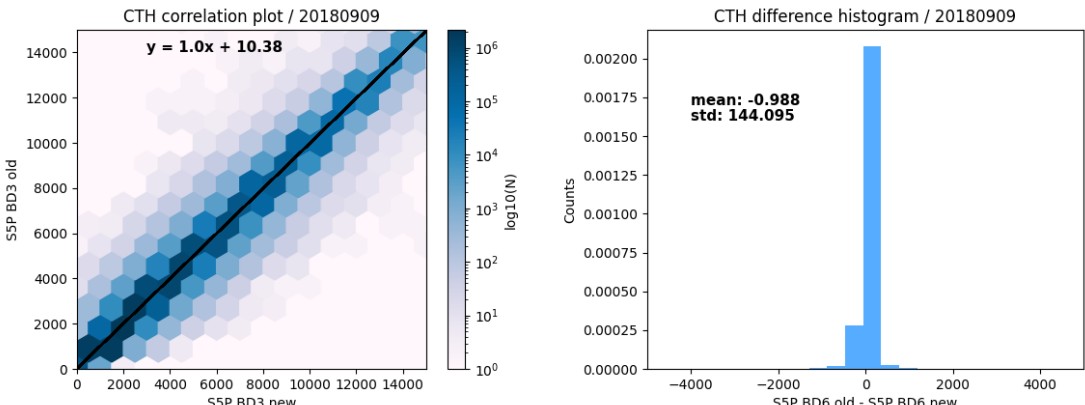

**Figure A7.** The co-registered ROCINN CAL cloud-top height for the new versus the old scheme: data analysis refers to 2018-09-09

Liu, C.-Y., Chiu, C.-H., Lin, P.-H., and Min, M.: Comparison of cloud-top property retrievals from Advanced Himawari Imager, MODIS, CloudSat/CPR, CALIPSO/CALIOP, and radiosonde, Journal of Geophysical Research: Atmospheres, 125, e2020JD032 683, https://doi.org/10.1029/2020JD032683, 2020.

Liu, S., Valks, P., Pinardi, G., Xu, J., Chan, K. L., Argyrouli, A., Lutz, R., Beirle, S., Khorsandi, E., Baier, F., Huijnen, V., Bais, A., Donner, S., Dörner, S., Gratsea, M., Hendrick, F., Karagkiozidis, D., Lange, K., Piters, A. J. M., Remmers, J., Richter, A., Van Roozendael,

M., Wagner, T., Wenig, M., and Loyola, D. G.: An improved TROPOMI tropospheric $NO_2$ research product over Europe, Atmospheric Measurement Techniques, 14, 7297–7327, https://doi.org/10.5194/amt-14-7297-2021, 2021.

Loyola, D.: Methodologies for solving Satellite Remote Sensing Problems using Neuro Computing Techniques, Ph.D. thesis, Technische Universität München, verlag Dr. Hut, ISBN 978-3-8439-1068-2, 2013.

Loyola, D., Lutz, R., Argyrouli, A., and Spurr, R.: S5P/TROPOMI Algorithm Theoretical Basis Document Cloud Products, S5P-DLR-L2-

ATBD-400I, issue 2.6.1, 2023.

Loyola, D. G.: Automatic cloud analysis from polar-orbiting satellites using neural network and data fusion techniques, in: IGARSS 2004. 2004 IEEE International Geoscience and Remote Sensing Symposium, vol. 4, pp. 2530–2533 vol.4, https://doi.org/10.1109/IGARSS.2004.1369811, 2004.

Loyola, D. G., Thomas, W., Spurr, R., and Mayer, B.: Global patterns in daytime cloud properties derived from GOME backscatter UV-VIS

measurements, International Journal of Remote Sensing, 31, 4295–4318, https://doi.org/10.1080/01431160903246741, 2010.

Loyola, D. G., Pedergnana, M., and Gimeno García, S.: Smart sampling and incremental function learning for very large high dimensional data, Neural Networks, 78, 75–87, https://doi.org/https://doi.org/10.1016/j.neunet.2015.09.001, special Issue on "Neural Network Learning in Big Data", 2016.

Loyola, D. G., Gimeno García, S., Lutz, R., Argyrouli, A., Romahn, F., Spurr, R. J. D., Pedergnana, M., Doicu, A., Molina García, V.,

and Schüssler, O.: The operational cloud retrieval algorithms from TROPOMI on board Sentinel-5 Precursor, Atmospheric Measurement Techniques, 11, 409–427, https://doi.org/10.5194/amt-11-409-2018, 2018.


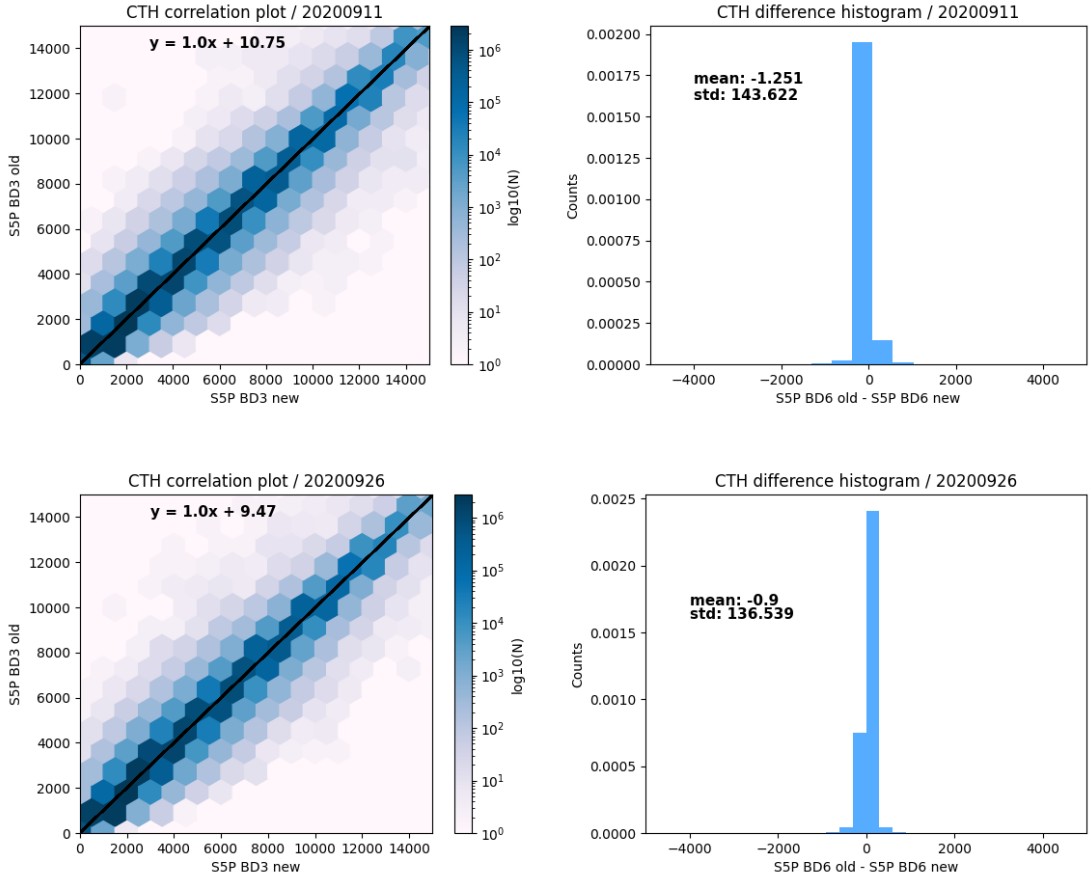

**Figure A8.** The co-registered ROCINN CAL cloud-top height for the new versus the old scheme: data analysis refer to 2020: upper panel for 11th Sept. and lower panel for 26th Sept.

Lutz, R., Loyola, D. G., Gimeno García, S., and Romahn, F.: OCRA radiometric cloud fractions for GOME-2 on MetOp-A/B, Atmospheric Measurement Techniques, 9, 2357–2379, https://doi.org/10.5194/amt-9-2357-2016, 2016.

Molina García, V.: Retrieval of cloud properties from EPIC/DSCOVR, Ph.D. thesis, Technische Universität München, https://mediatum.ub. tum.de/?id=1662361, 2022.

Nakajima, T. and King, M. D.: Determination of the optical thickness and effective particle radius of clouds from reflected solar radiation measurements. I - Theory, Journal of Atmospheric Sciences, 47, 1878–1893, https://doi.org/10.1175/1520-0469(1990)047<1878:DOTOTA>2.0.CO;2, 1990a.

Nakajima, T. and King, M. D.: Asymptotic theory for optically thick layers: application to the discrete ordinates method, Appl Optics, 31, 7669–7683, https://doi.org/10.1364/AO.31.007669, 1990b.





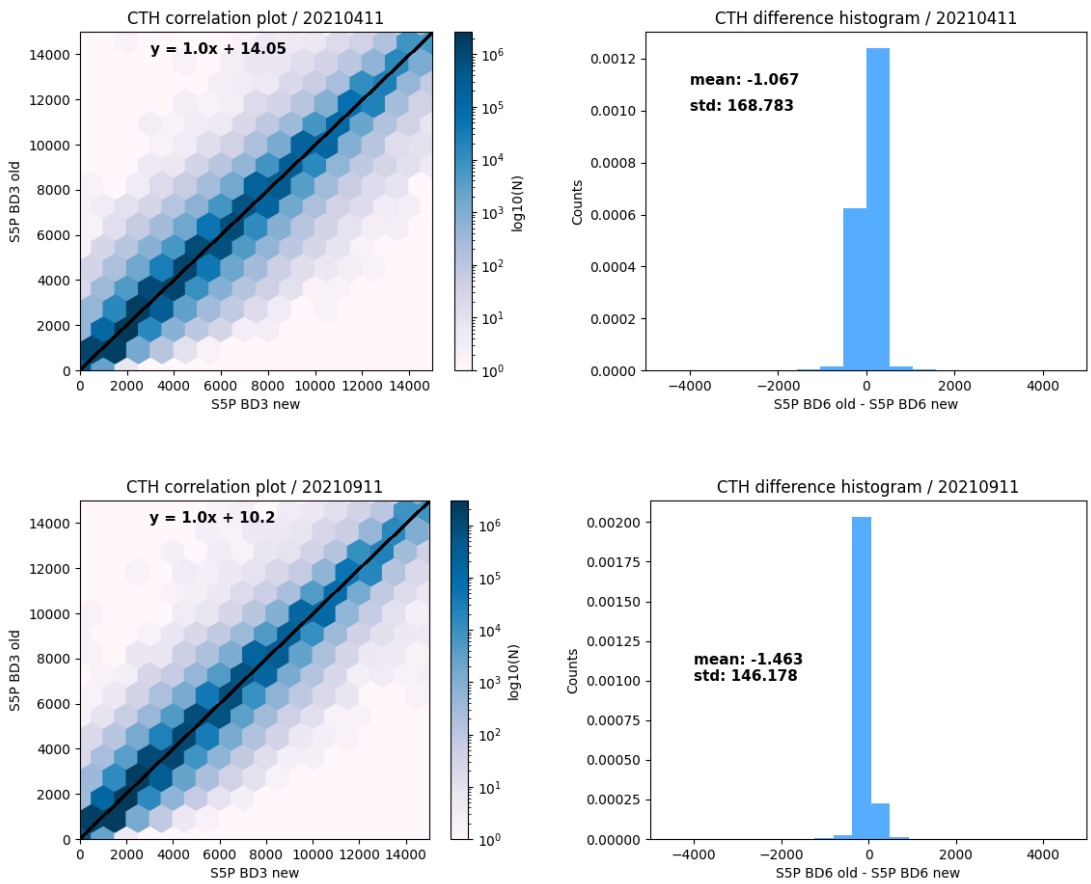

**Figure A9.** The co-registered ROCINN CAL cloud-top height for the new versus the old scheme: data analysis refer to 2021: upper panel for 11th Apr. and lower panel for 11th Sept.

NASA/LARC/SD/ASDC: CALIPSO Lidar Level 2 5 km Cloud Layer, V4-20, 10.5067/CALIOP/CALIPSO/LID_L2_05KMCLAY-STANDARD-V4-20, 2018.

Rodgers, C. D.: Retrieval of atmospheric temperature and composition from remote measurements of thermal radiation, Reviews of Geophysics, 14, 609–624, https://doi.org/10.1029/RG014i004p00609, 1976.

Schuessler, O., Loyola, D. G., Doicu, A., and Spurr, R. J. D.: Information Content in the Oxygen A-Band for the Retrieval of Macrophysical Cloud Parameters, IEEE Transactions on Geoscience and Remote Sensing, 52, 3246–3255, https://doi.org/10.1109/TGRS.2013.2271986, 2014.

Siddans, R.: S5P-NPP Cloud Processor ATBD, Tech. Rep. S5P-NPPC-RAL-ATBD-0001, issue 1.0, https://sentinel.esa.int/documents/247904/2476257/Sentinel-5P-NPP-ATBD-NPP-Clouds, 2016.

Siddans, R.: S5P-NPP Cloud Processor Algorithm Theoretical Basis Document, S5P-NPPC-RAL-ATBD-0001, issue 2.0.0, 2022.





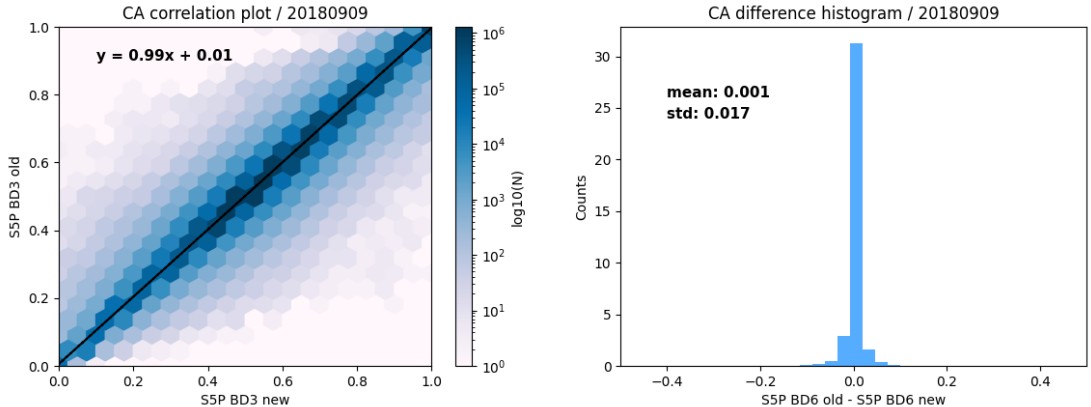

**Figure A10.** The co-registered ROCINN CRB cloud albedo for the new versus the old scheme: data analysis refer to 2018-09-09

Sneep, M.: Sentinel-5 precursor inter-band coregistration mapping tables, S5P-KNMI-L2-0129-TN, issue 4.0.0, 2015.

Spurr, R. J. D.: VLIDORT: A linearized pseudo-spherical vector discrete ordinate radiative transfer code for forward model and retrieval studies in multilayer multiple scattering media, J. Quant. Spectrosc. Ra., 102, 316–342, https://doi.org/10.1016/j.jqsrt.2006.05.005, 2006.

Spurr, R. J. D., Loyola, D., Heue, K.-P., Van Roozendael, M., and Lerot, C.: S5P/TROPOMI Total Ozone ATBD, Tech. Rep. S5P-L2-DLR-
ATBD-400A, issue 2.3, Deutsches Zentrum fur Luft- und Raumfahrt e.V. in der Helmholtz Gemeinschaft and Royal Belgian Institute for Space Aeronomy (BIRA-IASB), https://sentinel.esa.int/documents/247904/2476257/Sentinel-5P-TROPOMI-ATBD-Total-Ozone, 2021.

Theys, N., De Smedt, I., Yu, H., Danckaert, T., van Gent, J., Hörmann, C., Wagner, T., Hedelt, P., Bauer, H., Romahn, F., Pedergnana, M., Loyola, D., and Van Roozendael, M.: Sulfur dioxide retrievals from TROPOMI onboard Sentinel-5 Precursor: algorithm theoretical basis, Atmospheric Measurement Techniques, 10, 119–153, https://doi.org/10.5194/amt-10-119-2017, 2017.

Van de Hulst, H. C.: Light scattering by small particles, Wiley, NY, 1957.

Veefkind, J., Aben, I., McMullan, K., Förster, H., de Vries, J., Otter, G., Claas, J., Eskes, H., de Haan, J., Kleipool, Q., van Weele, M., Hasekamp, O., Hoogeveen, R., Landgraf, J., Snel, R., Tol, P., Ingmann, P., Voors, R., Kruizinga, B., Vink, R., Visser, H., and Levelt, P.: TROPOMI on the ESA Sentinel-5 Precursor: A GMES mission for global observations of the atmospheric composition for climate, air quality and ozone layer applications, Remote Sensing of Environment, 120, 70–83, https://doi.org/10.1016/j.rse.2011.09.027, the Sentinel
Missions - New Opportunities for Science, 2012.

Walther, A. and Heidinger, A. K.: Implementation of the Daytime Cloud Optical and Microphysical Properties Algorithm (DCOMP) in PATMOS-x, Journal of Applied Meteorology and Climatology, 51, 1371–1390, https://doi.org/10.1175/JAMC-D-11-0108.1, 2012.

Walther, A. and Straka, W.: Algorithm Theoretical Basis Document For Daytime Cloud Optical and Microphysical Properties (DCOMP), Version 1.2, https://www.star.nesdis.noaa.gov/jpss/documents/ATBD/ATBD_EPS_Cloud_DCOMP_v1.2.pdf, 2020.

Winker, D., Pelon, J., and Mccormick, M.: The CALIPSO mission: Spaceborne lidar for observation of aerosols and clouds, Proc. SPIE-Int. Soc. Opt. Eng., 4893, 1–11, https://doi.org/10.1117/12.466539, 2003.

Winker, D., Hunt, W., and Hostetler, C.: Status and performance of the CALIOP lidar, Proceedings of SPIE - The International Society for Optical Engineering, 5575, https://doi.org/10.1117/12.571955, 2004.

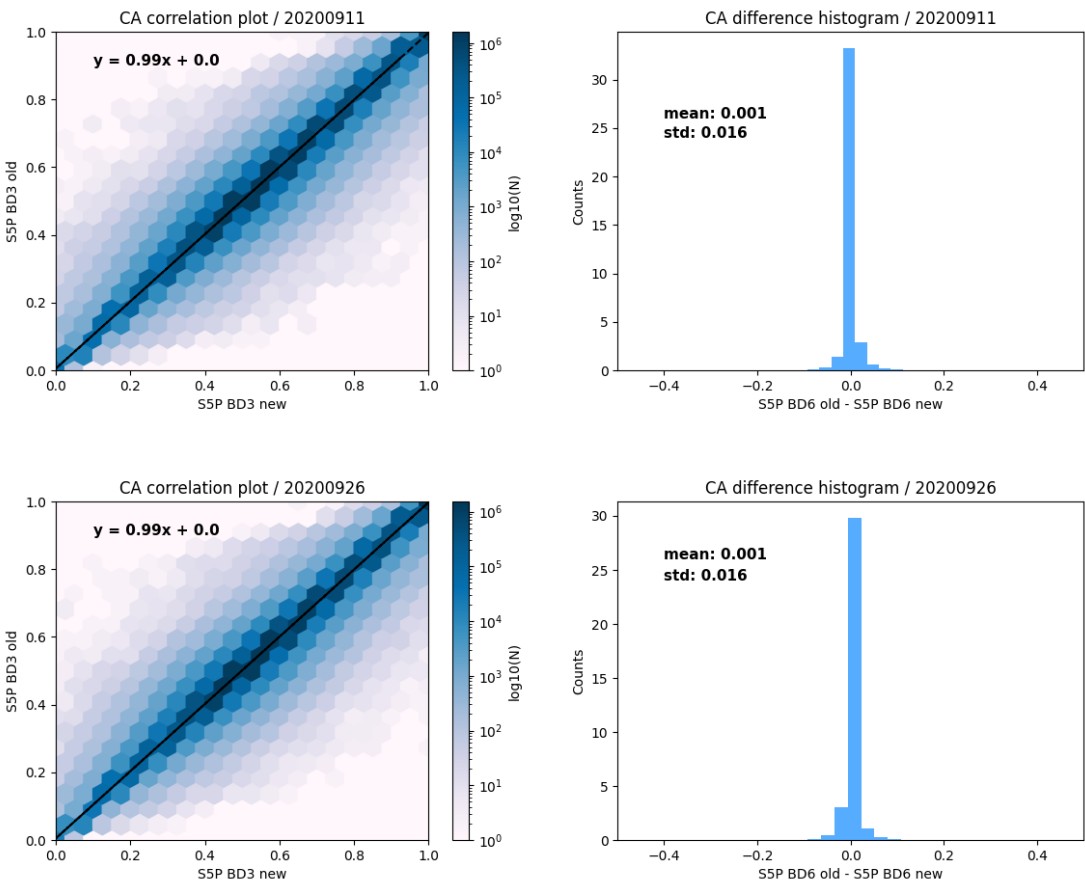

**Figure A11.** The co-registered ROCINN CRB cloud albedo for the new versus the old scheme: data analysis refer to 2020: upper panel for 11th Sept. and lower panel for 26th Sept.

Winker, D. M., Hunt, W. H., and McGill, M. J.: Initial performance assessment of CALIOP, Geophysical Research Letters, 34, https://doi.org/10.1029/2007GL030135, 2007.




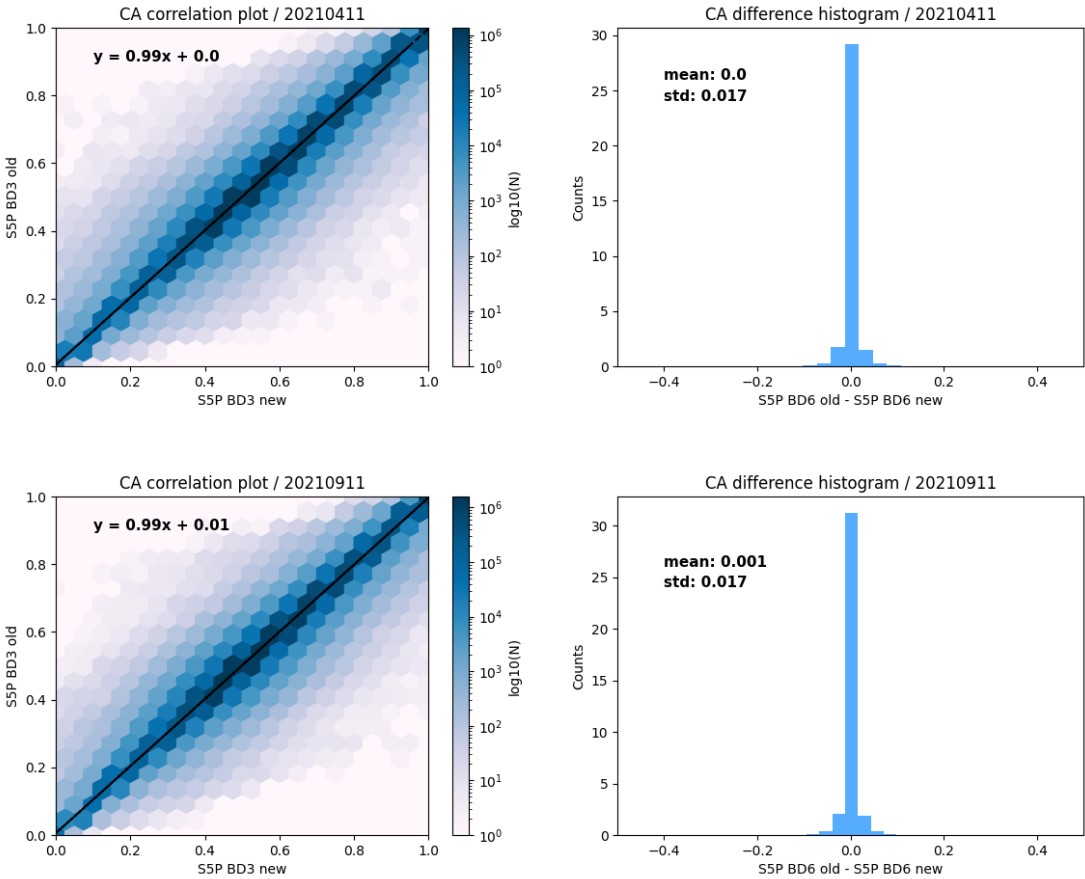

**Figure A12.** The co-registered ROCINN CRB cloud albedo for the new versus the old scheme: data analysis refer to 2021: upper panel for the 11th Apr. and lower panel for the 11th Sept.





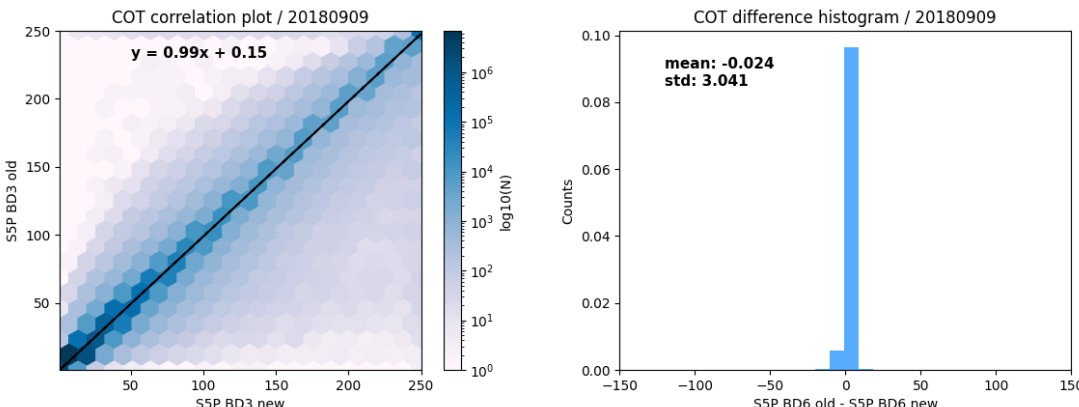

**Figure A13.** The co-registered ROCINN CAL cloud optical thickness for the new versus the old scheme: data analysis refers to 2018-09-09

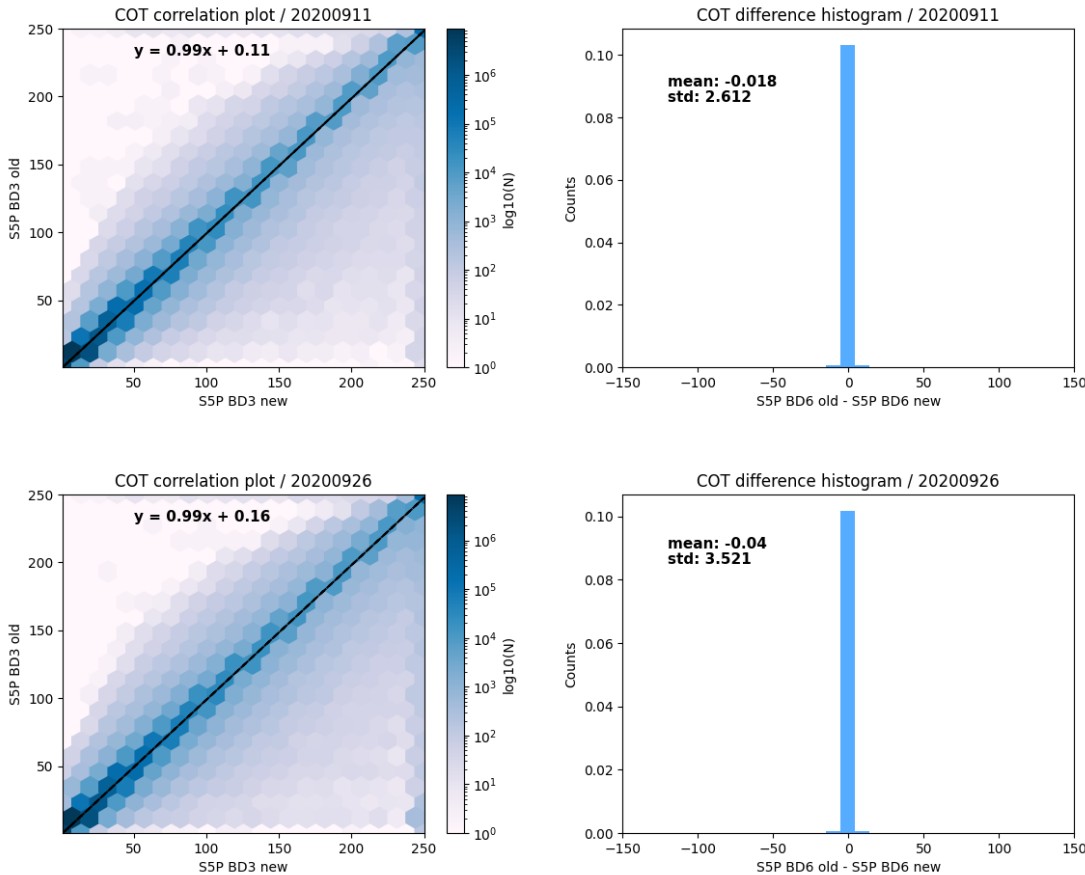

**Figure A14.** The co-registered ROCINN CAL cloud optical thickness for the new versus the old scheme: data analysis refers to 2020: upper panel for 11th Sept. and lower panel for 26th Sept.



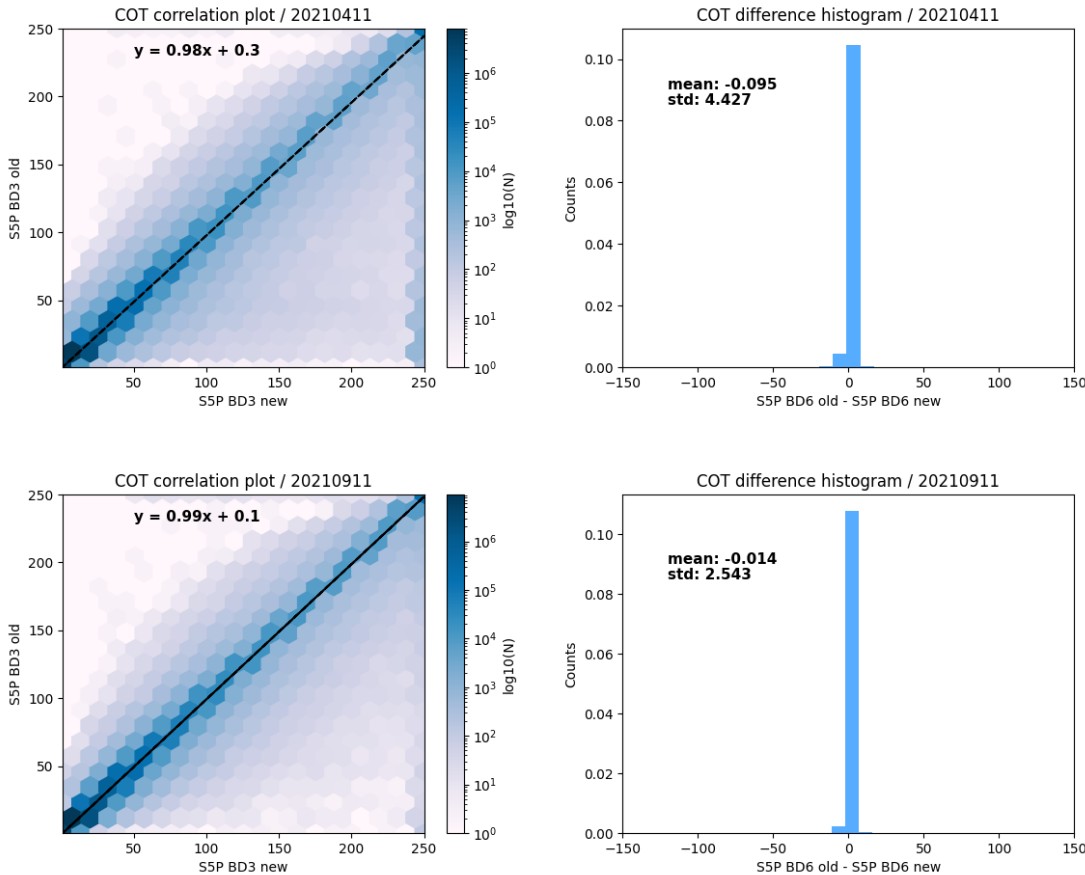

**Figure A15.** The co-registered ROCINN CAL cloud optical thickness for the new versus the old scheme: data analysis refers to 2021: upper panel for the 11th Apr. and lower panel for the 11th Sept.



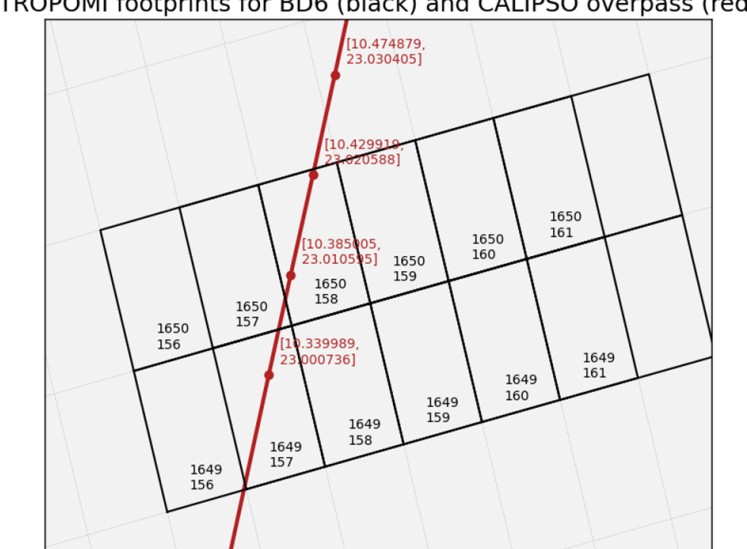

**Figure B1.** TROPOMI footprints at BD3 (black color) and CALIPSO measurements (red color). Every TROPOMI pixel contains two numbers; the first refers to the scanline and the second to the pixel id. The CALIPSO measurements are identified directly with the geographic coordinates.