# Peer review of "An advanced spatial co-registration of cloud properties for the atmospheric Sentinel missions: Application to TROPOMI"

_Atmospheric Measurement Techniques, 2024_

## Referee Comment (RC1)

**Review of AMT-2024-28**
**"An advanced spatial co-registration of cloud properties for the atmospheric Sentinel missions: Application to TROPOMI"**

**28th April 2024**

Review of the article submitted to AMT with the title "An advanced spatial co-registration of cloud properties for the atmospheric Sentinel missions: Application to TROPOMI", by Athina Argyrouli, Diego Loyola, Fabian Romahn, Ronny Lutz, Víctor Molina García, Pascal Hedelt, Klaus-Peter Heue, and Richard Siddans. AMT article number "AMT-2024-28".

In this review I will frequently refer to the L1B ATBD, which can be found in the Sentinel 5P document library. The article already refers to this document as (KNMI, 2022).

**1 General comments**

The article is generally well written. In some places details are left out that should in my view be included, in other places more detail than is perhaps desired can be found. The latter is especially the case when presenting the results. I'll get back to these cases in the specific comments section. The article is long, and should be reduced in size. I will give some suggestions.

The mapping equations are introduced without much justification as to why one would want to use these specific quantities from VIIRS te aid in the transformations. That you use a derived fractional cloud cover from VIIRS to map the cloud fraction from UV-VIS to NIR is fairly logical, but why not use the $\gamma$ factors derived from the VIIRS cloud fraction as well as the dimensionless scaling factor to aid in the mapping of the cloud height from NIR to UV-VIS? After all, you do get a weight that is related to the amount of clouds in the pixel, and therefore a means to obtain a weighted average.

This brings me to my second more important point. Using the cloud fractions would preclude extrapolation to row 0, but at the same time, how much of the information in row 0 is actually coming from the Tropomi instrument? A discussion on the effective contributions of the information from Tropomi and VIIRS in the final product is desired. Is the information from row 0 really VIIRS information that has been scaled locally by the ratio of Tropomi and VIIRS cloud heights to take into account the systematic model difference between the two instruments and retrieval methods? This discussion really needs to be added to the article before publication.

**2 Specific comments**

**2.1 Spatial resolution**

In the introduction, lines 33 to 37, the spatial resolution of Tropomi is discussed. For the bands used by OCRA/ROCINN the spatial resolution in the flight direction is unimportant. Because of the shared entrance slit between the VIS and NIR spectrometers (See L1B ATBD,

section 7.2), there is no spatial misalignment between those spectrometers in the flight direction. That also means that the change of the spatial resolution in the flight direction is unimportant for the subject of this article. I suggest to shorten this discussion in the introduction.

**2.2  Algorithm description**

In section 2 of the article a description of the cloud retrieval algorithm is given. The algorithm is based on a pair of algorithms, OCRA for the cloud fraction, and ROCINN for the cloud top height and cloud albedo or cloud optical thickness, depending on the cloud model. Please explain how a cloud fraction can be derived from OCRA without assuming a cloud albedo, or put differently, show that the three parameters are independent, and the cloud albedo (or optical thickness) isn't (implicitly) assumed already in the cloud fraction derivation in OCRA.

**2.3  Spatial alignment in the flight direction**

In section 3, on lines 97 to 99, the authors mention that the static lookup tables used up to now indicate a very small mis-match in the flight directions. These are rounding errors in the algorithm used to calculate these tables. The design of the instrument itself (See L1B ATBD, section 7.2) *guarantees* that there is no spatial mismatch in the flight direction between the VIS and NIR spectrometers (bands 3 – 6). So the algorithm does not neglect anything, but it handles the situation correctly. Please adjust the text accordingly.

**2.4  Previous treatment of the spatial mis-registration**

In section 3.1 the previous treatment of the mis-registration is described. Here a "cloud co-registration inhomogeneity parameter" is introduced. This parameter involves the difference between the cloud fractions $f_{ci}$ in the UV band and the cloud fraction $f_{cj}$ in the NIR band. The source of the cloud fractions is not indicated. Up to this pont in the article only a cloud fraction in the UV that is derived by OCRA has been described. Please indicate the source for these cloud fractions, especially the source for $f_{cj}$ in the NIR. If the cloud fractions are derived from the cloud mask information provided by the SNPP data, please indicate how the cloud fractions are calculated. Note specifically that equation 7 applies to the new method, so it does not apply here.

**2.5  Unavailable reference**

On line 116 a reference to (Sneep, 2015) is introduced. This is an unpublished and project-internal document with reference number S5P-KNMI-L2-0129-TN. Either use open literature, or find a way to make this document available to the readers of AMT, for instance by adding it to this article as supplementary material, or arrange for this document to be added to the Sentinel 5P document library.

**2.6  Choice for probably cloudy**

In equation 4 the "probably cloudy" pixels are counted as cloud free. Please elaborate on this choice. The number of pixels in both "probably" classes is small, so the impact is limited, but even that should be mentioned.

**2.7  Optimal Estimation reference**

The authors use (Rodgers, 1976) as a reference for the optimal estimation. This reference points to: Rodgers, C. D.: Retrieval of atmospheric temperature and composition

from remote measurements of thermal radiation, Reviews of Geophysics, 14, 609–624, DOI:10.1029/RG014i004p00609, 1976.

This is perhaps a little peculiar to use as a reference at this point in time, as the same author wrote *the* book on optimal estimation: Rodgers, C. D.: Inverse methods for atmospheric sounding. Theory and practice, volume 2 of Atmospheric, Oceanic and Planetary Physics. World Scientific, 2000. DOI:10.1142/3171.

I strongly suggest that the reference to the book is used instead of the much earlier article.

**2.8 Cloud optical thickness handling**

In lines 185 tot 190 and in equation 5 a transformation is provided to obtain a cloud albedo from a cloud optical thickness. Within the VIIRS mapping software, a similar transformation and its inverse must be used to translate the VIIRS cloud optical thickness into an albedo to be able to take the average, and then the inverse transformation to obtain an effective cloud optical thickness. Please add a statement whether the *same* transformation is used in the VIIRS remapping software (Siddans, 2026).

**2.9 "Missing" three pixel mapping in NIR to UV-VIS scheme**

In section 3.3.2 there is no equivalent to equations 10 – 12 in the UV-VIS to NIR mapping. I would expect such a three pixel equivalent near row 428 because of the symmetry of the binning factors in the Tropomi instrument and the shift between the detectors present in the instrument. A short remark about this absense can resolve the mystery. There is mention of a binning factor change at UV pixel 21, but the same change happens at the other side of the swath. Does that not have an impact, in a way that is similar to the UV-VIS to NIR mapping in section 3.3.1?

On lines 224 – 227 exceptions to the "normal" equations (13 and 14) are mentioned. However, for case (a) there is no explicit mention of *how* this affects the mapping. For clarity I advise to mention that UV/VIS pixel 21 is fully covered by one NIR pixel. The subsequent text makes a lot more sense with that knowledge.

**2.10 Co-registration of cloud albedo and cloud optical thickness**

On lines 240 – 242 the co-registration of cloud albedo and cloud optical thickness is mentioned, and the reader is referred to (Loyola et al., 2023). This is the algorithm theoretical baseline document for the cloud algorithm. In this reference, the only relevant statement to this subject that I can find is the following on page 31 of the reference:

> The basic principle is that the VIIRS and TROPOMI cloud data are interconnected and therefore, each point from the VIIRS dataset can be mapped to the respective TROPOMI point. The adjacent 15 pairs $(H_c^{UV}[i], Z_c^{UV})$, for $i \in [2, 17]$ are used to create the mapping function:
>
> $$Z_c^{UV} = f_{Z_c} \left( H_c^{UV}[i] \right)$$
>
> The mapping function for the cloud top height $f_{Z_c}$ follows a linear regression model for the entire range of the cloud heights. The respective function for the cloud albedo $f_{\omega_c}$ and the cloud optical thickness $f_{\tau_c}$ is a two-way function (i.e., a combination of a linear model with a logarithmic model).

The reference used is insufficient to describe the method used. The statement in the article is essentially repeated without providing substantionally more details. Either complete and update the ATBD, or include an appropriate and reasonably complete description in this article.

**2.11 More appropriate section**

At the start of section 4.1 a listing is provided when the new scheme cannot be used. I think this information should be included in section 3, perhaps in a new sub-section.

**2.12 Comparison results**

I have several remarks about page 17 – 20 that I will group here.

1. In the caption of figure 12 it is not mentioned which day is shown.
2. On line 265 it is mentioned that the scheme can be successfully applied up to a certain latitude.
    (a) Is this a limit of the latitude, or of the solar zenith angle?
    (b) Which of the three reasons that cause failure of the new scheme are a cause of this limitation?
3. On line 267 a sequence of 5 figures is introduced, showing the correlation between the old and the new approach. Which is it obvious after reading the axis labels, please mention that this shows the correlation *after* the transformation.
4. Figures 13 – 17 do not show the units along the axes of the graphs.
5. The axis range of the histograms in figures 13 – 17 is probably too wide given the results, and the number of pixels in the comparison.
6. The vertical axis of the histograms in figures 13 – 17 is not a count but something else, probably a density. Please adjust the labels.
7. Indicate the number of pixels in each comparison, and indicate if the data was selected, for instance for filter out pixels that relied on the old scheme.

**2.13 Not fully consistent terminology**

When discussing the comparison of the cloud top height results on lines 374 – 375, the following statement is made: "The cloud-top heights at the original BD6 were 7600 m and 9400 m at points A and B respectively. After the co-registration at point A, the CTH at BD3 was 6900 m with the new scheme and 6600 m with the old scheme. At point B, the co-registered CTH at BD3 was 9300 m with the new scheme versus 9000 m with the old." If the cloud top height at points A and B is already known, then why is the new co-registration scheme required? Given the location in the swath, there must be two pixels from band 6 contributing to each of the the result in band 3. The values of both contributing factors to both point A and point B should be mentioned for clarity.

The same point applies to table 4 and various other locations throughout the text. In particular the conclusion in lines 442 – 443 should probably be rephrased as well.

**2.14 Caliop results**

I strongly suggest to use the Caliop L1 backscatter intensity as a background for figure 35 – 37, as that provides more context, including – sometimes – the presence of multi-layer clouds. Caliop should also have information on the phase of the cloud top, allowing the authors to indicate where this may play a role.

**2.15 Suggestion for shortening the article**

The information presented in table 3, and the related figures 22 through 25, as well as the text in lines 284 – 312 are mostly descriptive, and the conclusions drawn from them are not very strong; mostly qualitative rather than quantitative. The article probably improves if this part is removed.

Reconsider if appendix A is really required.

**2.16 Appendix B**

Please provide a reference to the method used here. There are several method for calculating a great-circle distance between two points on a globe, with varying accuracy. This appears to be one of the spherical earth methods, rather than Vincenty's formular for an ellipsoid. Any method will be find, they all provide an answer with sufficient accuracy. At the very lease indicate that you are calculating the distance $d$ between the location of the Caliop profile and the points within the selection boxes, and the $R = R_{\text{earth}}$. If the equations in appendix B are to be used without reference, at least provide names to the steps for readability.

**3 Technical corrections**

**3.1 Spectrometer names**

In table 1 on page 2 an overview is provided of the spectrometers in the Tropomi instrument. This table does not make it clear that there are in fact 4 spectrometers on board of the instrument, with 4 detectors (3 CCD's, 1 CMOS). Each of the detectors is split electronically into two halves, leading to a total of 8 bands. But each pair originating from the same detector share the same spatial coverage, and have *no* interband mis-registration. Bands 1 and 2 are special, because the spatial binning is different for band 1 compared to band 2. But the observations in band 2 can still be co-added on ground to a perfect match to band 1.

The suggested solution is to put the spectrometer names centred above the two columns of the bands that they measure, as "UV", "UV/VIS", "NIR", and "SWIR". Note that in the official Tropomi L1B documentation the UV-1 and UV-2 (etc) naming system is not used, see for instance table 1 on page 21 in the L1B ATBD. Using UV/VIS instead of UVIS as the documentation of the Tropomi team does is probably beneficial for clarity here. And of course the references in the text need to be adapted as well.

I believe vertical lines are against the design guidelines of the journal, but they may actually be appropriate here for clarity.

**3.2 Using sun normalized radiances**

In line 55 on page 3 the first instance of "sun-normalized radiances" is encountered. Does the algorithm really use sun-normalized radiances internally, or is the reflectance actually calculated? The difference is a factor $\pi/\mu_0$, and especially look up tables and neural networks typically work better when using reflectances.

**3.3 Ground pixel footprints**

In figure 1 a section of an orbit is shown, to illustrate the spatial mismatch between the two bands involved in the retrieval. Please indicate the flight direction in this figure.

**3.4 The A train**

Line 104: Neither Sentinel 5P, nor Suomi-NPP are part of the A train, see atrain.nasa.gov.

**3.5 Typography**

In equation 4 on page 8 the superscripts are typeset in LaTeX mathematical typesetting mode. This is incorrect, compare the following two versions of the same equation, with the

correct method used on the right:

$$M = \delta_{jk}^{ConfidentlyCloudy} \qquad\qquad M = \delta_{jk}^{\text{ConfidentlyCloudy}}$$

The *amsmath* package is part of the Copernicus document class. This means that the `\text{}` macro is available to typeset fragments that are text within an equation. This improves the readability. It isn't just the italics, but especially the spacing that is different. Please use `\text{}` in all mathematical notation where a superscript or subscript is not an index or exponent but an abbreviation or a name. The symbol $f_c^{\text{NIR}_{\text{old}}}$ can be coded as `$f_c^{\text{NIR}_{\text{old}}}$`.

**3.6 Figure size**

While the schematic overview in figures 3, 5, 6, 8, 9, and 10 is much appreciated in explaining the method, the vertical spacing of these seem rather large. Reducing the vertical spacing of the "source" and "target" panels is likely to provide a better spacing in the article, and allow for easier reference to the figures while reading the description of the method in section 3.3.1, especially for the baseline case (equations 6 and 7).

**3.7 Readability of equation 12**

Suggestion to split the contribution of $\gamma_1$ and $\gamma_2$ to emphasize the symmetry on the equation.

**3.8 Missing introduction of the $H_c$ symbol**

In equation the symbols $H_c^{\text{UV}}$ and $H_c^{\text{NIR}}$ are used, but this symbol is never properly introduced. Extending table 2 to also introduce the symbols for the quantities taken from VIIRS is a possible solution.

**3.9 Language**

I'm not a native speaker, but the following sentences (line 314–316) do not seem to be correct English, or at least a slight rephrasing may make them more readable: "The first TROPOMI UV detector pixels were lacking of cloud height and cloud optical thickness properties up to now due to the missing TROPOMI NIR overlap detector pixels. The use of VIIRS data made possible to re-construct the UV/VIS cloud height information for these pixels through the mapping linear function of Equation 17."

I'll group this under language as well: on lines 324, 325, 327, and 487 the chemical symbol for ozone is given as "O3" rather than "$O_3$".

In the conclusions, on line 421–422, "…from UV/VIS to NIR with the new scheme results to a lower cloud fraction compared to the old scheme …" should probably be "…from UV/VIS to NIR with the new scheme results *in* a lower cloud fraction compared to the old scheme …".

In the conclusions, on line 432–433, "…allows the restructuring of the ROCINN retrieved parameters…", I would suggest "…allows the reconstruction of the ROCINN retrieved parameters…".

In the conclusions, on lines 444–445, "The old co-registration scheme is applied when the old scheme for co-registering the OCRA CF when the VIIRS CF at both UV/VIS and NIR bands are equal to 1.". Please check and rephrase, there is a repeated part in this sentence.

**3.10 Figures 26, 27, 30 through 37, and table 4**

While the difference between figures 26 and 27 is obvious if you can switch between them on a computer screen, if they are on adjacent pages and share the same position on the page, once printed it is nearly impossible to spot the difference between these two figures. This is an issue either for the journal, to make sure that these pages end up in a location where switching between them is possible (as in the current pre-print), or to combine these figures in a three pane single figure, showing both figures and adding a difference plot.

In figure 30 there are 6 pixels that are highlighted. At the printed scale these highlights are hardly visible. Either zoom in further or select another display technique. A zoomed in inset in the top-left of the figure may be an option.

In figures 31 through 37 the locations are displayed only with their longitude *or* their latitude. Please add a second axis with the other coordinate of the data points. Given the importance of the 20% cloud fraction for trace gas retrievals, I suggest to add a thin horizontal line to figures 30 through 33 to indicate that level as a visual guide to see the impact of the new scheme. The 5% level is important for the ROCINN retrieval, and may require an additional grid line in these figures. For clarity I suggest to use a consistent line style for band 6 and a different, but also consistent one for band 3, one continuous, and one dashed for instance. At the moment this is *almost* the case, except for the old band 6 lines.

The same additional coordinate – i.e. latitude – is required in table 4.

In Figure 35 I suggest to plot the "old" and "new" traces in reverse order. Right now the "new" trace is almost completely obscured by the "old" trace. Combined with the intensity of the colours of both traces it is somewhat hard to spot the "new" data. A suggestion specifically for figure 35 is to reduce the vertical scale to 0,-15 km.

**3.11 Caliop**

In section 4.3 on Caliop, there are some confusing statements, please double check them.

1. On line 384 the altitude of Caliop is given as 685 km. On line 385 and 386 Caliop is moved the a *lower* orbit at 688 km. The emphasis is mine.
2. On line 386 it is stated that "Caliop is a two-wavelength…". Unfortunately Caliop is no longer active since August 1st, 2023. Consider to use the past tense.
3. On line 460 the time coverage of version 4.21 of the Caliop processing is listed at up to the present. Please update this with the actual availability.

---

## Referee Comment (RC2)

**Review of "An advanced spatial co-registration of cloud properties for the atmospheric Sentinel missions: Application to TROPOMI"**

Comments based on https://doi.org/10.5194/amt-2024-28 Preprint, retrieved 8 April 2024

**General comments**

Dear authors,

Congratulations to the achieved improvements on your cloud property retrieval algorithm.

While the described improvements in UPAS version 2.6 are important to all users of the respective TROPOMI products, the approach and techniques are interesting for a much wider group of readers. The paper describes in great detail the previous method and the new algorithm. The results are compared extensively to the previous version and also another instrument.

Although the detailed comparison is useful to understand the impact of the changes depending on the cloud situation, I think that the paper would be easier to read when some parts could be shortened and some plots would be combined. Specifically for the correlation plots and histograms it is not clear why they are shown or what the reader should learn from them. A qualitative description and an interpretation in the context of the problem would be helpful. The grouping in the categories A-K is also difficult to follow for me, the respective parameters between the groups do not show a clear correlation. Please consider discussion the observations more along the groups and combining the barplots.

In the following I list my detailed comments/questions in a table referring to page and line number of the version I retrieved on the 8th of April. In separate sections I have also listed some remarks on typos/definitions/phrasing and suggestions for the figures.

**Detailed comments**

Please find detailed comments in the table below.

| # | Page | Line | Section | Comment |
|---|------|------|---------|---------|
| C1 | 1 | | abstract | Consider shortening the abstract and moving part of the content to the introduction |
| C2 | 1 | 2 | abstract | The use of the word "frequently" suggests a temporal dependence, although the misregistration is a design feature and should be constant with time. |
| C3 | 1 | 5 | abstract | Unclear: is it the UV band or the UVIS band or both of them? (see also points below) |
| C4 | 1 | 16 | abstract | Unclear: what is meant by the first detector pixel? I think you want to refer to the outermost groundpixel of a scanline. The first detector pixel would refer to (unbinned) detector pixels at the edge of detector which is not used for L2 retrievals. |

| # | Page | Line | Section | Comment |
|---|---|---|---|---|
| C5 | 2 | 30 | Intro/Table 2 | Four distinct spectrometers: Table 2 lists 8 spectrometers, this is not correct. Please be very clear in your distinction between bands and spectrometers. Also ensure that your naming is consistent, which spectrometer do you mean by UV/VIS band from the abstract? I suggest you use the convention form Veefkind et al. 2012: UV, UVIS, NIR and SWIR and refer to bands 1-8 if you want to refer to a spectral range of TROPOMI. |
| C6 | 2 | 32 | Intro | Different spectrometers/several bands/inter-band: Misalignment between groundpixels also occurs within one spectrometer, although it is much smaller than between the spectrometers (see (and cite?) https://amt.copernicus.org/articles/11/6439/2018/amt-11-6439-2018.html) . The phrasing that the misregistration is caused by having different spectrometers is therefore not accurate. Do you want to point out the misregistration between bands or between spectrometers? If you use the term bands both intra- and inter-band co-registration can be meant. |
| C7 | 2 | 34 | Intro | 'Interconnected to the spatial resolution'. What do you mean by this? If you refer to the changes in along-track resolution but later (p5 l99) dismiss the along-track impact, why refer to the along track resolution? The ground-pixel size is determined by the binning and along-track co-addition, see https://sentinel.esa.int/documents/247904/2476257/Sentinel-5P-TROPOMI-Level-1B-ATBD p 165, the mis-alignment is an optical pointing effect, it is present for all ground-pixel sizes. |
| C8 | 3 | 50 | 2 | UV/VIS: which TROPOMI bands/spectrometers do you refer to hear, see also comment above |
| C9 | 3 | 68/69 | 2 | The definition for $\lambda$ is missing. |
| C10 | 3 | 72 | 2 | It's great to have the table with the definitions, maybe you could extend it to add all parameters and then shorten/delete the description before the formulas? |
| C11 | 4 | 80 | 3 | Among others : add references to the other operational cloud products. |
| C12 | 4 | 88-92 | 3 | Unclear: do you mean detector pixel size? It might be clearer if you refer to the ground pixels and nadir vs edges of the swath |
| C13 | 4 | 90 | 3 | Instrument nadir angle: this phrasing is confusing. The effect is caused by a combination of instrument features (due to the large swath angle) and the Earth's curvature. |
| C14 | 4 | 92 | 3 | Minimum dispersion: Do you mean the increase of the size of the ground pixels towards the edge of the swath? Dispersion is generally related to wavelength. Suggestion: with the aim to minimize the difference in ground pixel size in across-track direction. |
| C15 | 4 | 94 | 3 | … this results in a ground pixel size of 15 km : this is phrased confusingly, the reduction of the binning factor doesn't increase the ground pixel size ... the binning is reduced to keep the groundpixel size at a reasonable value, it increases towards the edge of the swath due to optical limitations of the instrument and the curvature of the Earth. |

| # | Page | Line | Section | Comment |
|---|------|------|---------|---------|
| C16 | 6 | 109 | 3.1 | Do you calculate the mis-registration between bd 3 and bd6 AND bd4 and bd6? |
| C17 | 7 | Fig2 | | What is the difference between the upper and lower panel? They look almost identical. If they are, remove one of the two, if not provide a plot where the differences are clearly visible. |
| C18 | 8 | 163 | 3.3 | Eq. 4: Is this a normalization i.e. is the sum of all $\delta_{jk}^X$ = equal to the sum of all pixels? |
| C19 | 8 | 174ff | 3.3 | There is a lot of description on the VIIRS data retrieval algorithm, is this needed for the paper? Consider shortening the text/replacing it by a suitable reference. |
| C20 | 10 | 209 | 3.3.1 | The change in binning factor occurs twice in every scanline, or do you make a difference between 1->2 and 2->1 ?If yes, please add |
| C21 | | Fig3, Fig 5, Fig 6 Fig8-10 | | The images are quite large for their content. The axis(arrows) are not needed, or what is meant by the y and x axis? Suggestion: combine the three situations in one figure with 3 panels, then the labels only need to be shown once and the situations can be compared easily. Please explain the symbol used in the illustration. Otherwise it's not clear what differences the reader should see. What is the conceptual difference between Figs3-6 and Figs 8-10? It's only the source and target exchanged and other properties considered, or not? Why separate figures then? I do understand the case of 1 and 2 pixels contributing to the target, but when is it 3? Can you show this in figure 4 or 7? Or is it a along track overlap? The figures suggest only an overlap in across-track. Or does this happen when the binning changes in one of the spectrometers but not the other? This is not clear in the text. |
| C22 | | Fig 4, Fig 7 | | There are no horizontal magenta stripes visible. If they overlap with band 6 this should be indicated in the caption. Please combine Fig 4 and 7, then it's also clearer that the pixels are shifted about ½ pixel wrt each other and that it's not the black and magenta overlapping. |
| C23 | 12 | 221 | 3.3.2 | Define h |
| C24 | 12 | 224 | 3.3.2 | How are the pixels numbered 0 to ? From West to East? |
| C25 | 13 | 236 | 3.3.2 | Alpha and beta are not described/defined |
| C26 | 15/16 | 252ff | 4.1 | You mean if the values in the pixels are all the same for VIIRS but not for TROPOMI? This is not phrased very clearly. |
| C27 | 16 | Fig 11 | | What are we supposed to see in this plot? Or is it for illustration only? Please add this to the caption. What are the pixels without frames? Re-gridded VIIRS data? Would those pictures not be better suited to be included in the figures Fig3, Fig 5, Fig 6, Fig8-10? |
| C28 | 17 | Fig 17 | | I do not see a difference between the plots cloud-top height, optical thickness and cloud albedo. Is there one? If not you could consider omitting part of the plots from the paper/moving them to the appendix. Why are you showing the plots? To indicate there's no apparent dependence on where on Earth it is? Then please mention this in the caption. |

| # | Page | Line | Section | Comment |
|---|------|------|---------|---------|
| C29 | 17 | 265 | 4.1.1 | Why does it only work to a certain latitude? |
| C30 | | Figs 14-17, 267-275 | 4.1.1 | What do you want to show with all the scatter plots/histograms? They do not seem to add to the paper. Can the information be summarized/condensed? An interpretation given? If the plots are needed please consider to group them together and decrease the size (of at least the histograms). The text is descriptive, but is there also an interpretation/understanding of the differences? It is not very clear what the conclusion is. |
| C31 | 18 | 277 | 4.1.1 | It is expected: did you check that this is indeed true? In Fig 18 you show this. To make it clearer, please rephrase: The differences are exactly zero… |
| C32 | 19 | 284 ff Table 3 | 4.1.1 | It is a good idea to group the differences in different categories. Could you please also add the cloudiness situation in the table? And maybe add (verbally) the different parameters as an extra row? Why did you chose 10 different categories? The discussion on the different groups is very difficult to follow and the conclusion is not clear to me. It seems to be bins for the different parameters, but is there a relation for the lowest and highest bins? |
| C33 | | Figs.18-21 | | Are all these plots needed to support your conclusions? Please consider omitting some plots or moving them to the appendix. |
| C34 | | Figs 22-25 | | Are all these plots needed to support your conclusions? Please consider omitting some plots or moving them to the appendix. The axis labels have a too low resolutions. What is the difference between the days? Why can't they be combined? And again why the different groups? It is essentially a bin. Would it not be clearer to add the bin range to the plot axis instead of introducing the groups? I cannot follow what I should learn from these plots. What is the conclusion? |
| C35 | 24 | 317 | 4.1.2 | "closing the gaps" this phrasing is a bit confusing, the gap will only be reduced by one groundpixel and not closed. It should maybe be mentioned why the gap is there in the first place (and that it can't be reduced with the current orbit parameters) |
| C36 | 24 | 314 | 4.1.2 | 'first TROPOMI UV detector pixels'. What is meant by this? The western-most pixel? Please re-phrase, see also above. |
| C37 | 27ff | | 4.2 | In this section individuals cases are discussed. Are the results individual observations or can they be also be supported by statistics? How do these observations connect to the barplots? It would be helpful to make a connection here. |
| C38 | 29/30 | Fig. 26/27 | | The figures do not add much to the understanding. The point to demonstrate is only visible when clicking back and forth between the figures. Suggestion: Combine both figures, zoom into the plot from 144E to about 152E and highlight the additional available pixels. |
| C39 | 32 | Fig 30 | | Consider zooming in on the relevant part and reducing the size of the figure. |
| C40 | 34 | 380-390 | 4.3 | Not all the information on Calipso is needed to understand the paper, please consider to shorten this section. |

| # | Page | Line | Section | Comment |
|---|------|------|---------|---------|
| C41 | 35 | 396-398 | 4.3 | Forward model of TROPOMI: please be specific on the model you refer to. |
| C42 | 36 | 410 | 5 | Please refer to the algorithm name, as there are more TROPOMI cloud property schemes. |
| C43 | 36-39 | | 5 | The conclusion part is very clear and concise. In the detailed discussions in earlier sections it would be good to already mention these conclusions. |

**Technical comments/typos**

**General:**

Capitalization of acronyms: some acronyms are capitalized and some not when written out, please be consistent and use only one of the two.

Please be consistent with your spelling, there are inconsistencies, for example:

- sunglint sun-glint
- cloud-top cloud top
- earth Earth

**Detailed technical comments:**

| # | Page | Line | Section | Comment |
|---|------|------|---------|---------|
| T1 | 2 | 29 | Intro | Window -> windows |
| T2 | 2 | 38/39 | Intro | Have a big heritage .. which have already been applied: this sentence doesn't read well, skip 'have .. missions' |
| T3 | 3 | 75 | 2 | Big Data -> big data volume |
| T4 | 4 | 84 | 3 | On top -> in addition to |
| T5 | 5 | Fig 1 | | "descibed"→ described, also here: VIS-1/NIR-2 spectrometer is not correct |
| T6 | 5 | 106 | 3 | UV/VIS ? What is meant here? UVIS? |
| T7 | 6 | 113 | 3.1 | UV bands: what is meant here? Bands 1 & 2? |
| T8 | 6 | 135 | 3.1 | Remove 'from' or change 'showed that'  'can be seen that' |
| T9 | 8 | 159 | 3.3 | Well applied? Do you mean extensively? Successfully? |
| T10 | 9 | 192 | 3.3.1 | UV/VIS ? What is meant here? UVIS? |
| T11 | 9 | 195 | 3.3.1 | Add Y after weight |
| T12 | 10 | 211/212 | 3.3.1 | Add Y1 and Y2 in the sentence |
| T13 | 15 | 251 | 4.1 | Value -> values |
| T14 | 15 | 252 | 4.1 | Results to -> results in |
| T15 | 16 | 259 | 4.1.1 | From one day to the other -> from day to day |
| T16 | 17 | 269 | 4.1.1 | Not that extended -> there is less scatter than above … |
| T17 | 34 | 393 | 4.3 | North-> northern |
| T18 | 36 | 413 | 5 | Do you mean the 'following' instead of "aforementioned?" |
| T19 | 37 | 434 | 5 | Columnar -> column |
| T20 | 38 | 438/447 | 5 | Remove 'exercise' |
| T21 | 38 | 444 | 5 | There is something wrong in this sentence, please re-phrase. |
| T22 | 39 | 452 | 5 | Later ,… -> Once S-4 has been launched, … |
| T23 | 40 | 468 | App B | Rectangular -> rectangle |

| # | Page | Line | Section | Comment |
|-----|------|------|---------|---------|
| T24 | 42 | 483 | | Unclear: One or more than one? |

**References:**

The reference Sneep, 2015 is not available online it seems.

**Figures:**

On a more technical side: For people with red-green blindness or colour blind people the chosen blue, green and red hues do not work very well. Please consider to adapt your colour scheme.

---

## Community Comment (CC1)

**Replies to RC1 – from 28th April**

Dear Referee,

We would like to thank you very much for your constructive comments, which were helping us to understand the weaknesses of this article. We provide our replies below and we are positive that the revised version of the article will ensure that any inconsistencies are removed, the presentation/interpretation of the results and readability are improved.

**General comments**

We agree that the article should become shorter and we will follow the referee's suggestions in this effort.

The mapping equations are introduced without much justification as to why one would want to use these specific quantities from VIIRS te aid in the transformations. That you use a derived fractional cloud cover from VIIRS to map the cloud fraction from UV-VIS to NIR is fairly logical, but why not use the $\gamma$ factors derived from the VIIRS cloud fraction as well as the dimensionless scaling factor to aid in the mapping of the cloud height from NIR to UV-VIS? After all, you do get a weight that is related to the amount of clouds in the pixel, and therefore a means to obtain a weighted average.

We have never tried to use the $\gamma$ factors derived from the VIIRS cloud fraction for the co-registration of cloud top height. One main issue that we would have in this approach, is that in fully cloudy scenes the denominator becomes 0 (see Equation 6) and thus we cannot apply the new scheme. When we use the cloud top height property and Equation 13, it is very unlikely that the cloud top height values of the contributing pixels are identical and therefore we can exclude the aforementioned numerical errors.

This brings me to my second more important point. Using the cloud fractions would preclude extrapolation to row 0, but at the same time, how much of the information in row 0 is actually coming from the Tropomi instrument? A discussion on the effective contributions of the information from Tropomi and VIIRS in the final product is desired. Is the information from row 0 really VIIRS information that has been scaled locally by the ratio of Tropomi and VIIRS cloud heights to take into account the systematic model difference between the two instruments and retrieval methods? This discussion really needs to be added to the article before publication.

We would like to provide clarity on how the additional cloud information in row 0 is constructed. We take 15 adjacent (VIIRS, TROPOMI) points in each scanline (located at West part of the swath to eliminate any geometry dependencies in both instruments) and we find the mapping function (e.g., for cloud top height is a linear regression model) between from VIIRS to TROPOMI. After having such a mapping function, we use the first westernmost VIIRS cloud top height to calculate the corresponding value for TROPOMI. Of course, since this is at a certain point an extrapolation using a regression model we cannot consider that the values on those first ground pixels are as accurate as the rest of the pixels, for which L1B radiances are available. Because the information from TROPOMI in row 0 is dependent on how well the VIIRS/TROPOMI mapping function has been constructed in each scanline. Nevertheless, we have reflected this "limitation" in our QA value scheme by adding a penalty for the first row (please refer to L2 CLOUD ATBD). All in all, we have enough evidence that the additional values in the first row are beneficial, in particular since those values have replaced fill values and to our view, a less accurate cloud information is better than no information. Please refer to figure 11 (where we visually see that the CTH TROPOMI NIR (original parameter) does not contain values in the first magenta ground pixel). If we compare the co-registered CTH TROPOMI UV/VIS and original CTH TROPOMI NIR maps, we can be confident that the new co-registration scheme does not introduce inconsistencies or outliers also in the first row. If we visually compare the CTH TROPOMI UV/VIS and CTH VIIRS UV/VIS, we observe that the cloud top height values can be different in absolute numbers. But again, those differences do not alter the original cloud structures.

**Specific comments**

**[2.1 Spatial resolution]** In the introduction, lines 33 to 37, the spatial resolution of Tropomi is discussed. For the bands used by OCRA/ROCINN the spatial resolution in the flight direction is unimportant. Because of the shared entrance slit between the VIS and NIR spectrometers (See L1B ATBD, section 7.2), there is no spatial misalignment between those spectrometers in the flight direction. That also means that the change of the spatial resolution in the flight direction is unimportant for the subject of this article. I suggest to shorten this discussion in the introduction.

We agree with this suggestion.

**[2.2 Algorithm description]** In section 2 of the article a description of the cloud retrieval algorithm is given. The algorithm is based on a pair of algorithms, OCRA for the cloud fraction, and ROCINN for the cloud top height and cloud albedo or cloud optical thickness, depending on the cloud model. Please explain how a cloud fraction can be derived from OCRA without assuming a cloud albedo, or put differently, show that the three parameters are independent, and the cloud albedo (or optical thickness) isn't (implicitly) assumed already in the cloud fraction derivation in OCRA.

An explicit cloud albedo assumption is not made by OCRA. The main assumption of the OCRA algorithm is a wavelength independency of the reflectance of a cloudy spectrum over the considered wavelength ranges, i.e. that the reflectance for a fully cloudy pixel is equal for all considered OCRA colors, resulting in a "white" scene when the reflectances are transferred to color space. Since OCRA provides a radiometric cloud fraction and not a geometric cloud fraction, one could argue that cloud albedo (or cloud optical thickness) has an implicit impact on the retrieved radiometric cloud fraction, because an optically thick (or high albedo) cloud will be easier to be detected due to its higher contrast with the surface background than an optically thin (or low albedo) cloud. However, if the reflectances of a cloudy scene are equal in all considered wavelength ranges, OCRA will report a high cloud fraction regardless if the cloud is optically thick or optically thin. We will expand in the revised manuscript on this basic OCRA assumption of wavelength independency of cloud reflectance.

**[2.3 Spatial alignment in the flight direction]** In section 3, on lines 97 to 99, the authors mention that the static lookup tables used up to now indicate a very small mis-match in the flight directions. These are rounding errors in the algorithm used to calculate these tables. The design of the instrument itself (See L1B ATBD, section 7.2) guarantees that there is no spatial mismatch in the flight direction between the VIS and NIR spectrometers (bands 3 – 6). So the algorithm does not neglect anything, but it handles the situation correctly. Please adjust the text accordingly.

We will rephrase the text so that we reflect that there is no spatial mismatch in the flight direction between UVIS and NIR bands.

**[2.4 Previous treatment of the spatial mis-registration]** In section 3.1 the previous treatment of the mis-registration is described. Here a "cloud co-registration inhomogeneity parameter" is introduced. This parameter involves the difference between the cloud fractions *fci* in the UV band and the cloud fraction *fcj* in the NIR band. The source of the cloud fractions is not indicated. Up to this pont in the article only a cloud fraction in the UV that is derived by OCRA has been described. Please indicate the source for these cloud fractions, especially the source for *fcj* in the NIR. If the cloud fractions are derived from the cloud mask information provided by the SNPP data, please indicate how the cloud fractions are calculated. Note specifically that equation 7 applies to the new method, so it does not apply here.

The cloud co-registration inhomogeneity parameter is a quantity that we have introduced to the L2 CLOUD product to support the identification of areas with cloudiness heterogeneity. The fci in the UV band is the OCRA cloud fraction and fcj is the co-registered fcj in the NIR using the static KNMI LUTs. The SNPP data are not used here.

**[2.5 Unavailable reference]** On line 116 a reference to (Sneep, 2015) is introduced. This is an unpublished and projectinternal document with reference number S5P-KNMI-L2-0129-TN. Either use open literature, or find a way to make this document available to the readers of AMT, for instance by adding it to this article as supplementary material, or arrange for this document to be added to the Sentinel 5P document library.

Many thanks for highlighting this important information to us. We would approach ESA and the authors of this document to add it in the Sentinel 5P document library.

**[2.6 Choice for probably cloudy]** In equation 4 the "probably cloudy" pixels are counted as cloud free. Please elaborate on this choice. The number of pixels in both "probably" classes is small, so the impact is limited, but even that should be mentioned.

We have tried both approaches and we calculated the cloud fraction using only "confidently cloudy" at the nominator and compared to the cloud fraction when using both "confidently cloudy" and "probably cloudy". We found that the difference is minor and we decided to use the first definition.

**[2.7 Optimal Estimation reference]** The authors use (Rodgers, 1976) as a reference for the optimal estimation. This reference points to: Rodgers, C. D.: Retrieval of atmospheric temperature and composition from remote measurements of thermal radiation, Reviews of Geophysics, 14, 609–624, DOI:10.1029/RG014i004p00609, 1976. This is perhaps a little peculiar to use as a reference at this point in time, as the same author wrote the book on optimal estimation: Rodgers, C. D.: Inverse methods for atmospheric sounding. Theory and practice, volume 2 of Atmospheric, Oceanic and Planetary Physics. World Scientific, 2000. DOI:10.1142/3171. I strongly suggest that the reference to the book is used instead of the much earlier article.

We will replace the reference on the article with the suggested book reference.

**[2.8 Cloud optical thickness handling]** In lines 185 to 190 and in equation 5 a transformation is provided to obtain a cloud albedo from a cloud optical thickness. Within the VIIRS mapping software, a similar transformation and its inverse must be used to translate the VIIRS cloud optical thickness into an albedo to be able to take the average, and then the inverse transformation to obtain an effective cloud optical thickness. Please add a statement whether the same transformation is used in the VIIRS remapping software (Siddans, 2026).

To the best of our knowledge, the transformation in the VIIRS mapping software uses the same conversion formula. However, we will need to cross check it with Dr. Richard Siddans. We will add a statement in the revised manuscript whether the same transformation is used in the VIIRS remapping software.

**[2.9 "Missing" three pixel mapping in NIR to UV-VIS scheme]** In section 3.3.2 there is no equivalent to equations 10 – 12 in the UV-VIS to NIR mapping. I would expect such a three pixel equivalent near row 428 because of the symmetry of the binning factors in the Tropomi instrument and the shift between the detectors present in the instrument. A short remark about this absense can resolve the mystery. There is mention of a binning factor change at UV pixel 21, but the same change happens at the other side of the swath. Does that not have an impact, in a way that is similar to the UV-VIS to NIR mapping in section 3.3.1? On lines 224 – 227 exceptions to the "normal" equations (13 and 14) are mentioned. However, for case (a) there is no explicit mention of how this affects the mapping. For clarity I advise to mention that UV/VIS pixel 21 is fully covered by one NIR pixel. The subsequent text makes a lot more sense with that knowledge.

Even if the binning factor change happens at the east part of the swath too, we observe (by looking to the upper panel of figure 2) that the "special" case of having three BD3 source pixels contributing to the BD6 target pixel occurs only at pixel #19. We can also visually understand this "special" case if we look in figure 1: the red 19th BD6 footprint has an overlap with the three blue 20th, 21st and 22nd BD3 footprints. We will make clear this point in the revised text.

We will add the clarification that UVIS pixel 21 is fully covered by one NIR pixel.

**[2.10 Co-registration of cloud albedo and cloud optical thickness]** On lines 240 – 242 the co-registration of cloud albedo and cloud optical thickness is mentioned, and the reader is referred to (Loyola et al., 2023). This is the algorithm theoretical baseline document for the cloud algorithm. In this reference, the only relevant statement to this subject that I can find is the following on page 31 of the reference: The basic principle is that the VIIRS and TROPOMI cloud data are interconnected and therefore, each point from the VIIRS dataset can be mapped to the respective TROPOMI point. The adjacent 15 pairs (HUV c [i]; ZUV c ), for i 2 [2; 17] are used to create the mapping function: ZUV c = fZc HUV c [i] The mapping function for the cloud top height $fZc$ follows a linear regression model for the entire range of the cloud heights. The respective function for the cloud albedo $f!c$ and the cloud optical thickness $fτc$ is a two-way function (i.e., a combination of a linear model with a logarithmic model). The reference used is insufficient to describe the method used. The statement in the article is essentially repeated without providing substantionally more details. Either complete and update the ATBD, or include an appropriate and reasonably complete description in this article.

We will update the ATBD with a more detailed description. We would prefer not to load this article with additional information.

**[2.11 More appropriate section]** At the start of section 4.1 a listing is provided when the new scheme cannot be used. I think this information should be included in section 3, perhaps in a new sub-section.
We have preferred to place the non-applicability of the scheme in Section 4 and not in previous section 3 because it is only relevant for TROPOMI instrument, whereas the presented methodology in Section 3 is generic and applicable for more Sentinel missions.

**[2.12 Comparison results]** I have several remarks about page 17 – 20 that I will group here.
1. In the caption of figure 12 it is not mentioned which day is shown.
We will add the information on the day in the figure caption.
2. On line 265 it is mentioned that the scheme can be successfully applied up to a certain latitude.
(a) Is this a limit of the latitude, or of the solar zenith angle?
(b) Which of the three reasons that cause failure of the new scheme are a cause of this limitation?
This is a limitation simply due to the lack of VIIRS cloud optical thickness data in those latitudes. The VIIRS cloud optical thickness originates from the CloudDCOMP (Daytime Cloud Optical and Microphysical Properties) EDR (see Sention 3.3.), whereas the VIIRS cloud fraction and cloud top height originate from other EDRs where valid data points appear in high latitudes.

3. On line 267 a sequence of 5 figures is introduced, showing the correlation between the old and the new approach. Which is it obvious after reading the axis labels, please mention that this shows the correlation after the transformation.
We will mention that the figures show the correlation after the transformation.
4. Figures 13 – 17 do not show the units along the axes of the graphs.
We will include the units in the axes of those figures.
5. The axis range of the histograms in figures 13 – 17 is probably too wide given the results, and the number of pixels in the comparison.
We will adjust the x-axis range of the histograms in those figures.
6. The vertical axis of the histograms in figures 13 – 17 is not a count but something else, probably a density. Please adjust the labels.
We will adjust the labels to reflect that the histogram represents the probability density function of the data rather than the raw counts.
7. Indicate the number of pixels in each comparison, and indicate if the data was selected, for instance for filter out pixels that relied on the old scheme.

We will include the number of pixels which are used for the comparisons. The new scheme is applied on top of the old and therefore, we have decided not to apply any filtering on the method. However, the comparison is only made if both "old" and "new" data contain meaningful values. For instance, values corresponding to the first pixel of each scanline are not included in the plots since the "old" data contain fill values there.

**[2.13 Not fully consistent terminology]** When discussing the comparison of the cloud top height results on lines 374 – 375, the following statement is made: "The cloud-top heights at the original BD6 were 7600 m and 9400 m at points A and B respectively. After the co-registration at point A, the CTH at BD3 was 6900 m with the new scheme and 6600 m with the old scheme. At point B, the co-registered CTH at BD3 was 9300 m with the new scheme versus 9000 m with the old." If the cloud top height at points A and B is already known, then why is the new co-registration scheme required? Given the location in the swath, there must be two pixels from band 6 contributing to each of the the result in band 3. The values of both contributing factors to both point A and point B should be mentioned for clarity. The same point applies to table 4 and various other locations throughout the text. In particular the conclusion in lines 442 – 443 should probably be rephrased as well.

We will rephrase the statements on the mentioned lines and add more information in table 4 to exclude this inconsistency that the referee found in the article.

**[2.14 Caliop results]** I strongly suggest to use the Caliop L1 backscatter intensity as a background for figure 35 – 37, as that provides more context, including – sometimes – the presence of multi-layer clouds. Caliop should also have information on the phase of the cloud top, allowing the authors to indicate where this may play a role.

The information on the cloud phase and the presence of multi-layer clouds is also available on the L2 cloud layer "CAL_LID_L2_05kmCLay-Standard-V4-20" product. Thus, we do not need to use the Caliop L1 backscatter intensity for this task.

**[2.15 Suggestion for shortening the article]** The information presented in table 3, and the related figures 22 through 25, as well as the text in lines 284 – 312 are mostly descriptive, and the conclusions drawn from them are not very strong; mostly qualitative rather than quantitative. The article probably improves if this part is removed. Reconsider if appendix A is really required.

In our efforts to shorten the length of the article we will consider referee's suggestions whether (a) Table 3 and figures 22-25 are very necessary and (b) appendix A should be removed.

**[2.16 Appendix B]** Please provide a reference to the method used here. There are several method for calculating a great-circle distance between two points on a globe, with varying accuracy. This appears to be one of the spherical earth methods, rather than Vincenty's formular for an ellipsoid. Any method will be find, they all provide an answer with sufficient accuracy. At the very lease indicate that you are calculating the distance $d$ between the location of the Caliop profile and the points within the selection boxes, and the $R = R_{earth}$. If the equations in appendix B are to be used without reference, at least provide names to the steps for readability.

We do not have a reference included in the article since we have constructed our own approach for the spatial colocation between TROPOMI and CALIPSO data. This is the reason why we have an appendix for the description and not a literature reference. The distance "d" (i.e., given by Equation B4) between the Caliop profile and the middle points for TROPOMI footprints are calculated in a predefined spatial window. Then, the TROPOMI footprint with the smallest distance is selected for the cloud height comparisons. We will provide more information of the calculation steps in Appendix B.

**Technical corrections**

**[3.1 Spectrometer names]** In table 1 on page 2 an overview is provided of the spectrometers in the Tropomi instrument. This table does not make it clear that there are in fact 4 spectrometers on board of the instrument,

with 4 detectors (3 CCD's, 1 CMOS). Each of the detectors is split electronically into two halves, leading to a total of 8 bands. But each pair originating from the same detector share the same spatial coverage, and have *no* interband mis-registration. Bands 1 and 2 are special, because the spatial binning is different for band 1 compared to band 2. But the observations in band 2 can still be co-added on ground to a perfect match to band 1. The suggested solution is to put the spectrometer names centred above the two columns of the bands that they measure, as "UV", "UV/VIS", "NIR", and "SWIR". Note that in the official Tropomi L1B documentation the UV-1 and UV-2 (etc) naming system is not used, see for instance table 1 on page 21 in the L1B ATBD. Using UV/VIS instead of UVIS as the documentation of the Tropomi team does is probably beneficial for clarity here. And of course the references in the text need to be adapted as well. I believe vertical lines are against the design guidelines of the journal, but they may actually be appropriate here for clarity.

We will modify table 1 so that it becomes clear that TROPOMI payload consists of 4 spectrometers (not 8). In the revised version, when we refer to spectral range we will use "UV/VIS" and when we refer to TROPOMI spectrometer we will use "UVIS" according to the L1B documentation.
Moreover, we will remove the vertical lines to cope with the journal guidelines.

**[3.2 Using sun normalized radiances]** In line 55 on page 3 the first instance of "sun-normalized radiances" is encountered. Does the algorithm really use sun-normalized radiances internally, or is the reflectance actually calculated? The difference is a factor $\pi = \mu_0$, and especially look up tables and neural networks typically work better when using reflectances.

The NN is trained with sun-normalized radiances.

**[3.3 Ground pixel footprints]** In figure 1 a section of an orbit is shown, to illustrate the spatial mismatch between the two bands involved in the retrieval. Please indicate the flight direction in this figure.

We will adjust figure 1 such that the flight direction is shown.

**[3.4 The A train]** Line 104: Neither Sentinel 5P, nor Suomi-NPP are part of the A train, see atrain.nasa.gov
We will rephrase the sentence so that the reference to A-train is removed. The sentence will become: "This facilitates the so-called loose formation operation with the SNPP spacecraft, with only 3 to 5 minutes time difference from S5P".

**[3.5 Typography]** In equation 4 on page 8 the superscripts are typeset in LaTeX mathematical typesetting mode. This is incorrect, compare the following two versions of the same equation, with the correct method used on the right: $M = \delta_{Conf\,idently Cloudy\,jk}$ $M = \delta_{\text{ConfidentlyCloudy}jk}$ The *amsmath* package is part of the Copernicus document class. This means that the \text{} macro is available to typeset fragments that are text within an equation. This improves the readability. It isn't just the italics, but especially the spacing that is different. Please use \text{} in all mathematical notation where a superscript or subscript is not an index or exponent but an abbreviation or a name. The symbol $f_{c\text{NIR}_{\text{old}}}$ can be coded as

        $f_c^{\text{NIR}_{\text{old}}}$.

We will use the \text{} command for equation 4, as suggested by the referee.

**[3.6 Figure size]** While the schematic overview in figures 3, 5, 6, 8, 9, and 10 is much appreciated in explaining the method, the vertical spacing of these seem rather large. Reducing the vertical spacing of the "source" and "target" panels is likely to provide a better spacing in the article, and allow for easier reference to the figures while reading the description of the method in section 3.3.1, especially for the baseline case (equations 6 and 7).
We will apply a reduction in the vertical spacing of the "source" and "target" panels for the mentioned figures.

**[3.7 Readability of equation 12]** Suggestion to split the contribution of $\gamma_1$ and $\gamma_2$ to emphasize the symmetry on the equation.
We will apply this suggestion.

**[3.8 Missing introduction of the Hc symbol]** In equation the symbols $H_{UV\,c}$ and $H_{NIR\,c}$ are used, but this symbol is never properly introduced. Extending table 2 to also introduce the symbols for the quantities taken from VIIRS is a possible solution.

*Extending table 2 with including those symbols seems not an optimal solution for us since Table 2 contains cloud parameters which are retrieved from the OCRA/ROCINN algorithms. We will introduce the symbols properly in Section 3.3.*

**[3.9 Language]** I'm not a native speaker, but the following sentences (line 314 – 316) do not seem to be correct English, or at least a slight rephrasing may make them more readable: "The first TROPOMI UV detector pixels were lacking of cloud height and cloud optical thickness properties up to now due to the missing TROPOMI NIR overlap detector pixels. The use of VIIRS data made possible to re-construct the UV/VIS cloud height information for these pixels through the mapping linear function of Equation 17."

*We will rephrase the sentences as: "The first (westernmost) TROPOMI ground pixels in the UVIS grid were lacking any information about the cloud height and cloud optical thickness properties up to now. The use of VIIRS data made possible the reconstruction of cloud information for these pixels in the UVIS grid through the mapping linear function of Equation 17."*

I'll group this under language as well: on lines 324, 325, 327, and 487 the chemical symbol for ozone is given as "O3" rather than "$O_3$".

*We will adjust the ozone symbol.*

In the conclusions, on line 421 – 422, ". . . from UV/VIS to NIR with the new scheme results to a lower cloud fraction compared to the old scheme . . . " should probably be ". . . from UV/VIS to NIR with the new scheme results *in* a lower cloud fraction compared to the old scheme . . . ".

*We will apply the suggestion.*

In the conclusions, on line 432 – 433, ". . . allows the restructuring of the ROCINN retrieved parameters. . . ", I would suggest ". . . allows the reconstruction of the ROCINN retrieved parameters. . . ".

*We will apply the suggestion.*

In the conclusions, on lines 444 – 445, "The old co-registration scheme is applied when the old scheme for co-registering the OCRA CF when the VIIRS CF at both UV/VIS and NIR bands are equal to 1.". Please check and rephrase, there is a repeated part in this sentence.

*The sentence should be: "The old scheme is applied for co-registering the OCRA CF when the VIIRS CF at both UV/VIS and NIR bands are equal to 1."*

**[3.10 Figures 26, 27, 30 through 37, and table 4]** While the difference between figures 26 and 27 is obvious if you can switch between them on a computer screen, if they are on adjacent pages and share the same position on the page, once printed it is nearly impossible to spot the difference between these two figures. This is an issue either for the journal, to make sure that these pages end up in a location where switching between them is possible (as in the current pre-print), or to combine these figures in a three pane single figure, showing both figures and adding a difference plot.

*Adding a difference plot with actual values is not possible since there are no valid data for the first ground pixels in the old version. But we can combine both figures, zoom into the plot around 144E - 152E and highlight the additional available pixels.*

In figure 30 there are 6 pixels that are highlighted. At the printed scale these highlights are hardly visible. Either zoom in further or select another display technique. A zoomed in inset in the top-left of the figure may be an option.

*We will adjust figure 30 by zooming in the region of interest.*

In figures 31 through 37 the locations are displayed only with their longitude *or* their latitude. Please add a second axis with the other coordinate of the data points.

*We will add the second axis with the other coordinate as suggested.*

Given the importance of the 20 % cloud fraction for trace gas retrievals, I suggest to add a thin horizontal line to figures 30 through 33 to indicate that level as a visual guide to see the impact of the new scheme. The 5 % level is important for the ROCINN retrieval, and may require an additional grid line in these figures.

*We will add the two suggested horizontal lines (i.e., one at 20% cloud fraction and a second at 5% cloud fraction) in figures 30-33.*

For clarity I suggest to use a consistent line style for band 6 and a different, but also consistent one for band 3, one continuous, and one dashed for instance. At the moment this is *almost* the case, except for the old band 6 lines.

We will change the line style for the old band 6 to maintain the consistency in the figures.

The same additional coordinate – i.e. latitude – is required in table 4.

We will add the latitude information in table 4.

In Figure 35 I suggest to plot the "old" and "new" traces in reverse order. Right now the "new" trace is almost completely obscured by the "old" trace. Combined with the intensity of the colours of both traces it is somewhat hard to spot the "new" data. A suggestion specifically for figure 35 is to reduce the vertical scale to 0,- 15 km.

We will change the order as suggested (i.e., plot first the "old" and then the "new" trace) and we will reduce the vertical scale of figure 35 to the range 0-15 km.

**[3.11 Caliop]** In section 4.3 on Caliop, there are some confusing statements, please double check them.

1. On line 384 the altitude of Caliop is given as 685 km. On line 385 and 386 Caliop is moved the a *lower* orbit at 688 km. The emphasis is mine.

The information on line 384 is incorrect. The sentence will be corrected to clarify that the original altitude of Caliop was 705 km. Then, Caliop was moved to a lower orbit at 688 km altitude.

2. On line 386 it is stated that "Caliop is a two-wavelength. . . ". Unfortunately Caliop is no longer active since August 1st, 2023. Consider to use the past tense.

We will change the sentence to past tense.

3. On line 460 the time coverage of version 4.21 of the Caliop processing is listed at up to the present. Please update this with the actual availability.

We will update the data availability sentence on line 460.

---

## Community Comment (CC2)

**Replies to RC2 – from 3rd May**

Dear Referee,

we would like to thank you for taking the time to provide us with those constructive comments in a very direct and efficient way. We strongly believe that the revised article will benefit a lot after following your suggestions. Please see our replies to your comments below.

**General comments**

Although the detailed comparison is useful to understand the impact of the changes depending on the cloud situation, I think that the paper would be easier to read when some parts could be shortened and some plots would be combined.

We agree that the article should become shorter and we will follow the referee's suggestions in this effort.

Specifically, for the correlation plots and histograms it is not clear why they are shown or what the reader should learn from them. A qualitative description and an interpretation in the context of the problem would be helpful.

Several comments (both in the "general comments" and "detailed comments" sections) refer to the scatter plots and histograms, by asking why are those important. We believe that scatter plots and histograms are both valuable tools in data analysis. For instance, if the new scheme introduces any outliers, we would be able to spot them with the scatter plot and histogram analysis. Scatter plots are useful for identifying patterns, trends, and correlations between the "old" and "new" scheme. They allow us to see the overall distribution of data points and any clustering or dispersion. Histograms of the differences between "old" and "new" schemes are effective for identifying patterns such as symmetry, skewness, and multimodality in the new data. In the revised version, we will make the plots more condense and informative (e.g., by adding the cloudiness information) and improve their interpretation/explanation in the text.

The grouping in the categories A-K is also difficult to follow for me, the respective parameters between the groups do not show a clear correlation. Please consider discussion the observations more along the groups and combining the barplots.

From several comments understood that the classification into the A-K categories creates more confusion than providing qualitative interpretation of the new results. The main points that we wanted to communicate to the reader through those plots are: (a) The zero absolute differences are in general present under fully cloudy conditions with a cloud fraction higher than 0.8, which shows that the parameters remain unchanged with the new scheme under fully cloudy conditions (i.e., CF>80%). (b) The more negative differences are primarily present in clear sky conditions, which means that the new scheme results into statistically larger cloud fractions under clear sky conditions (i.e., CF<20%). (c) The positive differences are more relevant for the cloudy conditions. In the revised version, we will consider to remove the barplots (along with table 3) from the article and communicate the aforementioned statements through the updated scatter plots and histograms.

**Detailed comments**

C1 – OK, we will shorten the abstract.

C2 – The word "frequently" will be replaced and the sentence will become "The ground pixel footprints of the involved spectral bands should be fully aligned but when they are not, a special treatment is required within the operational algorithms."

C3 – It is actually the UVIS (BD3 and BD4) which we were naming UV/VIS in this article. In the revised version, when we refer to spectral range we will use "UV/VIS" and when we refer to TROPOMI spectrometer we will use "UVIS" according to the L1B documentation.

C4 – Yes, we refer to the Westernmost footprint of each scanline. We will change the term from "first UV detector pixel" to "first (westernmost) TROPOMI ground pixel in UVIS grid".

C5 – The comment probably refers to Table 1 and not Table 2. We will update the table so that it becomes clear that TROPOMI contains 4 spectrometers but each spectrometer is split into 2 bands. Relevant text will be also updated such that there are no inconsistencies of the terminology in the article.

C6 – Thank you very much for the reference. In the cloud retrieval algorithm, we only use BD3, BD4 (from the UVIS spectrometer) and BD6 (from the NIR spectrometer). In this article, we would like to emphasize the mis-registration between two different spectrometers (i.e., UVIS and NIR) and not between two bands within the same spectrometer. We actually consider this mis-registration of two bands negligible in our cloud retrieval algorithm. In the revised article, we will correct the statement that the mis-registration is only caused because of using different spectrometers and add the suggested reference.

C7 – We would like to point out that the mis-alignment is not a fixed number (e.g., 1.5 km shift to the West or East) but analogous to the footprint size coverage on Earth. But the way it is phrased causes confusion and thus in the revised version we will only keep the reference to the spatial resolution in the across-track direction.

C8 – We were referring to the UVIS spectrometer. In the revised version, when we refer to spectral range we will use "UV/VIS" and when we refer to TROPOMI spectrometer we will use "UVIS" according to the L1B documentation.

C9 – We will add that λ is the symbol for the wavelength.

C10 – We prefer to keep only the retrieved cloud parameters in the table to clearly separate them from the other variables used, which originate from other sources like an external climatology or auxiliary datasets.

C11 – We will remove the "Among others," because the S5P L2 CLOUD product is only based on OCRA/ROCINN algorithm. For NO2, there is also the FRESCO algorithm which is used for providing the cloud parameters as input for the trace gas AMF calculations, but FRESCO outputs are not included in the official TROPOMI L2 cloud product.

C12 – We will make clear that we refer to the ground pixel at nadir and the ground pixel footprints at the edges of the swath.

C13 – We will rephrase to: "… it becomes larger towards the edges of the swath due to the Earth's curvature and the instrument large swath angle."

C14 – We will rephrase to: "The so-called binning factors are selected to optimize the signal-to-noise ratio per pixel with the aim to minimize the difference in ground pixel size in across-track direction."

C15 – We will rephrase to: "At the edges of the swath the binning factor is reduced from 2 to 1 in order to keep the ground pixel size at a reasonable value. Due to optical limitations of the instrument and the curvature of the Earth, the ground pixel size at the edges of the swath is about 15 km (KNMI, 2022)."

C16 – Only between BD3 and BD6. We will remove the reference to TROPOMI BD4 from the sentence in line 109.

C17 – The upper panel of Figure 2 refers to the weights (based on the KNMI LUT) from the source BD3 to the target BD6, which is relevant for OCRA. The lower panel refers to the mapping weights from the source BD6 to the target BD3, which is relevant for ROCINN. To our view, both figures need to be included since both mapping directions are necessary within the OCRA/ROCINN algorithm. Mainly the differences are noticeable at the West and East part of the swath (before and after the binning factor change).

C18 – No, this is how we calculate a cloud fraction from VIIRS. VIIRS uses 4 bins/classes: "confidently cloudy", "probably cloudy", "confidently clear" and "probably clear". Then we construct a cloud fraction defined as the ratio of the "confidently cloudy" class over the summation of all classes.

C19 – In the efforts of shortening the article, we will consider to decrease the length of section 3.3.

C20 – Even if the binning factor change happens at the east part of the swath too, we observe (by looking to the upper panel of figure 2) that the "special" case of having three BD3 source pixels contributing to the BD6 target pixel occurs only at pixel #19. We can also visually understand this "special" case if we look in figure 1: the red 19[th] BD6 footprint has an overlap with the three blue 20[th], 21[st] and 22[nd] BD3 footprints. We will make clear this point in the revised text.

C21 – In order to decrease the size of those figures, we will apply a reduction in the vertical spacing of the "source" and "target". The vertical lines are only for illustration purposes to create an effect like the grid of the actual millimeter paper. If there is no grid in the illustration, the reader might have more difficulty to see the vertical alignment of the pixels in the three plots for "imager", "source band" and "target band". Combining all figures will cause an inconsistency problem to our view, as for each equation we have a respective figure clarifying which are the known variables and which the unknown depending on the case (i.e., 2 source pixels contributors, 1 source pixel contributor, 3 source pixel contributors from UVIS to NIR for the cloud fraction and the reverse direction for cloud top height and the other ROCINN parameters).

C22 – We will indicate in the caption that the horizontal lines are expected to be overlapping as the mis-registration is in the across-track direction. We would prefer not to combine the two plots as they refer to different cases and equations.

C23 – We will define H in Section 3.3.

C24 – They are named from 0 to 448 for BD6 and from 0 to 450 for BD3, from West to East. They are defined in the beginning of Section 3 (lines 86-87).

C25 – Yes, because they are just describing a linear regression model (i.e., *alpha* is the regression slope and *beta* the regression intercept). For each scanline, one *alpha-beta* pair is calculated and therefore, it would not make sense to include any numbers of those parameters in the article.

C26 – Yes, because the denominator in the weight calculation (see Eq. 6 and 13) becomes 0. We will rephrase it to: "… when the neighboring VIIRS pixels contain equal values, leading to numerical errors at the weight calculations".

C27 – From fig 11, the reader can learn the following: (a) visually see that the CTH TROPOMI NIR (original parameter) does not contain values in the first magenta ground pixel. (b) notice that the co-registered CTH TROPOMI UV/VIS and original CTH TROPOMI NIR maps are very similar, demonstrating that the new co-registration scheme does not introduce inconsistencies. (c) Compare visually the CTH TROPOMI UV/VIS and CTH VIIRS UV/VIS and observe that the cloud top height values can be very different in absolute numbers but still the weight calculation can be performed smoothly and the new co-registration scheme will not alter the original cloud structures.

C28 – We want to show that the new co-registration will not change the overall statistics (mean, standard deviation and correlation coefficient) for all the retrieved cloud parameters or introduce outliers. However, in our efforts to shorten the length of the article, we will make the content of those figures more condensed and informative (in line to our reply in the general comments).

C29 – The sentence should be rephrased to "... the new scheme is only applicable up to a certain latitude and the pixels around the poles are co-registered with the fallback." The applicability of the new scheme up to a certain latitude is a limitation simply due to the lack of VIIRS cloud optical thickness data in those latitudes. The VIIRS cloud optical thickness originates from the CloudDCOMP (Daytime Cloud Optical and Microphysical Properties) EDR (see Sention 3.3.), whereas the VIIRS cloud fraction and cloud top height originate from other EDRs where valid data points appear in high latitudes.

C30 – We want to show that the new co-registration will not change the overall statistics (mean, standard deviation and correlation coefficient) or introduce outliers. However, in our efforts to shorten the length of the article, we will make the content of those figures more condensed and informative (in line to our reply in the general comments).

C31 – Yes, we have checked this. And indeed, the reader can see it in figure 18. We will rephrase it to: "The differences are exactly zero when VIIRS data are not available because the co-registration is done with the fallback.".

C32 – The differences are split into those 10 symmetric bins around the "zero difference bin" covering the whole accepted range of each parameter. The binning is not done equidistantly because the data points are not equally populated around the median. Thus, a more arbitrary selection of the binning has been performed to ensure that each bin contains "enough pixels" and none of the bins is under-sampled. Nevertheless, in our efforts to shorten the length of the article we will re-consider if figures 22-25 (along with table 3) are very necessary or can be removed from the article.

C33 – We believe that the global maps difference plots for the co-registered parameters should not be removed as they demonstrate that (a) the differences are not systematically present in certain regions but rather spread everywhere, (b) there is not a latitudinal dependence and (c) viewing geometry dependencies are not present. However, in our efforts to shorten the length of the article we will consider to place the global maps for one or two parameters in the main article and move the other maps to the Appendix (e.g., keep figure 18 and move figures 19-21 to the Appendix).

C34 – In our efforts to shorten the length of the article we will question if figures 22-25 are very necessary or can be removed. Please see reply to C32.

C35 – We will rephrase that the gap is reduced by one ground pixel and include a reference why are those gaps present.

C36 – We will rename it to "first (westernmost) TROPOMI ground pixels in the UVIS grid".

C37 – The major improvements with the new co-registration appear at cloud edges and therefore we would like to highlight some example cases. The overall statistics do not change as can be seen from the scatter plots and histograms (figures 13-17).

C38 – We will follow the suggestion.

C39 – We will follow the suggestion.

C40 – In the efforts of shortening the article, we will consider to decrease the length of this section.

C41 – We refer to the forward model used for the training of ROCINN neural networks (since ROCINN was earlier in the article introduced as the operational cloud algorithm for TROPOMI). The text refers to the comparison figures 35-37 against CALIPSO, where the labels "TROPOMI new" and "TROPOMI old" have been used in the figure legends. The forward model for ROCINN is LIDORT; there is a reference and a short description in Section 2.

C42 – The TROPOMI operational cloud parameters are those stored in the S5P Level-2 CLOUD product, which can be accessed from the Copernicus Data Space Ecosystem search tool (https://dataspace.copernicus.eu/). Any other auxiliary cloud data (e.g. FRESCO used in NO2) should not be referred here in our view, as it can be confusing for the TROPOMI users.

C42 – Thank you for the comment and recommendation. We will try to follow this suggestion for the revised version of the article.

**Technical Comments**

General: We will adjust the acronyms and correct the spelling of inconsistent terminology.

We will apply the corrections T1, T4, T5, T11, T12, T13, T14, T15, T16, T17, T18, T19, T20, T22, T23, as suggested.

T2 – We will rephrase to: "The TROPOMI operational cloud algorithms OCRA/ROCINN (Loyola et al., 2018) have a long-standing heritage and have already been applied operationally to a large number…"

T3 – We will rephrase to: "… in near-real-time (NRT) the large data volume of TROPOMI."

T6 – In the revised version, when we refer to spectral range we will use "UV/VIS" and when we refer to TROPOMI spectrometer we will use "UVIS" according to the L1B documentation. See replies to relevant previous comments.

T7 – UV will be corrected to UVIS.

T8 – We will remove the word "From".

T9 – We will replace the word "well" with "successfully".

T10 – In the revised version, when we refer to spectral range we will use "UV/VIS" and when we refer to TROPOMI spectrometer we will use "UVIS" according to the L1B documentation. See replies to relevant previous comments.

T21 – We will rephrase it to: "The old scheme is applied for co-registering the OCRA CF when the VIIRS CF at both UV/VIS and NIR bands are equal to 1."

T24 – We will make clear that only one of the (co-)authors is a member of the editorial board of Atmospheric Measurement Techniques.

References: We will approach ESA and the authors of this document to add it in the Sentinel 5P document library.

Figures: We will follow the suggestion and adapt the colors for people with red-green blindness.

---

## Author Response (AR1)

**Replies to RC1 – from 28[th] April**

Dear Referee,

We would like to thank you very much for your constructive comments, which were helping us to understand the weaknesses of this article. We provide our replies below and we are positive that the revised version of the article will ensure that any inconsistencies are removed, the presentation/interpretation of the results and readability are improved.

**General comments**

We agree that the article should become shorter and we will follow the referee's suggestions in this effort.

The mapping equations are introduced without much justification as to why one would want to use these specific quantities from VIIRS te aid in the transformations. That you use a derived fractional cloud cover from VIIRS to map the cloud fraction from UV-VIS to NIR is fairly logical, but why not use the $\gamma$ factors derived from the VIIRS cloud fraction as well as the dimensionless scaling factor to aid in the mapping of the cloud height from NIR to UV-VIS? After all, you do get a weight that is related to the amount of clouds in the pixel, and therefore a means to obtain a weighted average.

We have never tried to use the γ factors derived from the VIIRS cloud fraction for the co-registration of cloud top height. One main issue that we would have in this approach, is that in fully cloudy scenes the denominator becomes 0 (see Equation 6) and thus we cannot apply the new scheme. When we use the cloud top height property and Equation 13, it is very unlikely that the cloud top height values of the contributing pixels are identical and therefore we can exclude the aforementioned numerical errors.

This brings me to my second more important point. Using the cloud fractions would preclude extrapolation to row 0, but at the same time, how much of the information in row 0 is actually coming from the Tropomi instrument? A discussion on the effective contributions of the information from Tropomi and VIIRS in the final product is desired. Is the information from row 0 really VIIRS information that has been scaled locally by the ratio of Tropomi and VIIRS cloud heights to take into account the systematic model difference between the two instruments and retrieval methods? This discussion really needs to be added to the article before publication.

We would like to provide clarity on how the additional cloud information in row 0 is constructed. We take 15 adjacent (VIIRS, TROPOMI) points in each scanline (located at West part of the swath to eliminate any geometry dependencies in both instruments) and we find the mapping function (e.g., for cloud top height is a linear regression model) between from VIIRS to TROPOMI. After having such a mapping function, we use the first westernmost VIIRS cloud top height to calculate the corresponding value for TROPOMI. Of course, since this is at a certain point an extrapolation using a regression model we cannot consider that the values on those first ground pixels are as accurate as the rest of the pixels, for which L1B radiances are available. Because the information from TROPOMI in row 0 is dependent on how well the VIIRS/TROPOMI mapping function has been constructed in each scanline. Nevertheless, we have reflected this "limitation" in our QA value scheme by adding a penalty for the first row (please refer to L2 CLOUD ATBD). All in all, we have enough evidence that the additional values in the first row are beneficial, in particular since those values have replaced fill values and to our view, a less accurate cloud information is better than no information. Please refer to figure 11 (where we visually see that the CTH TROPOMI NIR (original parameter) does not contain values in the first magenta ground pixel). If we compare the co-registered CTH TROPOMI UV/VIS and original CTH TROPOMI NIR maps, we can be confident that the new co-registration scheme does not introduce inconsistencies or outliers also in the first row. If we visually compare the CTH TROPOMI UV/VIS and CTH VIIRS UV/VIS, we observe that the cloud top height values can be different in absolute numbers. But again, those differences do not alter the original cloud structures.

**Specific comments**

**[2.1 Spatial resolution]** In the introduction, lines 33 to 37, the spatial resolution of Tropomi is discussed. For the bands used by OCRA/ROCINN the spatial resolution in the flight direction is unimportant. Because of the shared entrance slit between the VIS and NIR spectrometers (See L1B ATBD, section 7.2), there is no spatial misalignment between those spectrometers in the flight direction. That also means that the change of the spatial resolution in the flight direction is unimportant for the subject of this article. I suggest to shorten this discussion in the introduction.

We agree with this suggestion.

**[2.2 Algorithm description]** In section 2 of the article a description of the cloud retrieval algorithm is given. The algorithm is based on a pair of algorithms, OCRA for the cloud fraction, and ROCINN for the cloud top height and cloud albedo or cloud optical thickness, depending on the cloud model. Please explain how a cloud fraction can be derived from OCRA without assuming a cloud albedo, or put differently, show that the three parameters are independent, and the cloud albedo (or optical thickness) isn't (implicitly) assumed already in the cloud fraction derivation in OCRA.

An explicit cloud albedo assumption is not made by OCRA. The main assumption of the OCRA algorithm is a wavelength independency of the reflectance of a cloudy spectrum over the considered wavelength ranges, i.e. that the reflectance for a fully cloudy pixel is equal for all considered OCRA colors, resulting in a "white" scene when the reflectances are transferred to color space. Since OCRA provides a radiometric cloud fraction and not a geometric cloud fraction, one could argue that cloud albedo (or cloud optical thickness) has an implicit impact on the retrieved radiometric cloud fraction, because an optically thick (or high albedo) cloud will be easier to be detected due to its higher contrast with the surface background than an optically thin (or low albedo) cloud. However, if the reflectances of a cloudy scene are equal in all considered wavelength ranges, OCRA will report a high cloud fraction regardless if the cloud is optically thick or optically thin. We will expand in the revised manuscript on this basic OCRA assumption of wavelength independency of cloud reflectance.

**[2.3 Spatial alignment in the flight direction]** In section 3, on lines 97 to 99, the authors mention that the static lookup tables used up to now indicate a very small mis-match in the flight directions. These are rounding errors in the algorithm used to calculate these tables. The design of the instrument itself (See L1B ATBD, section 7.2) guarantees that there is no spatial mismatch in the flight direction between the VIS and NIR spectrometers (bands 3 – 6). So the algorithm does not neglect anything, but it handles the situation correctly. Please adjust the text accordingly.

We will rephrase the text so that we reflect that there is no spatial mismatch in the flight direction between UVIS and NIR bands.

**[2.4 Previous treatment of the spatial mis-registration]** In section 3.1 the previous treatment of the mis-registration is described. Here a "cloud co-registration inhomogeneity parameter" is introduced. This parameter involves the difference between the cloud fractions *fci* in the UV band and the cloud fraction *fcj* in the NIR band. The source of the cloud fractions is not indicated. Up to this pont in the article only a cloud fraction in the UV that is derived by OCRA has been described. Please indicate the source for these cloud fractions, especially the source for *fcj* in the NIR. If the cloud fractions are derived from the cloud mask information provided by the SNPP data, please indicate how the cloud fractions are calculated. Note specifically that equation 7 applies to the new method, so it does not apply here.

The cloud co-registration inhomogeneity parameter is a quantity that we have introduced to the L2 CLOUD product to support the identification of areas with cloudiness heterogeneity. The fci in the UV band is the OCRA cloud fraction and fcj is the co-registered fcj in the NIR using the static KNMI LUTs. The SNPP data are not used here.

**[2.5 Unavailable reference]** On line 116 a reference to (Sneep, 2015) is introduced. This is an unpublished and projectinternal document with reference number S5P-KNMI-L2-0129-TN. Either use open literature, or find a way to make this document available to the readers of AMT, for instance by adding it to this article as supplementary material, or arrange for this document to be added to the Sentinel 5P document library.

Many thanks for highlighting this important information to us. We would approach ESA and the authors of this document to add it in the Sentinel 5P document library.

**[2.6 Choice for probably cloudy]** In equation 4 the "probably cloudy" pixels are counted as cloud free. Please elaborate on this choice. The number of pixels in both "probably" classes is small, so the impact is limited, but even that should be mentioned.

We have tried both approaches and we calculated the cloud fraction using only "confidently cloudy" at the nominator and compared to the cloud fraction when using both "confidently cloudy" and "probably cloudy". We found that the difference is minor and we decided to use the first definition.

**[2.7 Optimal Estimation reference]** The authors use (Rodgers, 1976) as a reference for the optimal estimation. This reference points to: Rodgers, C. D.: Retrieval of atmospheric temperature and composition from remote measurements of thermal radiation, Reviews of Geophysics, 14, 609–624, DOI:10.1029/RG014i004p00609, 1976. This is perhaps a little peculiar to use as a reference at this point in time, as the same author wrote the book on optimal estimation: Rodgers, C. D.: Inverse methods for atmospheric sounding. Theory and practice, volume 2 of Atmospheric, Oceanic and Planetary Physics. World Scientific, 2000. DOI:10.1142/3171. I strongly suggest that the reference to the book is used instead of the much earlier article.

We will replace the reference on the article with the suggested book reference.

**[2.8 Cloud optical thickness handling]** In lines 185 to 190 and in equation 5 a transformation is provided to obtain a cloud albedo from a cloud optical thickness. Within the VIIRS mapping software, a similar transformation and its inverse must be used to translate the VIIRS cloud optical thickness into an albedo to be able to take the average, and then the inverse transformation to obtain an effective cloud optical thickness. Please add a statement whether the same transformation is used in the VIIRS remapping software (Siddans, 2026).

To the best of our knowledge, the transformation in the VIIRS mapping software uses the same conversion formula. However, we will need to cross check it with Dr. Richard Siddans. We will add a statement in the revised manuscript whether the same transformation is used in the VIIRS remapping software.

**[2.9 "Missing" three pixel mapping in NIR to UV-VIS scheme]** In section 3.3.2 there is no equivalent to equations 10 – 12 in the UV-VIS to NIR mapping. I would expect such a three pixel equivalent near row 428 because of the symmetry of the binning factors in the Tropomi instrument and the shift between the detectors present in the instrument. A short remark about this absense can resolve the mystery. There is mention of a binning factor change at UV pixel 21, but the same change happens at the other side of the swath. Does that not have an impact, in a way that is similar to the UV-VIS to NIR mapping in section 3.3.1? On lines 224 – 227 exceptions to the "normal" equations (13 and 14) are mentioned. However, for case (a) there is no explicit mention of how this affects the mapping. For clarity I advise to mention that UV/VIS pixel 21 is fully covered by one NIR pixel. The subsequent text makes a lot more sense with that knowledge.

Even if the binning factor change happens at the east part of the swath too, we observe (by looking to the upper panel of figure 2) that the "special" case of having three BD3 source pixels contributing to the BD6 target pixel occurs only at pixel #19. We can also visually understand this "special" case if we look in figure 1: the red 19th BD6 footprint has an overlap with the three blue 20th, 21st and 22nd BD3 footprints. We will make clear this point in the revised text.

We will add the clarification that UVIS pixel 21 is fully covered by one NIR pixel.

**[2.10 Co-registration of cloud albedo and cloud optical thickness]** On lines 240 – 242 the co-registration of cloud albedo and cloud optical thickness is mentioned, and the reader is referred to (Loyola et al., 2023). This is the algorithm theoretical baseline document for the cloud algorithm. In this reference, the only relevant statement to this subject that I can find is the following on page 31 of the reference: The basic principle is that the VIIRS and TROPOMI cloud data are interconnected and therefore, each point from the VIIRS dataset can be mapped to the respective TROPOMI point. The adjacent 15 pairs (HUV c [i]; ZUV c ), for i 2 [2; 17] are used to create the mapping function: ZUV c = fZc HUV c [i] The mapping function for the cloud top height $fZc$ follows a linear regression model for the entire range of the cloud heights. The respective function for the cloud albedo $flc$ and the cloud optical thickness $f\tau c$ is a two-way function (i.e., a combination of a linear model with a logarithmic model). The reference used is insufficient to describe the method used. The statement in the article is essentially repeated without providing substantionally more details. Either complete and update the ATBD, or include an appropriate and reasonably complete description in this article.

We will update the ATBD with a more detailed description. We would prefer not to load this article with additional information.

**[2.11 More appropriate section]** At the start of section 4.1 a listing is provided when the new scheme cannot be used. I think this information should be included in section 3, perhaps in a new sub-section.
We have preferred to place the non-applicability of the scheme in Section 4 and not in previous section 3 because it is only relevant for TROPOMI instrument, whereas the presented methodology in Section 3 is generic and applicable for more Sentinel missions.

**[2.12 Comparison results]** I have several remarks about page 17 – 20 that I will group here.
1. In the caption of figure 12 it is not mentioned which day is shown.
We will add the information on the day in the figure caption.
2. On line 265 it is mentioned that the scheme can be successfully applied up to a certain latitude.
(a) Is this a limit of the latitude, or of the solar zenith angle?
(b) Which of the three reasons that cause failure of the new scheme are a cause of this limitation?
This is a limitation simply due to the lack of VIIRS cloud optical thickness data in those latitudes. The VIIRS cloud optical thickness originates from the CloudDCOMP (Daytime Cloud Optical and Microphysical Properties) EDR (see Sention 3.3.), whereas the VIIRS cloud fraction and cloud top height originate from other EDRs where valid data points appear in high latitudes.

3. On line 267 a sequence of 5 figures is introduced, showing the correlation between the old and the new approach. Which is it obvious after reading the axis labels, please mention that this shows the correlation after the transformation.
We will mention that the figures show the correlation after the transformation.
4. Figures 13 – 17 do not show the units along the axes of the graphs.
We will include the units in the axes of those figures.
5. The axis range of the histograms in figures 13 – 17 is probably too wide given the results, and the number of pixels in the comparison.
We will adjust the x-axis range of the histograms in those figures.
6. The vertical axis of the histograms in figures 13 – 17 is not a count but something else, probably a density. Please adjust the labels.
We will adjust the labels to reflect that the histogram represents the probability density function of the data rather than the raw counts.
7. Indicate the number of pixels in each comparison, and indicate if the data was selected, for instance for filter out pixels that relied on the old scheme.

We will include the number of pixels which are used for the comparisons. The new scheme is applied on top of the old and therefore, we have decided not to apply any filtering on the method. However, the comparison is only made if both "old" and "new" data contain meaningful values. For instance, values corresponding to the first pixel of each scanline are not included in the plots since the "old" data contain fill values there.

**[2.13 Not fully consistent terminology]** When discussing the comparison of the cloud top height results on lines 374 – 375, the following statement is made: "The cloud-top heights at the original BD6 were 7600 m and 9400 m at points A and B respectively. After the co-registration at point A, the CTH at BD3 was 6900 m with the new scheme and 6600 m with the old scheme. At point B, the co-registered CTH at BD3 was 9300 m with the new scheme versus 9000 m with the old." If the cloud top height at points A and B is already known, then why is the new co-registration scheme required? Given the location in the swath, there must be two pixels from band 6 contributing to each of the the result in band 3. The values of both contributing factors to both point A and point B should be mentioned for clarity. The same point applies to table 4 and various other locations throughout the text. In particular the conclusion in lines 442 – 443 should probably be rephrased as well.

We will rephrase the statements on the mentioned lines and add more information in table 4 to exclude this inconsistency that the referee found in the article.

**[2.14 Caliop results]** I strongly suggest to use the Caliop L1 backscatter intensity as a background for figure 35 – 37, as that provides more context, including – sometimes – the presence of multi-layer clouds. Caliop should also have information on the phase of the cloud top, allowing the authors to indicate where this may play a role.

The information on the cloud phase and the presence of multi-layer clouds is also available on the L2 cloud layer "CAL_LID_L2_05kmCLay-Standard-V4-20" product. Thus, we do not need to use the Caliop L1 backscatter intensity for this task.

**[2.15 Suggestion for shortening the article]** The information presented in table 3, and the related figures 22 through 25, as well as the text in lines 284 – 312 are mostly descriptive, and the conclusions drawn from them are not very strong; mostly qualitative rather than quantitative. The article probably improves if this part is removed. Reconsider if appendix A is really required.

In our efforts to shorten the length of the article we will consider referee's suggestions whether (a) Table 3 and figures 22-25 are very necessary and (b) appendix A should be removed.

**[2.16 Appendix B]** Please provide a reference to the method used here. There are several method for calculating a great-circle distance between two points on a globe, with varying accuracy. This appears to be one of the spherical earth methods, rather than Vincenty's formular for an ellipsoid. Any method will be find, they all provide an answer with sufficient accuracy. At the very lease indicate that you are calculating the distance $d$ between the location of the Caliop profile and the points within the selection boxes, and the $R = R_{earth}$. If the equations in appendix B are to be used without reference, at least provide names to the steps for readability.

We do not have a reference included in the article since we have constructed our own approach for the spatial colocation between TROPOMI and CALIPSO data. This is the reason why we have an appendix for the description and not a literature reference. The distance "d" (i.e., given by Equation B4) between the Caliop profile and the middle points for TROPOMI footprints are calculated in a predefined spatial window. Then, the TROPOMI footprint with the smallest distance is selected for the cloud height comparisons. We will provide more information of the calculation steps in Appendix B.

**Technical corrections**

**[3.1 Spectrometer names]** In table 1 on page 2 an overview is provided of the spectrometers in the Tropomi instrument. This table does not make it clear that there are in fact 4 spectrometers on board of the instrument,

with 4 detectors (3 CCD's, 1 CMOS). Each of the detectors is split electronically into two halves, leading to a total of 8 bands. But each pair originating from the same detector share the same spatial coverage, and have *no* interband mis-registration. Bands 1 and 2 are special, because the spatial binning is different for band 1 compared to band 2. But the observations in band 2 can still be co-added on ground to a perfect match to band 1. The suggested solution is to put the spectrometer names centred above the two columns of the bands that they measure, as "UV", "UV/VIS", "NIR", and "SWIR". Note that in the official Tropomi L1B documentation the UV-1 and UV-2 (etc) naming system is not used, see for instance table 1 on page 21 in the L1B ATBD. Using UV/VIS instead of UVIS as the documentation of the Tropomi team does is probably beneficial for clarity here. And of course the references in the text need to be adapted as well. I believe vertical lines are against the design guidelines of the journal, but they may actually be appropriate here for clarity.

We will modify table 1 so that it becomes clear that TROPOMI payload consists of 4 spectrometers (not 8). In the revised version, when we refer to spectral range we will use "UV/VIS" and when we refer to TROPOMI spectrometer we will use "UVIS" according to the L1B documentation.
Moreover, we will remove the vertical lines to cope with the journal guidelines.

**[3.2 Using sun normalized radiances]** In line 55 on page 3 the first instance of "sun-normalized radiances" is encountered. Does the algorithm really use sun-normalized radiances internally, or is the reflectance actually calculated? The difference is a factor $\pi = \mu_0$, and especially look up tables and neural networks typically work better when using reflectances.

The NN is trained with sun-normalized radiances.

**[3.3 Ground pixel footprints]** In figure 1 a section of an orbit is shown, to illustrate the spatial mismatch between the two bands involved in the retrieval. Please indicate the flight direction in this figure.

We will adjust figure 1 such that the flight direction is shown.

**[3.4 The A train]** Line 104: Neither Sentinel 5P, nor Suomi-NPP are part of the A train, see atrain.nasa.gov
We will rephrase the sentence so that the reference to A-train is removed. The sentence will become: "This facilitates the so-called loose formation operation with the SNPP spacecraft, with only 3 to 5 minutes time difference from S5P".

**[3.5 Typography]** In equation 4 on page 8 the superscripts are typeset in L{\sc a}TEX mathematical typesetting mode. This is incorrect, compare the following two versions of the same equation, with the correct method used on the right: $M = \delta_{Confidently Cloudy\ jk}$ $M = \delta_{\text{ConfidentlyCloudy}jk}$ The *amsmath* package is part of the Copernicus document class. This means that the \text{} macro is available to typeset fragments that are text within an equation. This improves the readability. It isn't just the italics, but especially the spacing that is different. Please use \text{} in all mathematical notation where a superscript or subscript is not an index or exponent but an abbreviation or a name. The symbol $f_{c\text{NIR}_{\text{old}}}$ can be coded as

        $f_c^{\text{NIR}_{\text{old}}}$.

We will use the \text{} command for equation 4, as suggested by the referee.

**[3.6 Figure size]** While the schematic overview in figures 3, 5, 6, 8, 9, and 10 is much appreciated in explaining the method, the vertical spacing of these seem rather large. Reducing the vertical spacing of the "source" and "target" panels is likely to provide a better spacing in the article, and allow for easier reference to the figures while reading the description of the method in section 3.3.1, especially for the baseline case (equations 6 and 7).
We will apply a reduction in the vertical spacing of the "source" and "target" panels for the mentioned figures.

**[3.7 Readability of equation 12]** Suggestion to split the contribution of $\gamma_1$ and $\gamma_2$ to emphasize the symmetry on the equation.
We will apply this suggestion.

**[3.8 Missing introduction of the Hc symbol]** In equation the symbols $H_{UV\,c}$ and $H_{NIR\,c}$ are used, but this symbol is never properly introduced. Extending table 2 to also introduce the symbols for the quantities taken from VIIRS is a possible solution.

Extending table 2 with including those symbols seems not an optimal solution for us since Table 2 contains cloud parameters which are retrieved from the OCRA/ROCINN algorithms. We will introduce the symbols properly in Section 3.3.

**[3.9 Language]** I'm not a native speaker, but the following sentences (line 314 – 316) do not seem to be correct English, or at least a slight rephrasing may make them more readable: "The first TROPOMI UV detector pixels were lacking of cloud height and cloud optical thickness properties up to now due to the missing TROPOMI NIR overlap detector pixels. The use of VIIRS data made possible to re-construct the UV/VIS cloud height information for these pixels through the mapping linear function of Equation 17."

We will rephrase the sentences as: "The first (westernmost) TROPOMI ground pixels in the UVIS grid were lacking any information about the cloud height and cloud optical thickness properties up to now. The use of VIIRS data made possible the reconstruction of cloud information for these pixels in the UVIS grid through the mapping linear function of Equation 17."

I'll group this under language as well: on lines 324, 325, 327, and 487 the chemical symbol for ozone is given as "O3" rather than "$O_3$".

We will adjust the ozone symbol.

In the conclusions, on line 421 – 422, "... from UV/VIS to NIR with the new scheme results to a lower cloud fraction compared to the old scheme ..." should probably be "... from UV/VIS to NIR with the new scheme results *in* a lower cloud fraction compared to the old scheme ...".

We will apply the suggestion.

In the conclusions, on line 432 – 433, "... allows the restructuring of the ROCINN retrieved parameters...", I would suggest "... allows the reconstruction of the ROCINN retrieved parameters...".

We will apply the suggestion.

In the conclusions, on lines 444 – 445, "The old co-registration scheme is applied when the old scheme for co-registering the OCRA CF when the VIIRS CF at both UV/VIS and NIR bands are equal to 1.". Please check and rephrase, there is a repeated part in this sentence.

The sentence should be: "The old scheme is applied for co-registering the OCRA CF when the VIIRS CF at both UV/VIS and NIR bands are equal to 1."

**[3.10 Figures 26, 27, 30 through 37, and table 4]** While the difference between figures 26 and 27 is obvious if you can switch between them on a computer screen, if they are on adjacent pages and share the same position on the page, once printed it is nearly impossible to spot the difference between these two figures. This is an issue either for the journal, to make sure that these pages end up in a location where switching between them is possible (as in the current pre-print), or to combine these figures in a three pane single figure, showing both figures and adding a difference plot.

Adding a difference plot with actual values is not possible since there are no valid data for the first ground pixels in the old version. But we can combine both figures, zoom into the plot around 144E - 152E and highlight the additional available pixels.

In figure 30 there are 6 pixels that are highlighted. At the printed scale these highlights are hardly visible. Either zoom in further or select another display technique. A zoomed in inset in the top-left of the figure may be an option.

We will adjust figure 30 by zooming in the region of interest.

In figures 31 through 37 the locations are displayed only with their longitude *or* their latitude. Please add a second axis with the other coordinate of the data points.

We will add the second axis with the other coordinate as suggested.

Given the importance of the 20 % cloud fraction for trace gas retrievals, I suggest to add a thin horizontal line to figures 30 through 33 to indicate that level as a visual guide to see the impact of the new scheme. The 5 % level is important for the ROCINN retrieval, and may require an additional grid line in these figures.

We will add the two suggested horizontal lines (i.e., one at 20% cloud fraction and a second at 5% cloud fraction) in figures 30-33.

For clarity I suggest to use a consistent line style for band 6 and a different, but also consistent one for band 3, one continuous, and one dashed for instance. At the moment this is *almost* the case, except for the old band 6 lines.

We will change the line style for the old band 6 to maintain the consistency in the figures.

The same additional coordinate – i.e. latitude – is required in table 4.

We will add the latitude information in table 4.

In Figure 35 I suggest to plot the "old" and "new" traces in reverse order. Right now the "new" trace is almost completely obscured by the "old" trace. Combined with the intensity of the colours of both traces it is somewhat hard to spot the "new" data. A suggestion specifically for figure 35 is to reduce the vertical scale to 0,- 15 km.

We will change the order as suggested (i.e., plot first the "old" and then the "new" trace) and we will reduce the vertical scale of figure 35 to the range 0-15 km.

**[3.11 Caliop]** In section 4.3 on Caliop, there are some confusing statements, please double check them.

1. On line 384 the altitude of Caliop is given as 685 km. On line 385 and 386 Caliop is moved the a *lower* orbit at 688 km. The emphasis is mine.

The information on line 384 is incorrect. The sentence will be corrected to clarify that the original altitude of Caliop was 705 km. Then, Caliop was moved to a lower orbit at 688 km altitude.

2. On line 386 it is stated that "Caliop is a two-wavelength...". Unfortunately Caliop is no longer active since August 1st, 2023. Consider to use the past tense.

We will change the sentence to past tense.

3. On line 460 the time coverage of version 4.21 of the Caliop processing is listed at up to the present. Please update this with the actual availability.

We will update the data availability sentence on line 460.

**Replies to RC2 – from 3rd May**

Dear Referee,

we would like to thank you for taking the time to provide us with those constructive comments in a very direct and efficient way. We strongly believe that the revised article will benefit a lot after following your suggestions. Please see our replies to your comments below.

**General comments**

Although the detailed comparison is useful to understand the impact of the changes depending on the cloud situation, I think that the paper would be easier to read when some parts could be shortened and some plots would be combined.

We agree that the article should become shorter and we will follow the referee's suggestions in this effort.

Specifically, for the correlation plots and histograms it is not clear why they are shown or what the reader should learn from them. A qualitative description and an interpretation in the context of the problem would be helpful.

Several comments (both in the "general comments" and "detailed comments" sections) refer to the scatter plots and histograms, by asking why are those important. We believe that scatter plots and histograms are both valuable tools in data analysis. For instance, if the new scheme introduces any outliers, we would be able to spot them with the scatter plot and histogram analysis. Scatter plots are useful for identifying patterns, trends, and correlations between the "old" and "new" scheme. They allow us to see the overall distribution of data points and any clustering or dispersion. Histograms of the differences between "old" and "new" schemes are effective for identifying patterns such as symmetry, skewness, and multimodality in the new data. In the revised version, we will make the plots more condense and informative (e.g., by adding the cloudiness information) and improve their interpretation/explanation in the text.

The grouping in the categories A-K is also difficult to follow for me, the respective parameters between the groups do not show a clear correlation. Please consider discussion the observations more along the groups and combining the barplots.

From several comments understood that the classification into the A-K categories creates more confusion than providing qualitative interpretation of the new results. The main points that we wanted to communicate to the reader through those plots are: (a) The zero absolute differences are in general present under fully cloudy conditions with a cloud fraction higher than 0.8, which shows that the parameters remain unchanged with the new scheme under fully cloudy conditions (i.e., CF>80%). (b) The more negative differences are primarily present in clear sky conditions, which means that the new scheme results into statistically larger cloud fractions under clear sky conditions (i.e., CF<20%). (c) The positive differences are more relevant for the cloudy conditions. In the revised version, we will consider to remove the barplots (along with table 3) from the article and communicate the aforementioned statements through the updated scatter plots and histograms.

**Detailed comments**

C1 – OK, we will shorten the abstract.

C2 – The word "frequently" will be replaced and the sentence will become "The ground pixel footprints of the involved spectral bands should be fully aligned but when they are not, a special treatment is required within the operational algorithms."

C3 – It is actually the UVIS (BD3 and BD4) which we were naming UV/VIS in this article. In the revised version, when we refer to spectral range we will use "UV/VIS" and when we refer to TROPOMI spectrometer we will use "UVIS" according to the L1B documentation.

C4 – Yes, we refer to the Westernmost footprint of each scanline. We will change the term from "first UV detector pixel" to "first (westernmost) TROPOMI ground pixel in UVIS grid".

C5 – The comment probably refers to Table 1 and not Table 2. We will update the table so that it becomes clear that TROPOMI contains 4 spectrometers but each spectrometer is split into 2 bands. Relevant text will be also updated such that there are no inconsistencies of the terminology in the article.

C6 – Thank you very much for the reference. In the cloud retrieval algorithm, we only use BD3, BD4 (from the UVIS spectrometer) and BD6 (from the NIR spectrometer). In this article, we would like to emphasize the mis-registration between two different spectrometers (i.e., UVIS and NIR) and not between two bands within the same spectrometer. We actually consider this mis-registration of two bands negligible in our cloud retrieval algorithm. In the revised article, we will correct the statement that the mis-registration is only caused because of using different spectrometers and add the suggested reference.

C7 – We would like to point out that the mis-alignment is not a fixed number (e.g., 1.5 km shift to the West or East) but analogous to the footprint size coverage on Earth. But the way it is phrased causes confusion and thus in the revised version we will only keep the reference to the spatial resolution in the across-track direction.

C8 – We were referring to the UVIS spectrometer. In the revised version, when we refer to spectral range we will use "UV/VIS" and when we refer to TROPOMI spectrometer we will use "UVIS" according to the L1B documentation.

C9 – We will add that λ is the symbol for the wavelength.

C10 – We prefer to keep only the retrieved cloud parameters in the table to clearly separate them from the other variables used, which originate from other sources like an external climatology or auxiliary datasets.

C11 – We will remove the "Among others," because the S5P L2 CLOUD product is only based on OCRA/ROCINN algorithm. For NO2, there is also the FRESCO algorithm which is used for providing the cloud parameters as input for the trace gas AMF calculations, but FRESCO outputs are not included in the official TROPOMI L2 cloud product.

C12 – We will make clear that we refer to the ground pixel at nadir and the ground pixel footprints at the edges of the swath.

C13 – We will rephrase to: "… it becomes larger towards the edges of the swath due to the Earth's curvature and the instrument large swath angle."

C14 – We will rephrase to: "The so-called binning factors are selected to optimize the signal-to-noise ratio per pixel with the aim to minimize the difference in ground pixel size in across-track direction."

C15 – We will rephrase to: "At the edges of the swath the binning factor is reduced from 2 to 1 in order to keep the ground pixel size at a reasonable value. Due to optical limitations of the instrument and the curvature of the Earth, the ground pixel size at the edges of the swath is about 15 km (KNMI, 2022)."

C16 – Only between BD3 and BD6. We will remove the reference to TROPOMI BD4 from the sentence in line 109.

C17 – The upper panel of Figure 2 refers to the weights (based on the KNMI LUT) from the source BD3 to the target BD6, which is relevant for OCRA. The lower panel refers to the mapping weights from the source BD6 to the target BD3, which is relevant for ROCINN. To our view, both figures need to be included since both mapping directions are necessary within the OCRA/ROCINN algorithm. Mainly the differences are noticeable at the West and East part of the swath (before and after the binning factor change).

C18 – No, this is how we calculate a cloud fraction from VIIRS. VIIRS uses 4 bins/classes: "confidently cloudy", "probably cloudy", "confidently clear" and "probably clear". Then we construct a cloud fraction defined as the ratio of the "confidently cloudy" class over the summation of all classes.

C19 – In the efforts of shortening the article, we will consider to decrease the length of section 3.3.

C20 – Even if the binning factor change happens at the east part of the swath too, we observe (by looking to the upper panel of figure 2) that the "special" case of having three BD3 source pixels contributing to the BD6 target pixel occurs only at pixel #19. We can also visually understand this "special" case if we look in figure 1: the red 19[th] BD6 footprint has an overlap with the three blue 20[th], 21[st] and 22[nd] BD3 footprints. We will make clear this point in the revised text.

C21 – In order to decrease the size of those figures, we will apply a reduction in the vertical spacing of the "source" and "target". The vertical lines are only for illustration purposes to create an effect like the grid of the actual millimeter paper. If there is no grid in the illustration, the reader might have more difficulty to see the vertical alignment of the pixels in the three plots for "imager", "source band" and "target band". Combining all figures will cause an inconsistency problem to our view, as for each equation we have a respective figure clarifying which are the known variables and which the unknown depending on the case (i.e., 2 source pixels contributors, 1 source pixel contributor, 3 source pixel contributors from UVIS to NIR for the cloud fraction and the reverse direction for cloud top height and the other ROCINN parameters).

C22 – We will indicate in the caption that the horizontal lines are expected to be overlapping as the mis-registration is in the across-track direction. We would prefer not to combine the two plots as they refer to different cases and equations.

C23 – We will define H in Section 3.3.

C24 – They are named from 0 to 448 for BD6 and from 0 to 450 for BD3, from West to East. They are defined in the beginning of Section 3 (lines 86-87).

C25 – Yes, because they are just describing a linear regression model (i.e., *alpha* is the regression slope and *beta* the regression intercept). For each scanline, one *alpha-beta* pair is calculated and therefore, it would not make sense to include any numbers of those parameters in the article.

C26 – Yes, because the denominator in the weight calculation (see Eq. 6 and 13) becomes 0. We will rephrase it to: "… when the neighboring VIIRS pixels contain equal values, leading to numerical errors at the weight calculations".

C27 – From fig 11, the reader can learn the following: (a) visually see that the CTH TROPOMI NIR (original parameter) does not contain values in the first magenta ground pixel. (b) notice that the co-registered CTH TROPOMI UV/VIS and original CTH TROPOMI NIR maps are very similar, demonstrating that the new co-registration scheme does not introduce inconsistencies. (c) Compare visually the CTH TROPOMI UV/VIS and CTH VIIRS UV/VIS and observe that the cloud top height values can be very different in absolute numbers but still the weight calculation can be performed smoothly and the new co-registration scheme will not alter the original cloud structures.

C28 – We want to show that the new co-registration will not change the overall statistics (mean, standard deviation and correlation coefficient) for all the retrieved cloud parameters or introduce outliers. However, in our efforts to shorten the length of the article, we will make the content of those figures more condensed and informative (in line to our reply in the general comments).

C29 – The sentence should be rephrased to "… the new scheme is only applicable up to a certain latitude and the pixels around the poles are co-registered with the fallback." The applicability of the new scheme up to a certain latitude is a limitation simply due to the lack of VIIRS cloud optical thickness data in those latitudes. The VIIRS cloud optical thickness originates from the CloudDCOMP (Daytime Cloud Optical and Microphysical Properties) EDR (see Sention 3.3.), whereas the VIIRS cloud fraction and cloud top height originate from other EDRs where valid data points appear in high latitudes.

C30 – We want to show that the new co-registration will not change the overall statistics (mean, standard deviation and correlation coefficient) or introduce outliers. However, in our efforts to shorten the length of the article, we will make the content of those figures more condensed and informative (in line to our reply in the general comments).

C31 – Yes, we have checked this. And indeed, the reader can see it in figure 18. We will rephrase it to: "The differences are exactly zero when VIIRS data are not available because the co-registration is done with the fallback.".

C32 – The differences are split into those 10 symmetric bins around the "zero difference bin" covering the whole accepted range of each parameter. The binning is not done equidistantly because the data points are not equally populated around the median. Thus, a more arbitrary selection of the binning has been performed to ensure that each bin contains "enough pixels" and none of the bins is under-sampled. Nevertheless, in our efforts to shorten the length of the article we will re-consider if figures 22-25 (along with table 3) are very necessary or can be removed from the article.

C33 – We believe that the global maps difference plots for the co-registered parameters should not be removed as they demonstrate that (a) the differences are not systematically present in certain regions but rather spread everywhere, (b) there is not a latitudinal dependence and (c) viewing geometry dependencies are not present. However, in our efforts to shorten the length of the article we will consider to place the global maps for one or two parameters in the main article and move the other maps to the Appendix (e.g., keep figure 18 and move figures 19-21 to the Appendix).

C34 – In our efforts to shorten the length of the article we will question if figures 22-25 are very necessary or can be removed. Please see reply to C32.

C35 – We will rephrase that the gap is reduced by one ground pixel and include a reference why are those gaps present.

C36 – We will rename it to "first (westernmost) TROPOMI ground pixels in the UVIS grid".

C37 – The major improvements with the new co-registration appear at cloud edges and therefore we would like to highlight some example cases. The overall statistics do not change as can be seen from the scatter plots and histograms (figures 13-17).

C38 – We will follow the suggestion.

C39 – We will follow the suggestion.

C40 – In the efforts of shortening the article, we will consider to decrease the length of this section.

C41 – We refer to the forward model used for the training of ROCINN neural networks (since ROCINN was earlier in the article introduced as the operational cloud algorithm for TROPOMI). The text refers to the comparison figures 35-37 against CALIPSO, where the labels "TROPOMI new" and "TROPOMI old" have been used in the figure legends. The forward model for ROCINN is LIDORT; there is a reference and a short description in Section 2.

C42 – The TROPOMI operational cloud parameters are those stored in the S5P Level-2 CLOUD product, which can be accessed from the Copernicus Data Space Ecosystem search tool (https://dataspace.copernicus.eu/). Any other auxiliary cloud data (e.g. FRESCO used in NO2) should not be referred here in our view, as it can be confusing for the TROPOMI users.

C42 – Thank you for the comment and recommendation. We will try to follow this suggestion for the revised version of the article.

**Technical Comments**

General: We will adjust the acronyms and correct the spelling of inconsistent terminology.

We will apply the corrections T1, T4, T5, T11, T12, T13, T14, T15, T16, T17, T18, T19, T20, T22, T23, as suggested.

T2 – We will rephrase to: "The TROPOMI operational cloud algorithms OCRA/ROCINN (Loyola et al., 2018) have a long-standing heritage and have already been applied operationally to a large number..."

T3 – We will rephrase to: "... in near-real-time (NRT) the large data volume of TROPOMI."

T6 – In the revised version, when we refer to spectral range we will use "UV/VIS" and when we refer to TROPOMI spectrometer we will use "UVIS" according to the L1B documentation. See replies to relevant previous comments.

T7 – UV will be corrected to UVIS.

T8 – We will remove the word "From".

T9 – We will replace the word "well" with "successfully".

T10 – In the revised version, when we refer to spectral range we will use "UV/VIS" and when we refer to TROPOMI spectrometer we will use "UVIS" according to the L1B documentation. See replies to relevant previous comments.

T21 – We will rephrase it to: "The old scheme is applied for co-registering the OCRA CF when the VIIRS CF at both UV/VIS and NIR bands are equal to 1."

T24 – We will make clear that only one of the (co-)authors is a member of the editorial board of Atmospheric Measurement Techniques.

References: We will approach ESA and the authors of this document to add it in the Sentinel 5P document library.

Figures: We will follow the suggestion and adapt the colors for people with red-green blindness.

---

## Referee Report (RR1)

**Review of "An advanced spatial co-registration of cloud properties for the atmospheric Sentinel missions: Application to TROPOMI"**

Comments based on

https://editor.copernicus.org/index.php?_mdl=msover_md&_jrl=400&_lcm=oc73lcm74a&_acm=get_manuscript_file&_ms=118338&id=2395282&salt=1973031979915874526 , retrieved 1 August 2024

**General comments**

Dear authors,

It is very clear that you have put a lot of effort in revising and shortening the paper. The manuscript is now much more concise and therefore it is easier for the reader to follow the main results. In your answers you have addressed all my previous comments in a satisfactory way. However, during your revision some small oversights occurred:

**Detailed comments**

Please find detailed comments in the table below.

| # | Page | Line | Section | Comment |
|---|---|---|---|---|
| C4 |  , 4 23 | , 92, Fig 18 |  3, Fig 18 | There are still two occasions where ground and detector pixels are mixed up |

**Technical comments/typos**

On the second reading I found a minor point (T1) which I did not notice in my first review:

| # | Page | Line | Section | Comment |
|---|---|---|---|---|
| T1 | 1 | 18 | Abs | "Across-track flight direction" should be "across-track" or "across-flight" but not both |
| T2 | 2 | 30 | Intro | Here you still refer to UV-1 UV-2… , Is this an oversight? The table has been updated with the band numbering and spectrometer names. |
| T4 | 14,28 | , 275, 412 | , 4.1, 5 | On top -> in addition to |
| T5 | 5 | Fig 1 | |  →  here: VIS-1/NIR-2 spectrometer is still not correct |